# Mitochondrial fatty acid oxidation regulates adult muscle stem cell function through modulating metabolic flux and protein acetylation

Feng Yue [1,2,6]✉, Lijie Gu [1,6], Jiamin Qiu [1,6], Stephanie N Oprescu[1,3], Linda M Beckett[1], Jessica M Ellis[4], Shawn S Donkin [1] & Shihuan Kuang [1,5]✉

## Abstract

**During homeostasis and regeneration, satellite cells, the resident stem cells of skeletal muscle, have distinct metabolic requirements for fate transitions between quiescence, proliferation and differentiation. However, the contribution of distinct energy sources to satellite cell metabolism and function remains largely unexplored. Here, we uncover a role of mitochondrial fatty acid oxidation (FAO) in satellite cell integrity and function. Single-cell RNA sequencing revealed progressive enrichment of mitochondrial FAO and downstream pathways during activation, proliferation and myogenic commitment of satellite cells. Deletion of *Carnitine palmitoyltransferase 2* (*Cpt2*), the rate-limiting enzyme in FAO, hampered muscle stem cell expansion and differentiation upon acute muscle injury, markedly delaying regeneration. *Cpt2* deficiency reduces acetyl-CoA levels in satellite cells, impeding the metabolic flux and acetylation of selective proteins including Pax7, the central transcriptional regulator of satellite cells. Notably, acetate supplementation restored cellular metabolic flux and partially rescued the regenerative defects of *Cpt2*-null satellite cells. These findings highlight an essential role of fatty acid oxidation in controlling satellite cell function and suggest an integration of lipid metabolism and protein acetylation in adult stem cells.**

**Keywords** CPT2; Fatty Acid Oxidation; Muscle Satellite Cell; Protein Acetylation; Muscle Regeneration
**Subject Categories** Metabolism; Musculoskeletal System; Stem Cells & Regenerative Medicine

## Introduction

Skeletal muscle homeostasis and regeneration depend on resident muscle stem cells, known as satellite cells (SCs) (Lepper et al, 2011; Sambasivan et al, 2011). SCs possess a remarkable capacity for expansion and long-term self-renewal, which is essential for lifelong muscle maintenance and repair (Chakkalakal et al, 2012; Sacco et al, 2008; Yin et al, 2013). Under normal, homeostatic conditions, SCs remain in a mitotically quiescent state (Kuang et al, 2008). However, they can be rapidly activated in response to injury, transitioning into a proliferative phase (Yin et al, 2013). After expansion, most SCs are committed to differentiate and repair the damaged myofibers, while a small subset self-renews to replenish the stem cell pool (Kuang et al, 2008). The transitions from quiescence to activation, proliferation and the subsequent initiation of the myogenic program is regulated by the sequential expression of key muscle-specific transcription factors, including Pax7, Myf5, MyoD and myogenin (Günther et al, 2013; Seale et al, 2000; Zammit, 2017; Zammit et al, 2006). This core transcriptional program is further influenced by both extrinsic (niche) and intrinsic signals, such as Notch (Baghdadi et al, 2018; Bjornson et al, 2012; Mourikis et al, 2012; Verma et al, 2018; Wen et al, 2012), Wnt (Brack et al, 2007; Eliazer et al, 2019; Le Grand et al, 2009), p38/MAPK (Troy et al, 2012) and components of the extracellular matrix (Chang et al, 2018; Dumont et al, 2015; Urciuolo et al, 2013). Recent studies have also highlighted the crucial role of nutrient sensing pathways, such as the LKB1/AMPK axis (Fu et al, 2015; Shan et al, 2014; Theret et al, 2017), as well as the PI3K/AKT/mTOR metabolic pathway (Rodgers et al, 2014; Wang et al, 2018; Yue et al, 2017) in regulating SC behavior.

Metabolic plasticity has been implicated in adult stem cell homeostasis and fate determination (Folmes et al, 2012; Ghosh-Choudhary et al, 2020; Ito et al, 2014; Ryall et al, 2015a). A recent study showed that quiescent hematopoietic stem cells (HSCs) express higher levels of glycolytic enzymes but lower levels of oxidative phosphorylation (OxPhos) proteins compared to their activated counterparts (Takubo et al, 2013). However, fatty acid β-oxidation (FAO) is required for quiescence and maintenance of HSCs, as deletion of *PPARδ*, which controls FAO in HSCs, or pharmacological inhibition of FAO results in the symmetric commitment of newly divided HSCs and stem cell depletion (Ito et al, 2012). Moreover, quiescent neural stem/progenitor cells

[1]Department of Animal Sciences, Purdue University, West Lafayette, IN 47907, USA. [2]Department of Animal Sciences, University of Florida, Gainesville, FL 32611, USA. [3]Department of Biological Sciences, Purdue University, West Lafayette, IN 47907, USA. [4]East Carolina Diabetes and Obesity Institute and Department of Physiology, East Carolina University, Greenville, NC 27858, USA. [5]Department of Orthopaedic Surgery, Duke University School of Medicine, Durham, NC 27710, USA. [6]These authors contributed equally: Feng Yue, Lijie Gu, Jiamin Qiu. ✉E-mail: fengyue@ufl.edu; shihuan.kuang@duke.edu

(NSPCs) exhibit high levels of FAO activity, which is down-regulated in proliferating NSPCs (Knobloch et al, 2017). FAO is required for sustaining quiescence of NSPCs since deletion of carnitine palmitoyltransferase 1a (*Cpt1a*), a gene encoding the first-step enzyme of FAO leads to quiescence exit and increased expansion of NSPCs (Knobloch et al, 2017). In contrast, fasting activates FAO through Cpt1a to enhance the proliferation and regenerative capacity of intestinal stem cell (ISC) (Mihaylova et al, 2018). The known pleiotropic effects suggest that cellular FAO metabolism plays essential but distinct cell context-dependent roles in stem cell function.

Similar to other adult stem cells, the transitions of different cell states in SCs are associated with alterations in metabolic programs (Pala et al, 2018). It has been reported that SCs switch from oxidative to glycolytic metabolism during the transition from quiescence to proliferation (Ryall et al, 2015b). However, recent studies based on transcriptomics analysis showed that the main energy-producing pathways including glycolysis, FAO and OxPhos are all enriched in proliferating relative to quiescent SCs (Dell'Orso et al, 2019; Pala et al, 2018). Moreover, systemic inhibition of FAO using pharmacological inhibitors suggests that peroxisomal instead of mitochondrial FAO was required for myogenic cell differentiation and muscle regeneration (Pala et al, 2018). However, whether and how specific metabolic pathways regulate SC fate transitions and function remains to be determined by cell-specific targeting.

Here, by analyzing the single cell RNA sequencing data we show that mitochondrial FAO is enriched in activated and committed SCs during muscle regeneration. Conditional knockout of *Cpt2*, a gene encoding the rate-limiting enzyme of mitochondrial FAO, does not affect the maintenance of quiescent SCs in resting muscle, but impedes their proliferation and differentiation upon muscle injury, leading to a regenerative defect. *Cpt2* deletion alters the metabolic flux of SCs, causes energy insufficient and reduces cellular acetyl-CoA and protein acetylation of Pax7, altering Pax7 transcriptional function. Accordingly, supplementation of acetate restores the metabolic flux and energy production, and partially rescues the regenerative deficiency of *Cpt2*-null SCs. Our study for the first time demonstrates that mitochondrial FAO is required for SC function by controlling energy metabolism and protein acetylation.

## Results

### Mitochondrial FAO is coupled to SC expansion and commitment during muscle regeneration

SCs are highly heterogeneous and exhibit dynamic cell state transitions during muscle regeneration. To capture metabolic dynamics during SC fate decision and transition, we analyzed our single cell RNA-seq (scRNA-seq) data containing 3995 Pax7-lineage SCs isolated from non-injured and injured tibialis anterior (TA) muscles at 5.5- and 10-days post injury (dpi) (Yue et al, 2022). Using the Uniform Manifold Approximation and Projection (UMAP) tool, the cells were clustered in six unique subsets that were identified as quiescent (QSCs), self-renewal (SSCs), activated (ASCs), proliferating (PSCs), committed (CSCs) and differentiating (DSCs) SCs based on their gene expression signatures (top 20–30 DEGs) (Figs. 1A and EV1). The dataset confirms known SC subpopulations and reveals unique transcriptional features of

homeostatic, regenerating and recovering SC states throughout the regenerative process (Oprescu et al, 2020).

To understand whether specific metabolic programs were correlated with the transitions of SC fate, we evaluated the expression of all genes under Gene Ontology (GO) terms associated with major metabolic pathways. The gene expression heatmap and density plot analysis showed a progressive enrichment of "tricarboxylic acid cycle (TCA)", "electron transport chain (ETC)" and "OxPhos" pathways in QSCs, SSCs, ASCs, PSCs, CSCs and DSCs (Figs. 1B and EV2A), suggesting increasingly higher energetic requirements during myogenic progression of SCs. To determine the potential nutrient sources contributing to the energy production, we analyzed the metabolic pathways underlying glucose, FA and amino acid processing. Overall, genes of the "total glycolytic process" were enriched in DSCs, but not enriched clearly in other states (Fig. EV2A,B). Violin plots of representative glycolysis marker genes revealed that compared to quiescent and self-renewed SCs, the expression of *Pgam1* and *Pkm* was enriched in PSCs, *Pfkl* was enriched in PSCs and CSCs, and *Aldoa* and *Pgk1* was enriched in CSCs and DSCs (Fig. EV2C). Interestingly, genes of "acetyl-CoA biosynthetic process from pyruvate" (indicative of aerobic glycolysis) were markedly increased in CSCs and DSCs. These include genes encoding components of pyruvate dehydro-genase complex (PDC) such as *Pdha1*, *Pdhb*, *Dlat*, and *Dld* (Fig. EV2B,C). No obvious changes were observed for genes (for example, *Ldha*) related to "lactate production" (indicating anaerobic glycolysis) among different cell states (Fig. EV2B,C). These results indicate the different gene expression patterns associated with aerobic and anaerobic glycolysis which might be essential for SC differentiation.

Notably, we found that genes related to "mitochondrial FAO", but not "peroxisomal FAO", uniquely mirrored the dynamic changes of TCA, ETC, and OxPhos pathways during SC fate transitions (Figs. 1B,C and EV2A,D). Specifically, the key mitochondrial FAO-related genes *Cpt1a* and *Cpt1c* were expressed in all SC states, but enriched in ASCs (Fig. 1D), while *Cpt2, CACT* and many other FAO-related genes were highly expressed in ASCs, PSCs, CSCs and DSCs but not in QSCs and SSCs (Figs. 1D and EV2E). These observations highlight the potential role of mitochondrial FAO in SC expansion and commitment; and indicate fatty acids as a major energy source supporting the regenerative function of SCs.

To confirm the above observations, we performed qPCR analysis for *Cpt2* expression on SCs sorted from non-injured (QSCs) and cardiotoxin (CTX) injured (ASCs) muscle. Consist with our scRNA-seq data, *Cpt2* expression was nearly tenfold higher in ASCs compared to QSCs (Fig. 1E). We also examined Cpt2 protein expression in SCs grown on single extensor digitorum longus (EDL) myofibers. Cpt2 immunofluorescence was undetectable in Pax7[+] QSCs from freshly isolated (Day 0) EDL myofibers (Fig. 1F). After culturing for 1–3 days, Cpt2 signal appeared high in ASCs on Day 1, and remained high in SC clusters on Day 3, which contained both PSCs and DSCs (Fig. 1F). Additionally, qRT-PCR and western blot analysis of SC-derived primary myoblasts revealed that Cpt2 expression was lower in proliferating myoblasts, but was rapidly upregulated upon the induction of differentiation (Fig. 1G,H). Collectively, these findings suggest that the expansion and commitment of SCs are associated with increased expression of genes and proteins related to the mitochondrial FAO pathway.

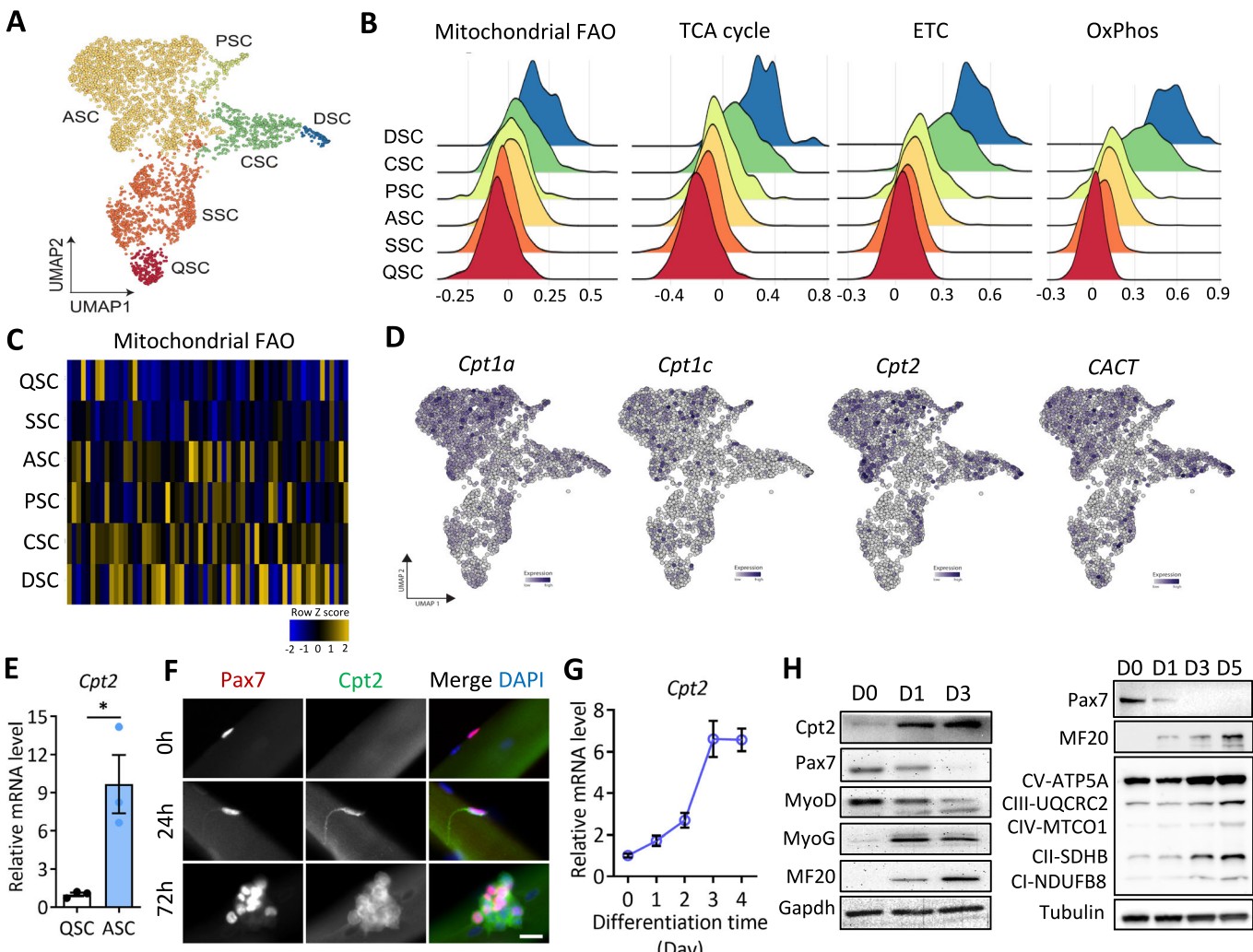

**Figure 1. Sequential upregulation of genes involved in FAO during activation, proliferation, commitment and differentiation of muscle satellite cells (SCs).**

(A) UMAP clustering of 3995 SCs isolated from non-injured and injured hindlimb muscles at 5- and 10-day post injury, based on single cell RNA-seq data ($n = 3$ mice). Clusters were annotated based on gene expression signatures: QSC quiescent SCs, SSC self-renewed SCs, ASC activated SCs, PSC proliferating SCs, CSC committed SCs, DSC differentiated SCs. (B) Density plot visualizing enrichment of genes involved in FAO and oxidative metabolic processes (TCA cycle, ETC and OxPhos) in various clusters of SCs. (C) Heatmap showing z-score of genes in the Gene Ontology term "fatty acid beta-oxidation" in various clusters of SCs. (D) UMAP-embedded scRNA-seq data showing expression levels of genes encoding rate-limiting proteins involved in mitochondrial long-chain FAO. (E) Relative mRNA levels of *Cpt2* in FACS-isolated QSC and ASC determined by qPCR. Error bars represent mean ± s.e.m. with $n = 3$ mice, each condition. *$P < 0.05$; two tailed, unpaired Student's $t$ test ($P = 0.0195$). (F) Cpt2 immunofluorescence in SCs grown on single myofibers cultured for 0 h, 24 h and 72 h. Scale bar: 20 μm. (G) Relative mRNA levels of *Cpt2* during differentiation of primary myoblasts. Error bars represent mean ± s.e.m. with $n = 3$ biological replicates, each condition. (H) Immunoblotting showing protein levels of Cpt2, myogenic factors and OxPhos proteins during differentiation of primary myoblasts. Source data are available online for this figure.

## Pharmacological inhibition of FAO disturbs mitochondrial respiration and SC function

To determine the bioenergetic role of mitochondrial FAO in SCs, we used Etomoxir, an irreversible inhibitor of CPT1, to block the entry of long-chain fatty acids (LCFAs) into mitochondria without affecting the oxidation of short-chain fatty acids (SCFAs) or peroxisomal FAO. Using the Seahorse Extracellular Flux Analyzer, we found that the oxygen consumption rate (OCR), an indicator of OxPhos activity, was significantly decreased at basal levels in Etomoxir-treated SC-derived primary myoblasts (Fig. EV3A,B), indicating that FAO contributes significantly to the overall oxygen

demands in myoblasts. In parallel, we measured OCR in response to oligomycin, FCCP, and rotenone and antimycin treatments, revealing bioenergetic capacities corresponding to ATP production, maximal mitochondrial activity, proton leak, and non-mitochondrial respiration, respectively. Notably, Etomoxir treatment significantly inhibited OCR for ATP production, proton leak, and non-mitochondrial respiration, but not the maximal respiratory capacity (Fig. EV3A,B). We also investigated the mitochondrial respiratory capacity in differentiating primary myoblasts. Compared to proliferating myoblasts, the differentiating myoblasts had higher OCR for basal respiration, ATP production and proton leak, which were all reduced by Etomoxir (Fig. EV3C,D). Thus,

mitochondrial FAO function as a major energy source during SC proliferation and differentiation.

We also examined the effect of several FAO inhibitors Etomoxir (CPT1 inhibitor), Oxfenicine (CPT1 inhibitor) and Perhexiline (CPT inhibitor) on proliferation and differentiation of primary myoblasts. Immunofluorescence of Ki67, a marker of proliferating cells, showed a significant reduction in the number of Ki67$^+$ myoblasts upon the treatment of Etomoxir (50 μM and 100 μM), Oxfenicine (10 mM), and Perhexiline (2.5 μM and 5 μM), respectively, corresponding to an overall reduction in cell number (Fig. EV4A–C). Furthermore, myoblast differentiation was suppressed by Etomoxir (50 μM and 100 μM), Oxfenicine (10 mM), and Perhexiline (5 μM) as indicated by reduced expression of MF20 after the treatments (Fig. EV5A–C). These data suggest that mitochondrial FAO is essential for the proliferation and differentiation of SCs.

## Cpt2 deletion does not affect maintenance of SCs but hampers their regenerative function

To explore the physiological function of mitochondrial FAO in SCs in vivo, we generated SC-specific Cpt2 KO mice by crossing Pax7$^{CreER}$ mice with Cpt2$^{flox/flox}$ mice in which exon 4 encoding the catalytic domain of Cpt2 was flanked by engineered LoxP sites (Lee et al, 2015). Genetic inactivation of Cpt2 was induced by five consecutive daily intraperitoneal (IP) injection of tamoxifen (TMX) in adult Pax7$^{CreER}$;Cpt2$^{flox/flox}$ (Cpt2$^{PKO}$) mice, with TMX-treated Cpt2$^{flox/flox}$ littermates as wild-type (WT) control (Fig. EV6A).

We first investigated the requirement of Cpt2 for maintenance of SCs by analyzing SC numbers on freshly isolated single myofibers, and found that there was no difference in the number of Pax7$^+$ SCs between WT and Cpt2$^{PKO}$ myofibers at 2 and 4 weeks after TMX administration (Fig. EV6B,C). Consistent results were observed by flow cytometry analysis, showing that the percentages of SCs (Sca1$^-$CD31$^-$CD45$^-$VCAM$^+$) in hindlimb muscles were comparable between WT and Cpt2$^{PKO}$ mice (Fig. EV6D,E). We also tested whether these Cpt2-null SCs still remained in a quiescent state. At 4 weeks after TMX administration, no MyoD (activation marker for SC) or Ki67 expression was detected in Pax7$^+$ SCs on myofibers freshly isolated from both WT and Cpt2$^{PKO}$ mice (Fig. EV6F,G), indicating that Cpt2-null SCs remain in a quiescent state in resting muscles. Thus, Cpt2 deletion does not affect the quiescent maintenance of SCs.

We then investigated how Cpt2 deletion affects the regenerative capacity of SCs. Following TMX-induction of Cpt2 deletion, TA muscles of Cpt2$^{PKO}$ and WT mice were injured by intramuscular injection of CTX and muscle repair was evaluated at 3.5, 5.5, 10 and 21 days post injury (dpi) to capture both early and late stage regeneration features (Fig. 2A). Notably, morphological and histological analyses showed that TA muscle regeneration in Cpt2$^{PKO}$ mice was defective, characterized by reduced muscle mass, a lower number of regenerated myofibers, smaller myofiber size, and increased fibrosis and mononuclear cell infiltration at 5.5 and 10 dpi, when compared to the WT muscles at the same time points (Fig. 2B–D). By immunostaining of embryonic Myosin Heavy Chain (eMyHC, a marker for newly regenerated myofibers) and dystrophin (a myofiber membrane protein) at 3.5 and 5.5 dpi, we detected numerous eMyHC$^+$ and dystrophin$^+$ myofibers in WT mice but very few in Cpt2$^{PKO}$ mice (Fig. 2E). By 10 dpi, most

regenerated (centronucleated) myofibers have lost eMyHC expression in WT mice, while nascent eMyHC$^+$ myofibers just became abundant in Cpt2$^{PKO}$ muscles (Fig. 2E), suggesting delayed regeneration in Cpt2$^{PKO}$ mice. Specifically, while 20.3% of the cross-sectional areas (CSA) were regenerated in WT mice by 5.5 dpi, only 4.9% areas were regenerated in Cpt2$^{PKO}$ mice (Fig. 2F). Densities (number per area) of eMyHC$^+$ myofibers in Cpt2$^{PKO}$ were 3.5-fold lower at 3.5 dpi (204 in WT versus 59 in Cpt2$^{PKO}$) but 3.1-fold higher at 10 dpi (90 in WT versus 279 in Cpt2$^{PKO}$) (Fig. 2G). Moreover, the number of regenerated dystrophin$^+$ myofibers in Cpt2$^{PKO}$ was reduced by 61.2% compared to WT mice at 5.5 dpi (Fig. 2H). Similarly, the CSA of regenerated myofibers was reduced by 36% and 52% in Cpt2$^{PKO}$ mice compared to WT mice at 5.5 and 10 dpi, respectively (Fig. 2I). Together, these results suggest that Cpt2 is required for timely muscle regeneration upon acute injury.

## Cpt2 deletion inhibits expansion and differentiation of SCs

The striking reduction of newly formed myofibers in Cpt2$^{PKO}$ mice at early regeneration stages prompted us to ask whether the defects were due to initial activation and expansion or subsequent differentiation of SCs. To address this, we first evaluated the number of Pax7$^+$ SCs in regenerating muscles at 3.5 dpi in vivo, when SCs were undergoing robust proliferation and expansion. As expected, Pax7$^+$ SCs were readily detected in WT muscles (Fig. 3A), however, they were less abundant in Cpt2$^{PKO}$ muscles (Fig. 3A). Quantitatively, the number of Pax7$^+$ SCs were reduced by 4.2-fold in Cpt2$^{PKO}$ compared to WT muscles (40.4 in WT versus 9.7 in Cpt2$^{PKO}$) (Fig. 3B). As severe fibrosis and inflammation may limit SCs expansion in the context of regeneration, we also evaluated the proliferation of SCs on single myofibers cultured ex vivo. While large clusters (≥ 4 SCs per cluster) were abundantly observed in WT myofibers after cultured for 3 days, the majority of SC clusters in Cpt2$^{PKO}$ were still doublets (Fig. 3C). The average SC number per cluster was ~38% fewer in Cpt2$^{PKO}$ compared to WT myofibers (Fig. 3D), suggesting that SC expansion was impaired in Cpt2$^{PKO}$ SCs. We performed TUNEL assay on cultured single myofiber and found that no significant increase of apoptotic SCs was observed in Cpt2$^{PKO}$ muscles, indicating that Cpt2$^{PKO}$ SCs were not undergoing apoptosis (Fig. 3E). To examine whether the impaired expansion was due to a proliferation defect, we performed 5-ethynyl-2′-deoxyuridine (EdU) incorporation assay. Mice were given EdU through drinking water one day before sacrifice at 3.5 dpi (Fig. 3F). While we were unable to detect any EdU$^-$ SCs in WT mice, EdU$^-$ SCs were readily identified in Cpt2$^{PKO}$ (Fig. 3G). Quantification of Pax7$^+$/EdU$^+$ SCs further highlighted the significant reduction of EdU$^+$ SCs in Cpt2$^{PKO}$ mice (Fig. 3H). We further sorted and cultured SCs from WT and Cpt2$^{PKO}$ mice and used Ki67 immunofluorescence to assess proliferation. Consistently, the total number of SCs and the percentage of Ki67$^+$ SCs were significantly lower in Cpt2$^{PKO}$ compared to WT (Fig. EV6H–I). These results indicate that loss of Cpt2 impairs proliferation and therefore expansion of SCs.

We then examined the differentiation capacity of Cpt2-null SCs. MyoG immunofluorescence in TA muscle cross-sections showed that the abundance of MyoG$^+$ cells was significantly decreased in Cpt2$^{PKO}$ mice at 3.5 dpi (Fig. 3I), which was 3-fold lower than WT mice (85.4 in WT versus 28.3 in Cpt2$^{PKO}$) (Fig. 3J). This result suggests that Cpt2 is required for myogenic cell commitment, consistent with endogenous Cpt2 expression being upregulated in

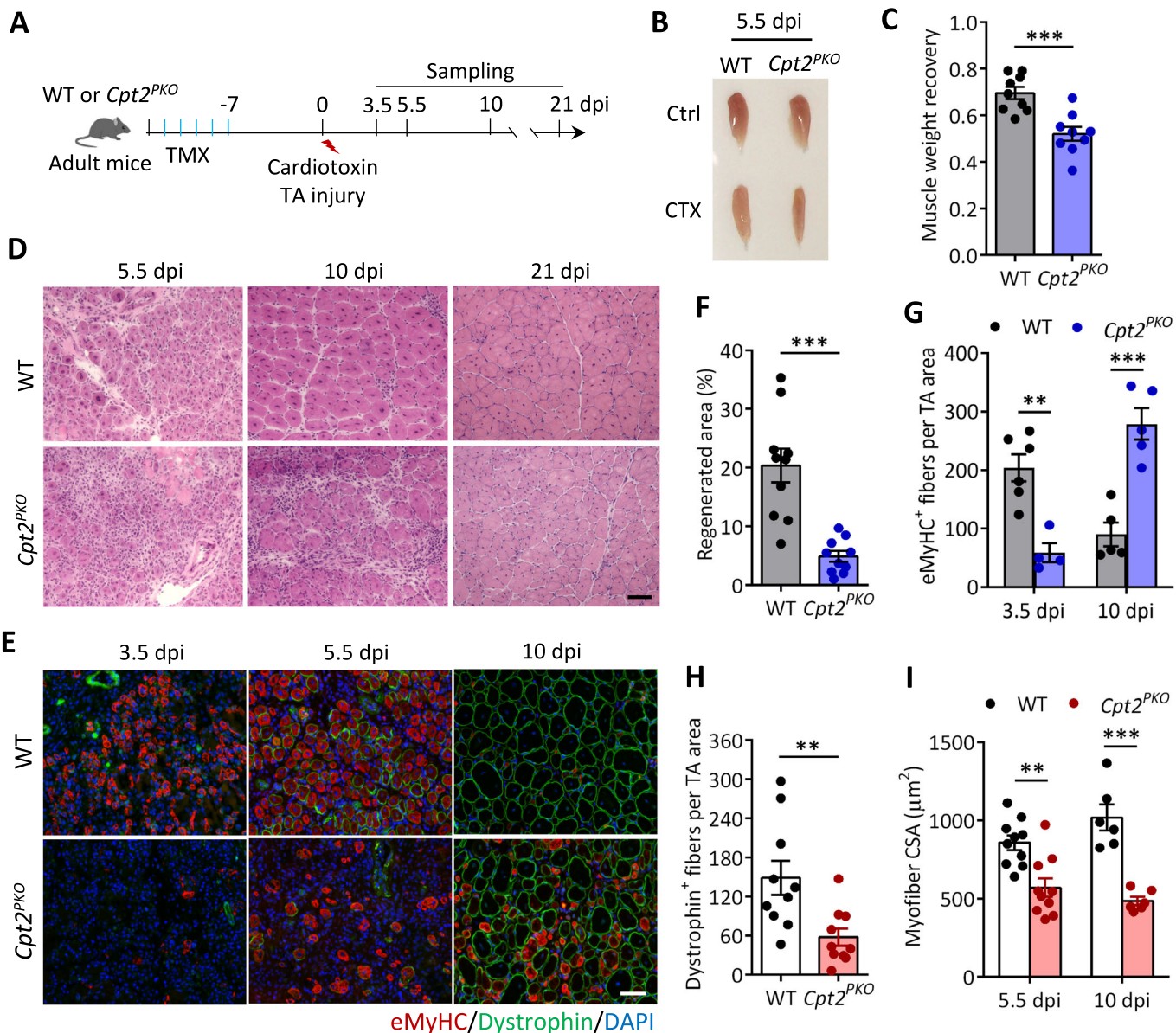

**Figure 2. Cpt2 deletion in satellite cells impairs muscle regeneration.**

(A) Schematic illustration of experimental design. Tamoxifen (TMX) injection was used to induce Cpt2 knockout in the Cpt2$^{PKO}$ mice. (B) Representative images of TA muscle from WT and Cpt2$^{PKO}$ mice at 5.5 dpi (day post injury). (C) Muscle weight recovery at 5.5 dpi, calculated as the ratio of injured to non-injured muscle weights. Error bars represent mean ± s.e.m. with $n = 9$ mice, each condition. ***$P < 0.001$; two-tailed, unpaired Student's $t$ test ($P = 0.00042$). (D) Representative H&E staining of TA muscle cross-sections from WT and Cpt2$^{PKO}$ mice at 5.5, 10 and 21 dpi. Scale bar, 50 μm. (E) Immunofluorescence of eMyHC (a marker of newly regenerated myofibers) and dystrophin (encircles all myofibers) in cross-sections of injured TA muscle at 3.5, 5.5 and 10 dpi. Scale bar, 50 μm. (F) Regenerated areas in TA muscle cross-sections at 5.5 dpi. Error bars represent mean ± s.e.m. with $n = 10$ mice, each condition. ***$P < 0.001$; two-tailed, unpaired Student's $t$ test ($P = 7.06 \times 10^{-5}$). (G) Numbers of eMyHC$^+$ cells per area of TA muscle cross-sections at 3.5 and 10 dpi. Error bars represent mean ± s.e.m. with $n = 6$ (WT) and $n = 4$ (Cpt2$^{PKO}$) mice at 3.5 dpi and $n = 5$ (WT and Cpt2$^{PKO}$) mice at 10 dpi. **$P < 0.01$, ***$P < 0.001$; two-tailed, unpaired Student's $t$ test (3.5 dpi: WT vs. Cpt2$^{PKO}$, $P = 0.00177$; 10 dpi: WT vs. Cpt2$^{PKO}$, $P = 0.0005$). (H) Numbers of newly regenerated myofibers per area at 5.5 dpi. Error bars represent mean ± s.e.m. with $n = 10$ mice, each condition. **$P < 0.01$; two-tailed, unpaired Student's $t$ test ($P = 0.00604$). (I) Average cross-sectional area (CSA) of regenerated myofibers at 5.5 and 10 dpi. Error bars represent mean ± s.e.m. with $n = 10$ mice at 3.5 dpi and $n = 6$ mice at 10 dpi. **$P < 0.01$, ***$P < 0.001$; two-tailed, unpaired Student's $t$ test (5.5 dpi: WT vs. Cpt2$^{PKO}$, $P = 0.001393$; 10 dpi: WT vs. Cpt2$^{PKO}$, $P = 0.000115$). Source data are available online for this figure.

differentiating SCs (Fig. 1). To further dissect the differentiation defects from proliferative defects, we isolated SCs from hind limb muscles of WT and Cpt2$^{PKO}$ mice using FACS and induced their differentiation under identical cell density. After 3 days in differentiation media, many large MF20$^+$ myotubes were identified in WT cultures, but only few small myotubes were observed in Cpt2$^{PKO}$ cultures, with most MF20$^+$ cells being mononuclear myocytes (Fig. 3K). Consistently, the differentiation index, calculated by the percentage of MF20$^+$ nuclei to total nuclei, was significantly lower (24.4%) in Cpt2$^{PKO}$ cultures, compared to 43.7%

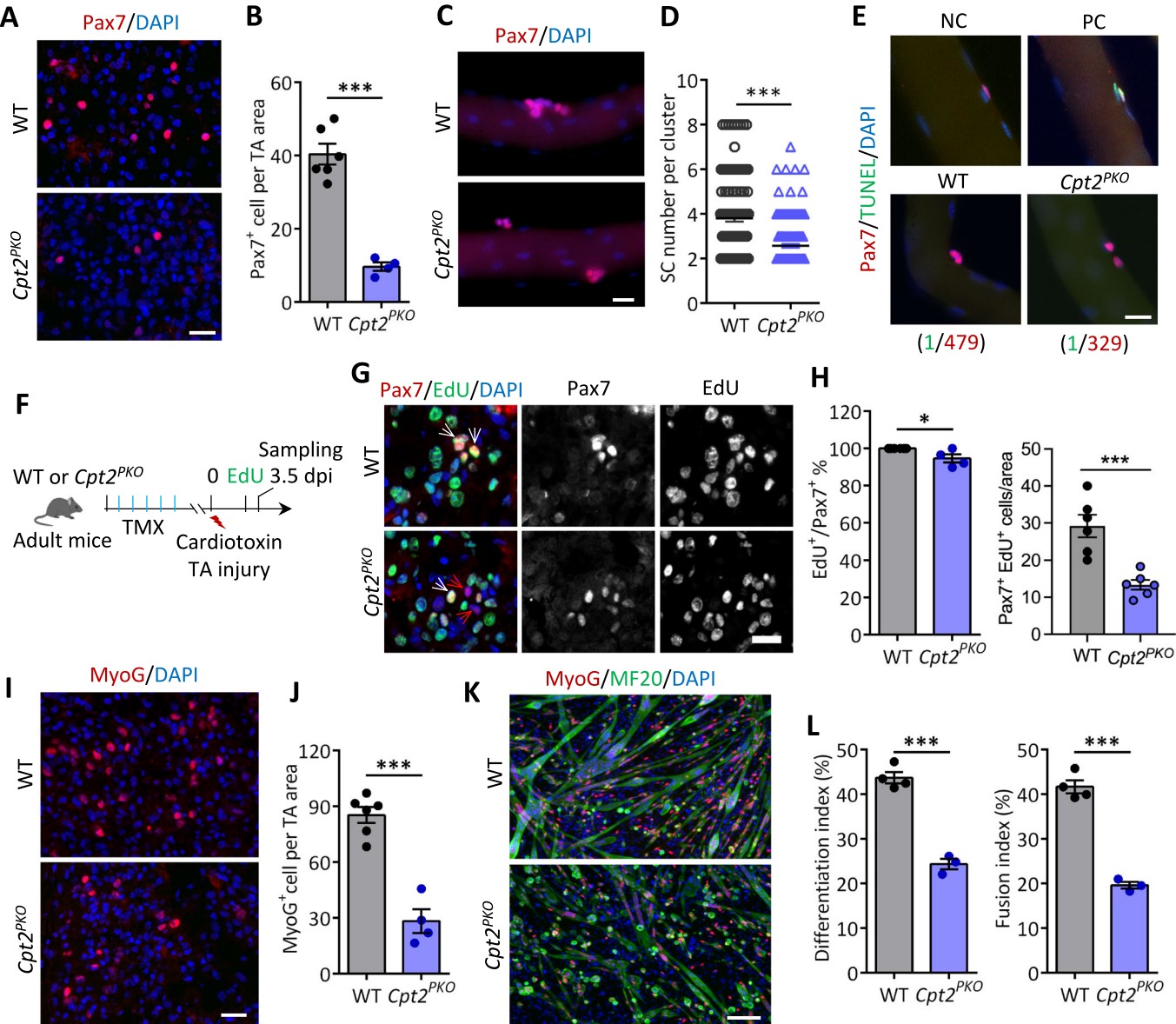

**Figure 3. Loss of *Cpt2* abrogates SC expansion and differentiation.**

(A, B) Pax7 immunofluorescence (A) and quantification of Pax7$^+$ cell numbers (B) in TA muscle cross-sections of WT and *Cpt2*$^{PKO}$ mice at 3.5 dpi. Error bars represent mean ± s.e.m. with n = 6 (WT) and n = 4 (*Cpt2*$^{PKO}$) mice. ***P < 0.001; two-tailed, unpaired Student's t test (P = 3.11 × 10$^{-5}$). Scale bar, 20 μm. (C, D) Representative immunofluorescence of Pax7 (C) and Pax7$^+$ cell numbers per cluster (D) on isolated WT and *Cpt2*$^{PKO}$ single myofibers cultured for 72 h. Error bars represent mean ± s.e.m. with n = 3 mice, each condition. ***P < 0.001; two tailed, unpaired Student's t test (P = 1.15 × 10$^{-12}$). Scale bar, 20 μm. (E) Representative immunofluorescence of Pax7 and TUNEL on isolated WT and *Cpt2*$^{PKO}$ single myofibers cultured for 40 h and quantification of Pax7$^+$TUNEL$^+$ cells (WT, n = 479; *Cpt2*$^{PKO}$, n = 329 from three mice for each group). NC negative control, PC positive control. Scale bar, 20 μm. (F) Schematics showing experimental design for in vivo EdU incorporation assay. (G) Immunofluorescence staining of Pax7 and EdU on TA muscle cross-sections at 3.5 dpi. White arrow indicates EdU$^+$/Pax7$^+$, red arrow indicates Pax7$^+$/EdU$^-$ cells. Scale bar, 20 μm. (H) Percentages of Pax7$^+$/EdU$^+$ cells and number of Pax7$^+$/EdU$^+$ cells per area. Error bars represent mean ± s.e.m. For EdU$^+$/Pax7$^+$ %, n = 6 (WT) and n = 4 (*Cpt2*$^{PKO}$) mice; For Pax7$^+$/EdU$^+$ cells per area, n = 6 (WT and *Cpt2*$^{PKO}$) mice. *P < 0.05, ***P < 0.001; two-tailed, unpaired Student's t test (EdU$^+$/Pax7$^+$ %, P = 0.01785; Pax7$^+$/EdU$^+$ cells per area, P = 0.00073). (I) Immunofluorescence staining of MyoG on TA muscle cross-sections at 3.5 dpi. Scale bar, 20 μm. (J) Quantification of MyoG$^+$ cells per TA area. Error bars represent mean ± s.e.m. with n = 6 (WT) and n = 4 (*Cpt2*$^{PKO}$) mice. ***P < 0.001; two-tailed, unpaired Student's t test (P = 5.80 × 10$^{-5}$). Scale bar, 20 μm. (K) Representative immunofluorescence staining MyoG and MF20 in WT and *Cpt2*$^{PKO}$ primary myoblasts differentiated for 2 days. Scale bar, 50 μm. (L) Quantification of differentiation and fusion index. Error bars represent mean ± s.e.m. with n = 4 (WT) and n = 3 (*Cpt2*$^{PKO}$) mice. ***P < 0.001; two-tailed, unpaired Student's t test (Differentiation index, P = 0.00012; Fusion index, P = 7.41 × 10$^{-5}$). Source data are available online for this figure.

in WT cultures (Fig. 3L). The fusion index calculated by the percentage of nuclei in myotubes to total nuclei was also significantly lower in $Cpt2^{PKO}$ than in WT cultures (Fig. 3L). Together, these results demonstrate that Cpt2 is indispensable for SC expansion and differentiation.

## *Cpt2* KO reduces FAO and diminishes mitochondrial respiration in SCs

The conventional function of Cpt2-mediated mitochondrial FAO is to oxidize LCFAs to generate acetyl-CoA, which subsequently enter into TCA cycle to provide energy substrates (Folmes et al, 2012; Ochocki et al, 2013). We hypothesized that the defective proliferation and differentiation of *Cpt2*-null SCs is due to deprivation of energy supply from FAO. To test this hypothesis, we first performed $^{13}$C-palmitic acid (PA) labelling assay to assess if FAO activity was affected by *Cpt2* deletion in SCs. SC-derived primary myoblasts from WT and $Cpt2^{PKO}$ mice were supplied with $^{13}$C-PA in culture for 24 h and cellular metabolites were extracted and subjected to GC/MS analysis (Fig. 4A). Each round of FAO should release two $^{13}$Carbon atoms from $^{13}$C-PA into acetyl-CoA, which will then be incorporated into sequential TCA intermediates that results in the addition of two ions to the molecular mass $(M + 2)$ (Fig. 4B). A significant proportion of $^{13}$C was detected in each TCA intermediate extracted from WT primary myoblasts, especially in citric acid, isocitric acid and oxaloacetic acid (Fig. 4C), indicating that primary myoblasts utilize fatty acid as an energy source. As negative controls, incorporation of $^{13}$C into lactic acid and pyruvate was rarely detected and indistinguishable in $Cpt2^{PKO}$ and WT myoblasts (Fig. EV7A). Strikingly, the incorporation of PA-derived $^{13}$C into all TCA intermediates was reduced in $Cpt2^{PKO}$ myoblasts compared WT myoblasts (Fig. 4C). Specifically, the relative level of $M + 2$ citric, isocitric, a-ketoglutaric, malic, succinic and oxaloacetic acid was reduced by 3.9-, 3.4-, 2.3-, 3.1-, 3.0- and 10.6-fold in $Cpt2^{PKO}$ compared to WT myoblasts (Fig. 4C), and the level of $M + 4$ citric, isocitric, malic and oxaloacetic acid was reduced by 4.9-, 6.4-, 3.6- and 36-fold in $Cpt2^{PKO}$ (Fig. 4C). These results demonstrate that FAO activity in SCs was blunted in the absence of Cpt2. We also examined the FAO activity by using the radioactive $^{14}$C-labeled PA. In consistent, the production of $^{14}$CO$_2$ was significantly decreased in $Cpt2^{PKO}$ myoblasts compared to WT (Fig. 4D). To determine if the reduced FAO stimulates a compensatory increase in glucose oxidation, we measured $^{14}$CO$_2$ production derived from $^{14}$C-labeled glucose. The results showed that the $^{14}$CO2 levels were comparable in $Cpt2^{PKO}$ and WT myoblasts, excluding a compensatory increase in glucose metabolism in the absence of Cpt2 (Fig. EV7B). Thus, *Cpt2* deletion impairs mitochondrial FAO activity in SCs.

To determine if the deficiency of mitochondrial FAO disrupts the subsequential energy metabolism of $Cpt2^{PKO}$ SCs, we assessed the mitochondrial respiratory capacity of SC-derived primary myoblasts from WT and $Cpt2^{PKO}$ mice. Under basal conditions, the OCR was lower in $Cpt2^{PKO}$ relative to WT primary myoblasts (Fig. 4E,F), indicating a reduction in overall oxygen demands for mitochondrial oxidation in the absence of Cpt2. The OCR for maximal respiratory capacity and ATP production were also significantly reduced in $Cpt2^{PKO}$ primary myoblasts when compared to WT myoblasts (Fig. 4E,F). In accordance with this, the total cellular ATP level was declined in $Cpt2^{PKO}$ primary myoblasts

(Fig. 4G). Taken together, these observations indicate that loss of *Cpt2* inhibits FAO flux and causes energy insufficiency in SCs.

## *Cpt2* deficiency reduces cellular acetyl-CoA level and protein acetylation

Increasing evidence suggests that beyond the bioenergetic role, nutrient-derived metabolites are essential mediators of cell signaling and epigenetic regulation (Madiraju et al, 2009; Mews et al, 2017; Moussaieff et al, 2015; Sperber et al, 2015; Sutendra et al, 2014; Tao Xu et al, 2017; Yucel et al, 2019; Zhou et al, 2019). We then sought to explore other potential mechanisms that might underlie the impaired SC expansion and commitment in $Cpt2^{PKO}$ mice. In addition to serving as an intermediate of cellular metabolism, acetyl-CoA is also a major acetyl donor for acetylation of proteins (Mews et al, 2017; Sutendra et al, 2014; Zhou et al, 2019), crucial for histone acetylation and gene regulation during embryonic stem cell development and differentiation (Moussaieff et al, 2015). We confirmed that *Cpt2* KO resulted in a dramatic 4.5-fold reduction of acetyl-CoA level in primary myoblasts (Fig. 5A). Given this observation, we postulated that *Cpt2* KO might disrupt protein acetylation, and influence gene regulation and eventually cause SC dysfunction. To test this hypothesis, we first examined the changes of global protein acetylation. Immunoblotting with anti-acetylated lysine antibody showed that while several bands exhibit similar acetylation levels between WT and $Cpt2^{PKO}$ primary myoblast, the intensities of some strongly acetylated bands were much lower in $Cpt2^{PKO}$ than in WT protein lysates from primary myoblasts (Fig. 5B), This observation suggests *Cpt2* KO affects acetylation of a selective subset of proteins in primary myoblast. Consistently, Coomassie blue gel staining of proteins pulled down with the anti-acetylated lysine antibody confirmed the reduction of acetylation of selective proteins in $Cpt2^{PKO}$ primary myoblasts (Fig. 5C). Thus, *Cpt2* KO depletes the cellular acetyl-CoA pool and disrupts protein acetylation in primary myoblasts.

To address if the change of protein acetylation is linked to dysfunction of *Cpt2*-null SCs, we examined acetylation of key myogenic protein Pax7, MyoD and MyoG, responsible for SC identity, commitment and differentiation, respectively. We performed immunoprecipitation with acetylated-lysine antibody followed by immunoblotting with Pax7, MyoD and MyoG antibodies (Fig. 5D,E). No differences in the level of acetylated MyoD and MyoG were observed between WT and $Cpt2^{PKO}$ primary myoblasts (Fig. 5D). In contrast, we detected much lower levels of acetylated-Pax7 in $Cpt2^{PKO}$ than in WT primary myoblasts, while the levels total Pax7 protein were comparable in WT and $Cpt2^{PKO}$ primary myoblasts (Fig. 5D). To confirm this finding, we performed reverse pull down with the Pax7 antibody. Consistently, a lower level of acetylated-Pax7 protein was detected in $Cpt2^{PKO}$ compared to WT primary myoblasts (Fig. 5E). Moreover, in situ proximity ligation assay (PLA) using anti-Pax7 and anti-acetylated lysine antibodies showed very strong puncta in the WT myoblasts (marked by Integrin-α7) grown on single myofibers cultured for 48 h (Fig. 5F), suggesting abundant acetylation of Pax7. In contrast, fewer and weaker PLA puncta were observed in $Cpt2^{PKO}$ SCs (Fig. 5F). These results together indicate that *Cpt2* KO abolishes Pax7 acetylation in SCs. To test whether the acetyl group in acetylated Pax7 is derived directly from FAO, we conducted 14C-PA labeling assay in primary myoblasts and performed protein pull

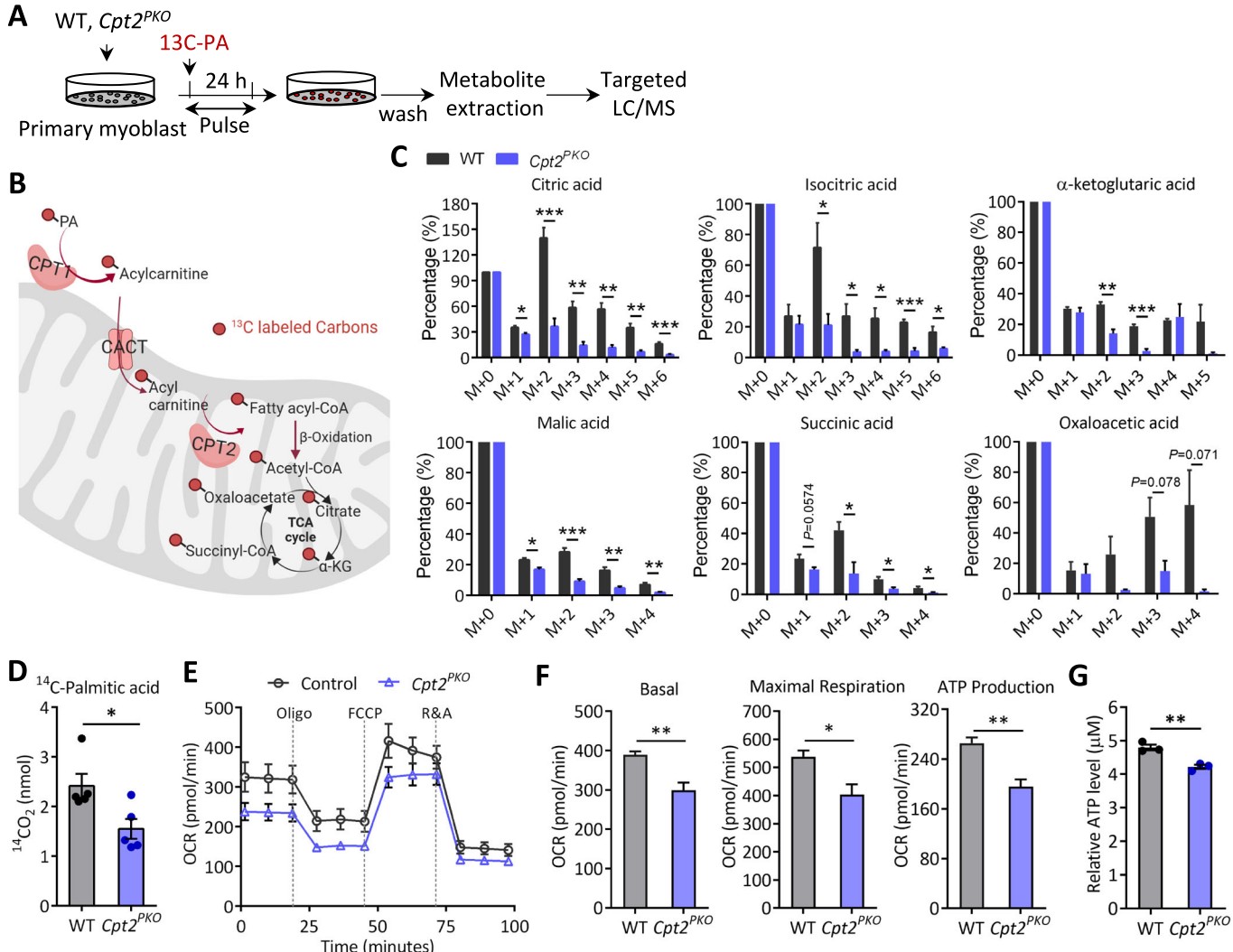

**Figure 4. Cpt2 KO inhibits FAO flux and causes energy insufficiency in satellite cells.**

(A) Experimental design for ¹³C-labeled palmitic acid (PA) incorporation and targeted metabolite analysis. Myoblasts were fed with ¹³C-labeled PA for 24 h, followed by metabolite extraction and LC/MS analysis. (B) Metabolic pathway of ¹³C-PA after it is transported into mitochondria and incorporated into the TCA cycle. (C) Targeted metabolite profiling showing the relative incorporation of ¹³C derived from ¹³C-PA in TCA cycle metabolites. M + 0 to M + 6 indicates the number of ¹³C atom incorporated in each metabolite, where the metabolite content of M + 0 (no ¹³C incorporation) is normalized to 100%. Error bars represent mean ± s.e.m. with $n = 4$ biological replicates, each condition. *$P < 0.05$, **$P < 0.01$, ***$P < 0.001$; two tailed, unpaired Student's $t$ test. Citric acid, $P = 0.019$ (M + 1), 0.00052 (M + 2), 0.0019 (M + 3), 0.0011 (M + 4), 0.0013 (M + 5), 0.00071 (M + 6); Isocitric acid, $P = 0.574$ (M + 1), 0.028 (M + 2), 0.0288 (M + 3), 0.0197 (M + 4), 0.00062 (M + 5), 0.0356 (M + 6); α-ketoglutaric acid, $P = 0.474$ (M + 1), 0.00104 (M + 2), 0.00053(M + 3), 0.85 (M + 4), 0.169 (M + 5); Malic acid, $P = 0.014$ (M + 1), 0.0008 (M + 2), 0.00267 (M + 3), 0.00686 (M + 4); Succinic acid, $P = 0.0574$ (M + 1), 0.0247 (M + 2), 0.0183 (M + 3), 0.0452 (M + 4); Oxaloacetic acid, $P = 0.793$ (M + 1), 0.153 (M + 2), 0.078 (M + 3), 0.071 (M + 4). (D) Radioactive FAO measurements using ¹⁴C-labeled PA revealed a significant decrease in ¹⁴CO₂ in Cpt2-null satellite cells compare to WT. Error bars represent mean ± s.e.m. with $n = 3$ mice, each condition. *$P < 0.05$; two-tailed, unpaired Student's $t$ test ($P = 0.0249$). (E) Representative seahorse curves showing oxygen consumption rate (OCR) of WT and Cpt2PKO primary myoblasts. (F) Quantification of the OCR related to basal and maximal respiration, and ATP production measured from the Seahorse assay. Error bars represent mean ± s.e.m. with $n = 5$ biological replicates. *$P < 0.05$, **$P < 0.05$; two-tailed, unpaired Student's $t$ test (basal OCR, $P = 0.00242$; maximal respiration OCR, $P = 0.01412$; ATP production OCR, $P = 0.00164$). (G) ATP production of cultured primary myoblasts isolated from WT and Cpt2PKO mice. Error bars represent mean ± s.e.m. with $n = 3$ mice, each condition. *$P < 0.05$; two-tailed, unpaired Student's $t$ test ($P = 0.00568$). Source data are available online for this figure.

down with Pax7 antibody. By measuring the radioactivity of agarose beads with pull downed protein, we found the significant enrichment of radioactivity in Pax7 pull down sample compared to the IgG control (Fig. 5G), suggesting the acetyl-CoA derived from PA contributed to the Pax7 acetylation.

To gain insight into how the reduced Pax7 acetylation affects Pax7 function in Ctp2-null SCs, we performed RNA sequencing on

SCs from WT and Cpt2PKO mice (Fig. 5H). This analysis identified 1250 genes whose expression is significantly increased or decreased in Cpt2PKO compared to WT myoblasts (Fig. 5H). We then questioned if a subset of these genes are regulated by Pax7, by intersecting the Cpt2 KO affected genes with the Pax7-reponsive genes from a publicly available dataset comparing gene expression of WT and Pax7-overexpression myoblasts (GSE77478) (Banerji

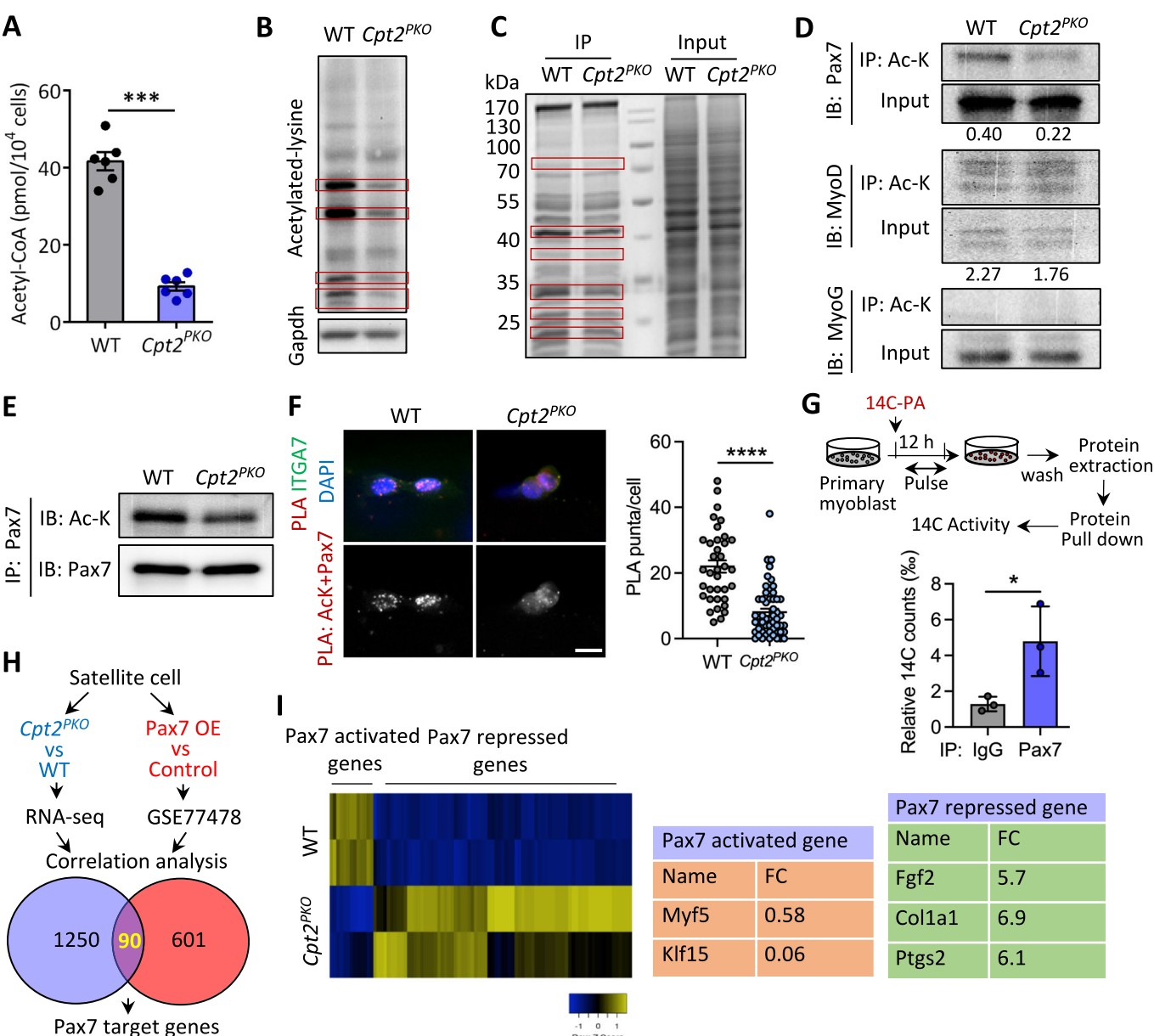

**Figure 5. Cpt2 deficiency reduces cellular acetyl-CoA and protein acetylation.**

(A) Cellular Acetyl-CoA level measured in cultured WT and *Cpt2^PKO* primary myoblasts. Error bars represent mean ± s.e.m. with $n = 6$ independent biological replicates. ***$P < 0.001$; two tailed, unpaired Student's $t$ test ($P = 2.2 \times 10^{-7}$). (B) An immunoblot image showing acetylated proteins revealed by an anti-acetylated lysine antibody in cultured WT and *Cpt2^PKO* primary myoblast (representative of $n = 3$ independent biological replicates). Red frame indicates bands with difference. (C) A Coomassie blue staining image of proteins pulled down by anti-acetylated lysine antibody in cultured WT and *Cpt2^PKO* primary myoblasts, as well as input controls ($n = 3$ independent biological replicates). Red frame indicates bands with difference. (D) Immunoprecipitation (IP) of lysates from WT and *Cpt2^PKO* primary myoblasts using an anti-acetylated lysine antibody, followed by immunoblotting with antibodies against Pax7, MyoD and MyoG ($n = 3$ independent biological replicates). (E) Immunoblot analysis of acetylated-Pax7 and total Pax7 in cultured WT and *Cpt2^PKO* primary myoblasts ($n = 3$ independent biological replicates). (F) In situ proximity ligation assay (PLA) with anti-Pax7 and anti-acetylated lysine antibodies showing decrease of PLA signaling in *Cpt2^PKO* satellite cells on single myofibers cultured for 48 h. Error bars represent mean ± s.e.m. with $n = 36$ SCs (WT) and $n = 53$ SCs (*Cpt2^PKO*). ****$P < 0.001$; two tailed, unpaired Student's $t$ test ($P = 6.1 \times 10^{-10}$). Scale bar, 10 μm. (G) Experimental design for $^{14}$C-labeled palmitic acid (PA) incorporation and protein pull down analysis for measuring $^{14}$C activity and the quantification of 14 C incorporation into Pax7 protein. Error bars represent mean ± s.e.m. with $n = 3$ independent biological replicates. *$P < 0.05$; two-tailed, unpaired Student's $t$ test ($P = 0.03768$). (H) Strategy for probing how *Cpt2* KO affects expression of Pax7-target genes. Among the 1250 genes differentially expressed between WT and *Cpt2^PKO* primary myoblasts, 90 were Pax7-target genes based on a published dataset (GSE77478). (I) Heatmap showing relative expression of Pax7-activated and repressed genes in WT and *Cpt2^PKO* primary myoblasts. FC indicates fold change. Source data are available online for this figure.

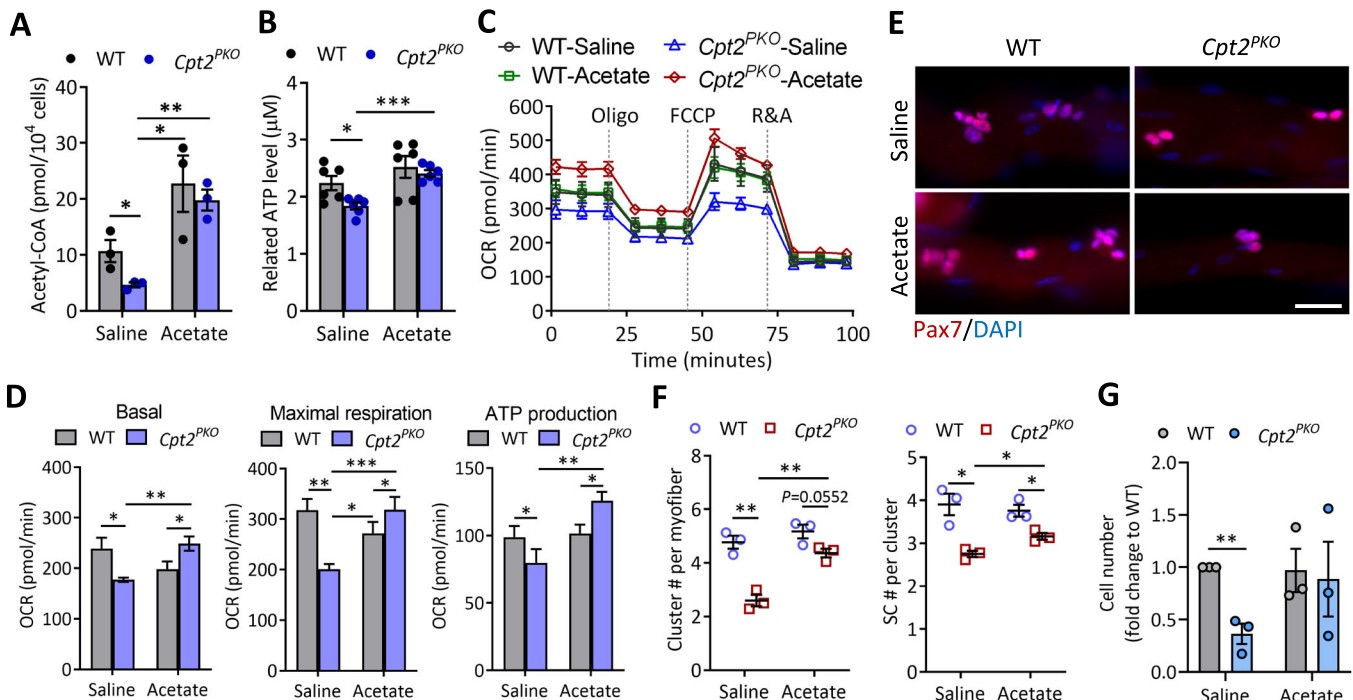

**Figure 6. Acetate repletion mitigates the metabolic defects of *Cpt2*-null satellite cells.**

(A) Cellular Acetyl-CoA level measured in WT and *Cpt2*$^{PKO}$ myoblasts. Primary myoblasts were cultured in low glucose medium with saline control or 0.5 mM sodium acetate for 24 h. Error bars represent mean ± s.e.m. with $n = 3$ biological replicates. *$P < 0.05$, **$P < 0.01$; two-tailed, unpaired Student's $t$ test (Saline:WT vs. Saline:*Cpt2*$^{PKO}$, $P = 0.0399$; Saline:*Cpt2*$^{PKO}$ vs. Acetate:WT, $P = 0.0233$; Saline:*Cpt2*$^{PKO}$ vs. Acetate:*Cpt2*$^{PKO}$, $P = 0.0014$). (B) Cellular ATP levels measured in WT and *Cpt2*$^{PKO}$ primary myoblasts. Error bars represent mean ± s.e.m. with $n = 6$ biological replicates. *$P < 0.05$, ***$P < 0.001$; two-tailed, unpaired Student's $t$ test (Saline:WT vs. Saline:*Cpt2*$^{PKO}$, $P = 0.0168$; Saline:*Cpt2*$^{PKO}$ vs. Acetate:*Cpt2*$^{PKO}$, $P = 5.4 \times 10^{-5}$). (C) Representative seahorse curves showing oxygen consumption rate (OCR) measured in WT and *Cpt2*$^{PKO}$ primary myoblasts. (D) Quantification of the OCR associated with basal and maximal respiration, and ATP production. Error bars represent mean ± s.e.m. with $n = 4$ or 5 biological replicates. *$P < 0.05$, **$P < 0.01$, ***$P < 0.001$; two tailed, unpaired Student's $t$ test. Basal OCR, Saline:WT vs. Saline:*Cpt2*$^{PKO}$, $P = 0.0292$; Acetate:WT vs. Acetate:*Cpt2*$^{PKO}$, $P = 0.0389$; Saline:*Cpt2*$^{PKO}$ vs. Acetate:*Cpt2*$^{PKO}$, $P = 0.0032$. Maximal respiration OCR, Saline:WT vs. Saline:*Cpt2*$^{PKO}$, $P = 0.0046$; Acetate:WT vs. Acetate:*Cpt2*$^{PKO}$, $P = 0.048$, Acetate:WT vs. Saline:*Cpt2*$^{PKO}$, $P = 0.049$; Saline:*Cpt2*$^{PKO}$ vs. Acetate:*Cpt2*$^{PKO}$, $P = 0.0006$. ATP production OCR, Saline:WT vs. Saline:*Cpt2*$^{PKO}$, $P = 0.0124$; Acetate:WT vs. Acetate:*Cpt2*$^{PKO}$, $P = 0.0316$, Saline:*Cpt2*$^{PKO}$ vs. Acetate:*Cpt2*$^{PKO}$, $P = 0.0024$. (E) Representative immunofluorescence staining of Pax7 on individual WT and *Cpt2*$^{PKO}$ myofibers cultured with 0.5 mM sodium acetate or saline control for 60 h. Scale bar, 50 μm. (F) Quantification of the number of cell clusters (containing 2 or more Pax7$^+$ cells) per myofiber and the Pax7$^+$ cell number per cluster. Error bars represent mean ± s.e.m. with $n = 3$ mice. *$P < 0.05$, **$P < 0.01$; two-tailed, unpaired Student's $t$ test. Cluster per myofiber, Saline:WT vs. Saline:*Cpt2*$^{PKO}$, $P = 0.0025$; Acetate:WT vs. Acetate:*Cpt2*$^{PKO}$, $P = 0.0552$, Saline:*Cpt2*$^{PKO}$ vs. Acetate:*Cpt2*$^{PKO}$, $P = 0.0026$. SC per cluster, Saline:WT vs. Saline:*Cpt2*$^{PKO}$, $P = 0.0114$; Acetate:WT vs. Acetate:*Cpt2*$^{PKO}$, $P = 0.02$, Saline:*Cpt2*$^{PKO}$ vs. Acetate:*Cpt2*$^{PKO}$, $P = 0.0193$. (G) Quantification of the number of FACS-sorted SCs after cultured with 0.5 mM sodium acetate or saline control for 72 h. Error bars represent mean ± s.e.m. with $n = 3$ mice. **$P < 0.01$; two tailed, unpaired Student's $t$ test (Saline:WT vs. Saline:*Cpt2*$^{PKO}$, $P = 0.0028$). Source data are available online for this figure.

et al, 2017). This correlation analysis yielded 90 overlapping genes from a total of 601 Pax7-responsive genes (Fig. 5H). Of these 90 genes, 77 Pax7-repressed genes were upregulated by *Cpt2* KO, including *Col1a1*, *Fgf2* and *Ptgs2* (Fig. 5I), all are essential regulators of SC differentiation. The remaining 13 genes proposed to be Pax7-activated targets were downregulated in *Cpt2*$^{PKO}$ SCs, including the known Pax7 target gene *Myf5* and *Klf15* (Fig. 5I). Given that lack of Pax7 acetylation affects the expression of its target genes (Sincennes et al, 2021), our unbiased gene expression analyses suggest that the reduced acetylation of Pax7 in *Cpt2* KO myoblasts might affect the normal transcriptional activity of Pax7.

## Acetate repletion restores the energetic flux and proliferation of *Cpt2*-null SCs

Acetate, a SCFA mainly acquired from diet, can be converted to acetyl-CoA by acetyl-CoA synthetase (ACS) and contributes to energy metabolism and protein acetylation in mammalian cells

(Bose et al, 2019; Gao et al, 2016; Qiu et al, 2019). Given the reduced cellular acetyl-CoA level in *Cpt2*-null SCs, we hypothesized that acetate supplementation should restore acetyl-CoA levels and cellular metabolism in *Cpt2*-null SCs. To test the hypothesis, we first treated *Cpt2*-null primary myoblasts with sodium acetate and examined the cellular acetyl-CoA level. Acetate supplementation significantly elevated the acetyl-CoA level in WT myoblasts, with a 2.1-fold increase over the saline-treated control (Fig. 6A). As expected, acetate supplement restored acetyl-CoA in *Cpt2*-null myoblasts, to a level similar to that in WT myoblasts supplemented with acetate (Fig. 6A). The restoration of cellular acetyl-CoA is accompanied by restoration of cellular ATP level in the *Cpt2*-null SCs, to a level identical to that in WT myoblasts supplemented with acetate (Fig. 6B). Interestingly, although acetate treatment did not affect mitochondrial OCRs in WT myoblasts, it significantly increased the basal and maximal OCRs and OCR for ATP production in *Cpt2*-null myoblasts to levels even higher in the *Cpt2*-null than in WT myoblasts when both were supplemented

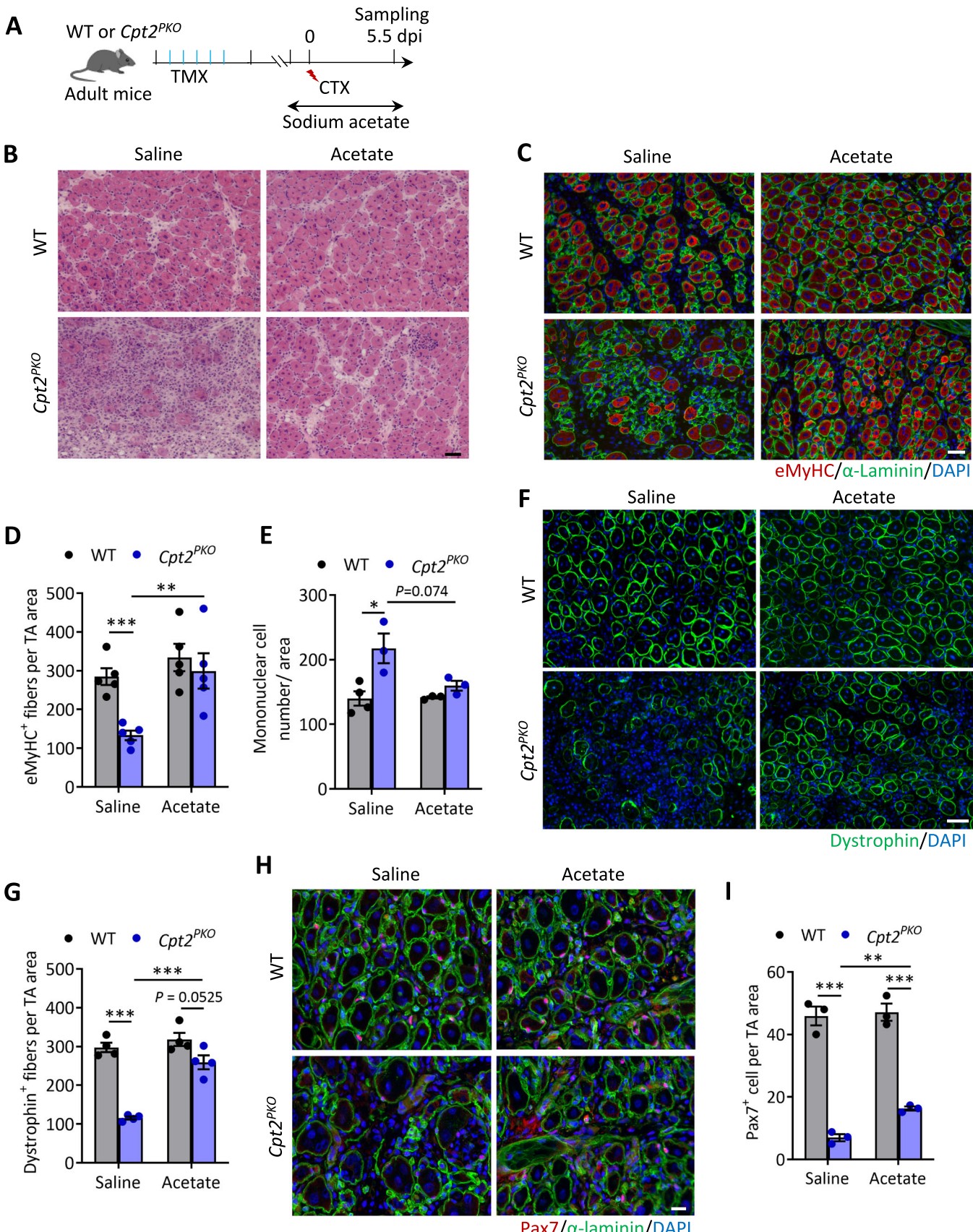

**Figure 7. Acetate supplementation partially rescues the regenerative defects of *Cpt2*-null satellite cells in vivo.**

(A) Experimental design for sodium acetate supplementation (via drinking water at 0.2 M dosage) during muscle regeneration in vivo. (B) Representative H&E staining of TA muscle cross-sections from WT and $Cpt2^{PKO}$ mice at 5.5 dpi. Scale bar, 50 μm. (C and D) Immunofluorescence of eMyHC (C) and quantification of eMyHC$^+$ cell number per area (D) on injured TA muscle cross-sections at 5.5 dpi. Error bars represent mean ± s.e.m. with $n = 5$ mice. **$P < 0.01$, ***$P < 0.001$; two-tailed, unpaired Student's $t$ test. Saline:WT vs. Saline:$Cpt2^{PKO}$, $P = 0.00028$; Saline:$Cpt2^{PKO}$ vs. Acetate:$Cpt2^{PKO}$, $P = 0.00797$. Scale bar, 50 μm. (E) Quantification of infiltrated mononuclear cells in (C). Error bars represent mean ± s.e.m. with $n = 3$ or 4 mice. *$P < 0.05$; two-tailed, unpaired Student's $t$ test. Saline:WT vs. Saline:$Cpt2^{PKO}$, $P = 0.0207$; Saline:$Cpt2^{PKO}$ vs. Acetate:$Cpt2^{PKO}$, $P = 0.074$. (F, G) Immunofluorescence of dystrophin (F) and quantification of dystrophin$^+$ myofiber number per area (G) on injured TA muscle cross-sections at 5.5 dpi. Error bars represent mean ± s.e.m. with $n = 4$ mice. *$P < 0.05$, ***$P < 0.001$; two-tailed, unpaired Student's $t$ test. Saline:WT vs. Saline:$Cpt2^{PKO}$, $P = 7.4 \times 10^{-6}$; Acetate:WT vs. Acetate:$Cpt2^{PKO}$, $P = 0.0525$; Saline:$Cpt2^{PKO}$ vs. Acetate:$Cpt2^{PKO}$, $P = 0.00023$. Scale bar, 50 μm. (H) Immunofluorescence staining of Pax7 on TA muscle cross-sections of WT and $Cpt2^{PKO}$ mice at 5.5 dpi. (I) Quantification of Pax7$^+$ cell number on TA muscle cross-sections shown in (H). Error bars represent mean ± s.e.m. with $n = 3$ mice. **$P < 0.01$, ***$P < 0.001$; two-tailed, unpaired Student's $t$ test. Saline:WT vs. Saline:$Cpt2^{PKO}$, $P = 0.00026$; Acetate:WT vs. Acetate:$Cpt2^{PKO}$, $P = 0.00039$; Saline:$Cpt2^{PKO}$ vs. Acetate:$Cpt2^{PKO}$, $P = 0.00184$. Scale bar, 20 μm. Source data are available online for this figure.

with acetate (Fig. 6D). These seahorse OCR results were reproduced in a separated independent investigation (data not included). Thus, acetate repletion effectively restores the cellular acetyl-CoA pool and energetic flux in *Cpt2*-null SCs.

We next sought to determine if acetate supplementation restores the proliferative defect of $Cpt2^{PKO}$ SCs. We first examined the proliferation of SCs on myofiber cultured with acetate ex vivo (Fig. 6E). In saline-treated control, the number of SC cluster was significantly lower in $Cpt2^{PKO}$ myofibers compared to WT, whereas acetate treatment markedly increased the number of SC clusters in $Cpt2^{PKO}$ myofibers (Fig. 6F), to a level comparable to the WT. The number of SC per cluster was also significantly increased in acetate-treated $Cpt2^{PKO}$ myofibers, though the level remained lower than that of WT cells (Fig. 6F). We further performed acetate rescue assay using FACS-sorted SCs. Consistently, acetate supplementation significantly restored the proliferation capacity of $Cpt2^{PKO}$ SCs (Fig. 6G). These results indicate that acetate supplementation partially rescues the proliferative defects of $Cpt2^{PKO}$ SCs.

## Acetate supplementation rescues the regenerative defect of *Cpt2*-null SCs

We next evaluated whether the regenerative defect of $Cpt2^{PKO}$ SCs could be rescued by acetate supplementation in vivo. We administrated a dose of acetate via IP injection one day prior to muscle injury in WT and $Cpt2^{PKO}$ mice, and continuously administrated acetate to the mice via drinking water and IP injection during the experimental period (Fig. 7A). Interestingly, acetate treatment had no significant influence on the regeneration of WT muscles, as indicated by similar muscle histology and number of eMyHC$^+$ and dystrophin$^+$ myofibers in saline and acetate treatment groups (Fig. 7B–G). Strikingly, while muscle regeneration in $Cpt2^{PKO}$ mice was significantly worse compared to WT mice in the saline-treated group, acetate supplement normalized muscle regeneration in the $Cpt2^{PKO}$ mice at 5.5 dpi, reflected by identical abundance of central nucleated regenerated myofibers, absence of fibrosis and comparable mononuclear cell infiltration in $Cpt2^{PKO}$ and WT mice (Fig. 7D,E). In addition, immunofluorescence staining of eMyHC and dystrophin revealed that the numbers of newly formed (eMyHC$^+$) and regenerated (dystrophin$^+$) myofibers were both dramatically increased in acetate-treated $Cpt2^{PKO}$ mice, to levels compared to those of WT mice (Fig. 7C–G). Moreover, acetate supplementation significantly repopulated the SC pool upon muscle injury, though the SC number was still lower than that in WT muscles (Fig. 7H,I). Taken together, these results

indicate that acetate supplementation restores the regenerative capacity of *Cpt2*-null SCs in vivo.

## Discussion

The fate transition of adult stem cells is accompanied by distinct metabolic requirements. Whether these metabolic changes determine the stem cell fate transitions and function remains poorly understood. Our present study uncovers a critical metabolic requirement on mitochondrial fatty acid utilization for skeletal muscle stem cell regenerative capacity. We show that silencing Cpt2-mediated mitochondrial FAO in SCs depletes the cellular acetyl-CoA pool and suppresses energy metabolic flux and protein acetylation, leading to defective cell proliferation and muscle regeneration. Our observations provide the first evidence that mitochondrial FAO links energy metabolism and protein acetylation to regulate stem cell function.

Accumulating efforts have been taken to characterize the metabolic demands of SCs during fate transitions (Dell'Orso et al, 2019; Pala et al, 2018; Ryall et al, 2015b, Wüst et al, 2018). SCs in non-injured muscles exist in a quiescent cell state with low metabolic demand and undergo massive mitochondrial biogenesis to meet the increasing energy demands upon proliferation and differentiation (Pala et al, 2018). A previous study reported that quiescent SCs exhibit high FAO and switch from oxidative to glycolytic metabolism upon activation and proliferation (Ryall et al, 2015b). Another study found that glucose metabolism is dispensable for mitochondrial respiration in proliferating SCs, but important for quiescent and differentiating SCs (Yucel et al, 2019). Additionally, analysis of SC transcriptomic datasets suggested that in response to injury-induced activation and proliferation, SCs exhibit a general upregulation for main energy production pathways including glycolysis, fatty acid metabolism and OxPhos at 3 dpi (Pala et al, 2018). However, by 5 dpi, SCs reduce glycolytic activity when undergoing self-renewal divisions and differentiation (Pala et al, 2018). These contradictory observations may be due to the high heterogeneity of SCs and the methodological limitation of bulk SC analysis. Our analysis of scRNA-seq data obtained from SCs at different regenerating stages provides a transcriptional snapshot of how expression of metabolic genes changes during SC fate transitions. Notably, our observations indicate that mitochondrial FAO is the major metabolic pathways couples with the vast energy demands in proliferating and differentiating SCs. We did not observe significant gene

enrichments in quiescent and self-renewed SCs for specific metabolic pathways. However, our scRNA-seq data suggest aerobic instead of anaerobic glycolysis is also enriched in the differentiating SCs. This observation is consistent with previous studies demonstrating that PDH activity controls myogenic differentiation by promoting mitochondrial energy metabolism (Yucel et al, 2019). These results suggest that a tradeoff mechanism may exist in different SC states for FA- or glucose-derived metabolites utilized for energy or other biosynthetic requirements.

It is well-known that FAO is critical for tissues with high-energy demands, such as skeletal and cardiac muscles, but direct evidence demonstrating the role of FAO in muscle stem cell biology has been lacking. Through systemic inhibition of FAO using pharmacological inhibitors, a recent study reported that peroxisomal but not mitochondrial FAO regulates myogenic cell differentiation and muscle regeneration (Pala et al, 2018). However, the cell-specific effect of FAO in SCs in vivo is unclear due to systemic effects of the pharmacological inhibitors used. Using conditional KO approaches for SC-specific depletion of Cpt2, our study identified a critical role of mitochondrial FAO in SC expansion and differentiation. In the absence of Cpt2-dependent mitochondrial FAO, SCs fail to expand in response to injury, which in turn impairs muscle regeneration. In support of our finding, genetic deletion of *Cpt1a* diminishes the proliferative capacity of intestinal stem cells and results in the defective intestinal regeneration (Mihaylova et al, 2018). Previous studies in HSCs and NPSCs suggest that FAO is essential for the maintenance of HSCs and the quiescent state of NPSCs (Ito et al, 2012; Knobloch et al, 2017). In contrast, we showed that mitochondrial FAO is dispensable for the quiescent maintenance of SCs, consistent with the low expression level of FAO-related genes and lack of Cpt2 protein expression in quiescent SCs. As we only evaluated SC maintenance within 4 weeks after *Cpt2* deletion, the long-term effects of *Cpt2* KO on the maintenance of SCs has yet to be determined. Nonetheless, our observations in conjunction with the existing literature highlight a critical and stage-specific role of FAO in adult muscle stem cell maintenance and function.

Although several studies have proposed that FAO is required for adult stem cell function (Ito et al, 2012; Knobloch et al, 2017; Mihaylova et al, 2018), the mechanisms underlying the distinct role of FAO in quiescence, self-renewal or expansion remain largely unknown. In response to injury, SCs exit from quiescence and then proliferate, differentiate, and fuse to form multinucleated myotubes. Our observations indicate that these myogenic stages are associated with sequentially increasing energy demands. Accordingly, FAO is crucial for proliferation and differentiation of SCs, as pharmaceutical and genetic inhibition of Cpt2-dependent FAO significantly reduces SC proliferation and differentiation both in vivo and in vitro. Our results demonstrating that supplementation of SCFA acetate restored the acetyl-CoA pool and metabolic flux in *Cpt2*-deficient SCs, and partially rescues their regenerative defects in vivo suggest a causal role of deficient acetyl-CoA level in SC dysfunction in the *Cpt2* KO mice. In humans, CPT II deficiency causes several autosomal recessive disorders: a myopathic form with adult onset (type 1) characterized by skeletal muscle weakness and myoglobinuria; a severe life-threatening infantile form (type 2) with hypoketotic hypoglycemia and cardiomyopathy; and a fatal neonatal form with organ abnormalities (type 3) (Corti et al, 2008). Our results suggest that nutritional compensation with short chain fatty acid such as acetate may ameliorate metabolic disorders caused by CPT2 deficiency.

Emerging evidence suggests that metabolites derived from cellular metabolism are essential for stem cell function through epigenetic modifications in mammals (Folmes et al, 2012; Somasundaram et al, 2020). Central metabolites, such as acetyl-CoA, are the acetyl group donor for enzymes that catalyze the deposition of covalent modifications on histones, DNA, and RNA (Sabari et al, 2017). As a main metabolic pathway to breakdown long chain FA, mitochondrial FAO generates several essential metabolites including acetyl-CoA. It has been reported that glycolysis-mediated changes in acetyl-CoA and histone acetylation control the differentiation of embryonic and adult stem cells (Moussaieff et al, 2015; Yucel et al, 2019; Zhou et al, 2019). Our data suggest that beyond providing energy substrates, mitochondrial FAO is involved in the regulation of SC function by controlling the homeostasis of intracellular acetyl-CoA and protein acetylation. In support of this, a recent study demonstrated that fatty acid-derived acetyl-CoA is a major carbon source for histone and mitochondrial protein acetylation (McDonnell et al, 2016; Pougovkina et al, 2014). Thus, the effect of mitochondrial FAO on SC activation and proliferation could be a result of combined defects in energy production and protein acetylation, or even through other yet to be defined mechanisms mediated by bioactive lipids and metabolites. Although acetate supplementation rescued the effect of *Cpt2* KO on SCs, whether the rescue is mainly mediated through energetic, acetylation or other pathways remained to be dissected. Nevertheless, we provide the first line of evidence that Cpt2 regulates intracellular acetyl-CoA levels and acetylation of non-histone transcription factor such as Pax7, a key regulator of muscle stem cell function.

An intriguing observation in our study is that loss of *Cpt2* in SCs reduces protein acetylation in a selective manner, but not globally. Previous studies have demonstrated that acetylation of myogenic regulatory factor MyoD is essential for SC commitment and differentiation (Duquet et al, 2006; Sartorelli et al, 1999). Interestingly, we did not observe a difference in levels of acetylated MyoD and MyoG after *Cpt2* KO. Notably, our experiments demonstrated that Cpt2 affect acetylation of Pax7, a key transcriptional factor for SC proliferation and maintenance. In addition, the acetylation of Pax7 directly affects its transcriptional activity as shown by alterations in the expression Pax7-target genes in *Cpt2*-null SCs. In consistent, a recent study shows that acetylation of Pax7 controls muscle stem cell self-renewal and differentiation potential in mice. Loss of Pax7 acetylation reduced SC numbers following repeated acute injury in young mice or single acute injury in aged mice (Sincennes et al, 2021). In consistent with our observations of SC phenotypes, these findings support the notion that reduced Pax7 acetylation in $Cpt2^{PKO}$ SCs might be one of the mechanisms causing SC defect, in parallel with the energetic defect. However, our data could not differentiate the contribution between Pax7-dependent effects and effects that result from altered energetics in determining the dysfunction of $Cpt2^{PKO}$ SCs. It is possible that the transcriptional regulatory function of myogenic factors differs in various cellular status (e.g. quiescent, activated, proliferating SCs), and acetylation of Pax7 and other myogenic factors acts as a regulatory mechanism that determines the stage-specific functions. It would be interesting to define in future studies how the selective acetylation of Pax7 and other factors is regulated by metabolic pathways. Additionally, as we only examined lysine acetylation in this study, the potential role of Cpt2-controled acetyl-CoA production in N-terminal acetylation of proteins has yet to be investigated. Future work identifying the identities of

proteins whose acetylation is affected by Cpt2 in SCs will shed light on the precise mechanism underlying the selective protein acetylation mediate by mitochondrial FAO.

# Methods

## Reagents and tools table

| Reagent/resource | Reference or source | Identifier or catalog number |
|---|---|---|
| **Experimental models** | | |
| Mouse: B6;129-*Pax7*tm2.1(cre/ERT2)Fan/J | The Jackson Laboratory | JAX stock: #012476 |
| Mouse: B6.Cg-*Pax7*tm1(cre/ERT2)Gaka/J | The Jackson Laboratory | JAX stock: #017763 |
| Mouse: B6;129-*Gt(ROSA)26Sor*tm5(CAG-Sun1/sfGFP)Nat/J | The Jackson Laboratory | JAX stock: #021039 |
| Mouse: B6-*Cpt2*flox/flox | Produced by Dr. Michael J. Wolfgang lab at Johns Hopkins University School of Medicine | N/A |
| Single myofiber ex vivo culture | This study | N/A |
| Satellite cell and primary myoblast in vitro culture | This study | N/A |
| **Antibodies** | | |
| Mouse monoclonal anti-PAX7 | DSHB | Cat# PAX7, RRID:AB_2299243 |
| Mouse monoclonal anti-MyoD | Santa Cruz Biotechnology | Cat# sc-377460, RRID:AB_2813894 |
| Mouse monoclonal anti-MyoG | DSHB | Cat# F5D, RRID:AB_2146602 |
| Mouse monoclonal anti-MyHC | DSHB | Cat# MF 20, RRID:AB_2147781 |
| Mouse monoclonal anti-eMyHC | DSHB | Cat# F1.652, RRID:AB_528358 |
| Rabbit polyclonal anti-Laminin | Sigma-Aldrich | Cat# L9393, RRID:AB_477163 |
| Rabbit polyclonal anti-Dystrophin | Abcam | Cat# ab15277, RRID:AB_301813 |
| Rabbit polyclonal anti-Ki67 | Abcam | Cat# ab15580, RRID:AB_443209 |
| Rabbit polyclonal anti-Cpt2 | MilliporeSigma | Cat# ABS85 |
| Mouse monoclonal anti-GAPDH | Santa Cruz Biotechnology | Cat# sc-32233, RRID:AB_627679 |
| Rabbit polyclonal anti-Acetylated-Lysine | Cell Signaling Technology | Cat# 9441 |
| PE Rat anti-mouse CD31 antibody | BD Biosciences | Cat# 553373, RRID:AB_394819 |
| PE anti-mouse CD45 antibody | eBioscience | Cat# 12-0451-82, RRID:AB_465668 |
| Pacific Blue anti-mouse Ly-6A/E (Sca-1) antibody | BioLegend | Cat# 122520, RRID:AB_2143237 |
| APC anti-mouse CD106 antibody | BioLegend | Cat# 105718, RRID:AB_1877141 |
| Alexa 568 goat anti-mouse IgG1 | Invitrogen | Cat# A-21124, RRID:AB_2535766 |
| Alexa 488 goat anti-mouse IgG1 | Invitrogen | Cat# A-21121, RRID:AB_2535764 |
| Alexa 647 goat anti-mouse IgG2b | Invitrogen | Cat# A-21242, RRID:AB_2535811 |
| Alexa 488 goat anti-rabbit IgG | Invitrogen | Cat# A-11034, RRID:AB_2576217 |
| Alexa 647 goat anti-rabbit IgG | Invitrogen | Cat# A-21244, RRID:AB_2535812 |
| Normal mouse IgG | Santa Cruz Biotechnology | Cat# sc-2025 |

| Reagent/resource | Reference or source | Identifier or catalog number |
|---|---|---|
| Normal rabbit IgG | Cell Signaling Technology | Cat# 2729 |
| HRP AffiniPure goat anti-mouse IgG | Jackson ImmunoResearch | Cat# 115-035-003, RRID:AB_10015289 |
| HRP AffiniPure goat anti-rabbit IgG | Jackson ImmunoResearch | Cat# 111-035-003, RRID:AB_2313567 |
| **Oligonucleotides and other sequence-based reagents** | | |
| *Cpt2* forward primer: CAACTCGTATACCCAAACCCAGTC | This study | N/A |
| *Cpt2* reverse primer: GTTCCCATCTTGATCGAGGACATC | This study | N/A |
| *18 s* rRNA forward primer: AGTCCCTGCCCTTTGTACACA | This study | N/A |
| *18 s* rRNA reverse primer: CGATCCGAGGGCCTCACTA | This study | N/A |
| **Chemicals, enzymes, and other reagents** | | |
| Tamoxifen (TMX) | Calbiochem | Cat# 579000 |
| 4-Hydroxytamoxifen (4-OHT) | Sigma-Aldrich | Cat# H6278 |
| Sodium acetate | Sigma-Aldrich | Cat# 241245 |
| Cardiotoxin | Sigma-Aldrich | Cat# 217503 |
| 5-ethynyl-2-deoxyuridine (EdU) | Carbosynth | Cat# NE08701 |
| Tetramethylrhodamine (TAMRA) Azide | Invitrogen | Cat# T10182 |
| Ketamine HCl | Akron | Cat# 59399-114-10 |
| Xylazine | Akron | Cat# 59399-110-20 |
| Collagenase, Type I | Worthington | Cat# LS004197 |
| Collagenase, Type II | Worthington | Cat# LS004179 |
| Dispase II | Roche | Cat# 04942078001 |
| Zombie Violet Live/Dead dye | Biolegend | Cat# 423113 |
| Red blood cell lysis solution | Promega | Cat# Z3141 |
| Ham's F-10 Nutrient Mix | Gibco | Cat# 11550043 |
| Dulbecco's Modified Eagle Medium | Gibco | Cat# 11995065 |
| DMEM, low glucose, pyruvate | Gibco | Cat# 11885084 |
| Fetal bovine serum | HyClone | Cat# SH30080.03 |
| Donor Horse Serum | Corning | Cat# MT35030CV |
| Penicillin–Streptomycin | Sigma-Aldrich | Cat# P4333 |
| Etomoxir | Sigma-Aldrich | Cat# E1905 |
| Perhexiline (maleate) | Cayman Chemical | Cat# 16982 |
| Oxfenicine | TCI AMERICA | Cat# H1389 |
| Seahorse XF DMEM Medium pH 7.4 | Agilent Technologies | Cat# 103575-100 |
| Oligomycin | Sigma-Aldrich | Cat# O4876 |
| FCCP | Tocris Biosciences | Cat# 045310 |
| Rotenone | Sigma-Aldrich | Cat# 557368 |
| Antimycin A | Sigma-Aldrich | Cat# A8674 |
| Phosphate-buffered saline (PBS) | Gibco | Cat# 21600-069 |
| M.O.M. (Mouse on Mouse) Blocking Reagent | Vector lab | Cat# MKB-2213 |
| Hoechst33342 | Invitrogen | Cat# H3570 |
| DAPI | Invitrogen | Cat# D1306 |
| Paraformaldehyde | Sigma-Aldrich | Cat# P6148 |
| Glycine | Sigma-Aldrich | Cat# 50046 |
| Sucrose | Fisher Scientific | Cat# S5-500 |
| O.C.T. Compound | Fisher Scientific | Cat# 23-730-571 |
| Fibroblast growth factor, basic | Promega | Cat# 9PIG507 |
| BD Matrigel Matrix | BD Biosciences | Cat# 356235 |
| Collagen from rat tail | Sigma-Aldrich | Cat# C7661 |

| Reagent/resource | Reference or source | Identifier or catalog number |
|---|---|---|
| 5-Ethynyl-2'-deoxyuridine (EdU) | Cayman Chemical | Cat# 20518 |
| Tetramethylrhodamine (TAMRA) azide | Invitrogen | Cat# T10182 |
| Ethanol | Decon Labs, Inc | Cat# 2701 G |
| Nonfat dry milk | Fisher Scientific | Cat# NC9952266 |
| EMbed-812 kit | Fisher Scientific | Cat# 50-980-391 |
| Goat serum | MP Biomedicals | Cat# 08642921 |
| Bovine Serum Albumin | GeminiBio | Cat# 700-105 P |
| Bovine Serum Albumin (fatty acid free) | Sigma-Aldrich | Cat# A6003 |
| Palmitic acid (U-$^{13}C_{16}$) | Cambridge Isotope Laboratories | Cat# CLM-409-0.5 |
| [1-14 C] Palmitic acid | Moravek | Cat# MC121 |
| [14 C(U)] D-Glucose | Moravek | Cat# MC144 |
| Palmitic acid | Sigma-Aldrich | Cat# P5585 |
| Glucose | Sigma-Aldrich | Cat# G7021 |
| TRI Reagent | Sigma-Aldrich | Cat# T9424 |
| Chloroform | VWR Chemicals | Cat# BDH1109 |
| Methanol | Fisher Scientific | Cat# A412-20 |
| NP-40 | ThermoFisher Scientific | Cat# 85124 |
| Sodium deoxycholate | Sigma-Aldrich | Cat# D6750 |
| Sodium dodecyl sulfate (SDS) | Fisher Scientific | Cat# 02-004-080 |
| Ammonium acetate | Fisher Scientific | Cat# A637-500 |
| Tris Base | Fisher Scientific | Cat# BP152-10 |
| Sodium Chloride | Fisher Scientific | Cat# S2711 |
| EDTA disodium salt | Sigma-Aldrich | Cat# E4884 |
| EGTA | Sigma-Aldrich | Cat# 324626 |
| Triton X-100 | Sigma-Aldrich | Cat# T8787 |
| Perchloric acid | Sigma-Aldrich | Cat# 244252 |
| Protease Inhibitor Cocktail | Sigma-Aldrich | Cat# P8340 |
| PMSF | Calbiochem | Cat# 7110-OP |
| L-Glutamine solution | Sigma-Aldrich | Cat# G7513 |
| Sodium pyruvate | Sigma-Aldrich | Cat# P5280 |
| Protein A/G PLUS-Agarose | Santa Cruz Biotechnology | Cat# sc-2003 |
| Normal rabbit IgG | Cell Signaling Technology | Cat# 2729 |
| Formic acid | Sigma-Aldrich | Cat# 5.33002 |
| Acetonitrile | Sigma-Aldrich | Cat# 1.00029 |
| Potassium Bicarbonate (KHCO3) | Fisher Scientific | Cat# ICN15255780 |
| Phenethylamine | Sigma-Aldrich | Cat# 241008 |
| Hydrochloric acid (HCl) | Fisher Scientific | Cat# A142-212 |
| M-MLV reverse transcriptase | Invitrogen | Cat# 28025021 |
| PicoPure RNA Isolation Kit | Applied Biosystems | Cat# DUO92101 |
| FastStart Essential DNA Green Master | Roche | Cat# 06402712001 |
| Pierce BCA Protein Assay Reagent | ThermoFisher Scientific | Cat# 23225 |
| Western Blotting Chemiluminescence Luminol Reagent | Santa Cruz Biotechnology | Cat# sc-2048 |
| Duolink In Situ Fluorescence Kit | Sigma-Aldrich | Cat# K578-100 |
| PicoProbe Acetyl-CoA Fluorometric Assay kit | BioVision Inc | Cat# K317 |
| ATP Detection Assay Kit | Cayman Chemical | Cat# 700410 |
| Seahorse XF Cell Mito Stress Test Kit | Agilent Technologies | Cat# 103015-100 |

| Reagent/resource | Reference or source | Identifier or catalog number |
|---|---|---|
| CF488A TUNEL Assay Apoptosis Detection Kit | Biotium | Cat# 30063 |
| **Software and algorithms** | | |
| Seurat v3.1 | Stuart et al, 2019 | Satija lab: satijalab.org/seurat/ |
| Cell Ranger v3.1 | 10x Genomics | https://support.10xgenomics.com/single-cell-gene-expression/software/downloads/3.0/#cellrangertab |
| RStudio | RStudio Team, 2015 | RRID:SCR_000432 |
| TopHat2 | Kim et al, 2013 | RRID:SCR_013035 |
| HTSeq V0.6.1 | Anders et al, 2015 | RRID:SCR_005514 |
| DESeq2 | Love et al, 2014 | RRID:SCR_015687 |
| BD FACSDiva Software | BD Biosciences | RRID:SCR_001456 |
| FlowJo 10 | FLOWJO, LLC | RRID:SCR_008520 |
| Fiji-ImageJ | https://imagej.net/software/fiji/ | RRID:SCR_002285 |
| MetaMorph Microscopy Automation and Image Analysis Software | Molecular Devices, LLC | RRID:SCR_002368 |
| Agilent Masshunter Quantitative Analysis software | Agilent Technologies | RRID:SCR_015040 |
| Seahorse Wave | Agilent Technologies | RRID:SCR_014526 |
| Adobe Photoshop | Adobe Inc. | RRID:SCR_014199 |
| Prism 8.0 | GraphPad Prism | RRID:SCR_002798 |
| **Equipment** | | |
| Leica CM1850 cryostat | Leica Biosystems | N/A |
| Leica DM 6000B microscope | Leica Microsystems | N/A |
| Roche Lightcycler 96 Realtime PCR system | Roche | N/A |
| FluorChem R System | ProteinSimple | N/A |
| Seahorse XFe24 Analyzer | Agilent Technologies | N/A |
| Agilent 6470 series QQQ mass spectrometer | Agilent Technologies | N/A |
| Agilent 1260 Rapid Resolution liquid chromatography (LC) system | Agilent Technologies | N/A |
| Liquid scintillation counter | Beckman Coulter | N/A |
| Spark 10 M multimode microplate reader | TECAN | N/A |

## Mice

*Cpt2^{flox/flox}* mouse was generated by Dr. Michael J. Wolfgang lab (Johns Hopkins University School of Medicine) and provided by Dr. Jessica M. Ellis (East Carolina University) (Lee et al, 2015). All other mouse strains were obtained from Jackson Laboratory (Bar Harbor, ME) under the following stock numbers: *Pax7^{CreER}* (#012476), *Pax7^{CreERT2(Gaka)}* (#017763), and *ROSA26^{LSL-sfGFP}* (#021039). Mice were genotyped by PCR of ear DNA using genotyping protocols described by the supplier. The genotypes of experimental KO and associated control animals are as follows: *Cpt2^{PKO}* (*Pax7^{CreER}/+;Cpt2^{flox/flox}*) and wild type (*Cpt2^{flox/flox}*). *Pax7^{CreERT2(Gaka)};ROSA26^{LSL-sfGFP}* mice were used as reporter mice for cell sorting. Mice were housed and maintained in the animal facility with free access to standard rodent chow and water. All procedures involving mice were approved by the Institutional Animal Care and

Use Committee. If not stated differently, 2- to 6-month-old mice were used for all experiments. All mice were randomly allocated into each experimental groups from different sex and litters, but always sex- and age-matched for each specific experiment.

## In vivo treatment

Tamoxifen (TMX, Calbiochem) was prepared in corn oil at a concentration of 10 mg/ml, and experimental and control mice were injected intraperitoneally with 0.1 mg TMX per gram body weight per day for 5 days to induce Cre-mediated deletion. TMX injections were initiated on adult mice, and experimental mice were used at the time stated in the text. 5-ethynyl-2-deoxyuridine (EdU, Carbosynth) (0.3 mg/ml) was administrated to mice by drinking water 24 h prior to the mouse sacrifice. Sodium acetate was administrated uninterruptedly through drinking water (200 mM) with the combination of IP injection (0.41 g/kg body weight) consecutively as indicated days.

## Muscle injury and regeneration

Muscle regeneration was induced by cardiotoxin (CTX) injection. Adult mice were anesthetized using a ketamine–xylazine cocktail and CTX was injected (50 µl of 10 µM solution, Sigma-Aldrich) into tibialis anterior (TA) muscle. Muscles were then harvested at the stated time to assess the completion of regeneration and repair.

## Flow cytometry

Satellite cell isolation by flow cytometry was performed as previously described (Liu et al, 2015). Briefly, hind limb muscles were minced and digested with 700 U/ml type II collagenase (Worthington) at 37 °C for 1 h. Samples were then centrifuged and digested with 100 U/ml type II collagenase and 1 U/ml Dispase II (Roche) at 37 °C for 30 min. Each sample was consecutively filtered through 70-µm cell strainers. Cell suspension was then centrifuged, washed with Ham's F-10 Nutrient Mix medium containing 10% horse serum. For WT and $Cpt2^{PKO}$ mice, cells were stained with antibody cocktail including anti-CD31-PE, anti-CD45-PE, anti-Sca1-Pacific Blue, and anti-VCAM1-APC (details listed in Reagents and tools table) for 20 min at 4 °C, prevent from light. Satellite cells were identified by gating with CD31$^-$, CD45$^-$, Sca1$^-$, and VCAM1$^+$ using a BD-FACS Aria III fluorescence activated cell sorting (FACS) system (BD Biosciences). FACS-isolated SCs were cultured in Matrigel-coated 24-well plates (NUNC) in Ham's F-10 Nutrient Mix medium supplemented with fetal bovine serum (FBS, HyClone), 4 ng/ml basic fibroblast growth factor (Promega), and 1% penicillin–streptomycin (Sigma-Aldrich). To induce the differentiation, proliferating cells were incubated with Dulbecco's Modified Eagle's Medium (DMEM, Gibco) supplemented with 2% horse serum (Corning). In addition, satellite cells were isolated from hind limb muscles of $Pax7^{CreERT2(Gaka)}$;$ROSA26^{LSL-sfGFP}$ reporter mice with or without injury. After digestion as described above, cells were stained with zombie Live/Dead dye (Biolegend) and live GFP$^+$ satellite cells were sorted for gene expression assay.

## Single myofiber isolation and culture

Single muscle fibers were isolated from EDL muscles of adult mice and cultured as previously described (Pasut et al, 2013). In brief,

EDL muscles were dissected carefully and subjected to digestion with collagenase I (2 mg/ml, Worthington) in DMEM for 1 h at 37 °C. Digestion was stopped by carefully transferring EDL muscles to a pre-warmed Petri dish (60-mm) with 6 ml of DMEM and single myofibers were released by gently flushing muscles with large bore glass pipette. Released single myofibers were then transferred and cultured in a horse serum-coated Petri dish (60-mm) in DMEM supplemented with 20% FBS, 4 ng/ml basic FGF, and 1% penicillin–streptomycin at 37 °C for indicated days.

## Satellite cell-derived primary myoblast culture and differentiation

Satellite cell-derived primary myoblasts were isolated from hind limbs of age of 4–6 weeks mice. Muscles were minced and digested in type I collagenase and Dispase II mixture. The digestions were stopped with F-10 Ham's medium containing 20% FBS. Cells were then filtered from debris, centrifuged and cultured in growth medium (Ham's F-10 Nutrient Mix medium supplemented with 20% FBS, 4 ng/ml basic FGF, and 1% penicillin–streptomycin) on collagen-coated cell culture plates at 37 °C, 5% $CO_2$. For differentiation, primary myoblasts were seeded on Matrigel-coated culture plates and induced to differentiate in differentiation medium (DMEM supplemented with 2% horse serum and 1% penicillin–streptomycin). For in vitro inhibition of FAO, different concentrations of etomoxir (Sigma-Aldrich, 10 µM, 50 µM, 100 µM, and 200 µM) (Ma et al, 2020; Nomura et al, 2016; O'Sullivan et al, 2014; Raud et al, 2018), Perhexiline (Cayman Chemical, 2.5 µM and 5 µM) (Kaczara et al, 2024), and Oxfenicine (TCI AMERICA, 2.5 mM and 10 mM) (Hermann et al, 2018; Ma et al, 2020; Patella et al, 2015) were added into the culture medium, respectively, and cells were collected after 2 days' treatment. For genetic deletion in culturing primary myoblasts, 4-OH TMX (0.4 mM, Sigma-Aldrich) was added in culture medium for 2 days.

## Hematoxylin–eosin and immunofluorescence staining

Whole muscle tissues were dissected and frozen immediately in Optimal cutting temperature compound (OCT compound). Frozen muscles were cross sectioned (10 mm) using a Leica CM1850 cryostat. For hematoxylin and eosin staining, the slides were first stained in hematoxylin for 15 min, rinsed in running tap water and then stained in eosin for 1 min. Slides were dehydrated in graded ethanol and Xylene, and then covered using Permount.

For immunofluorescence staining, cross-sections, single myofibers or cultured cells were fixed in 4% PFA in PBS for 10 min, quenched with 100 mM glycine for 10 min, and incubated in blocking buffer (5% goat serum, 2% bovine serum albumin, 0.1% Triton X-100 and 0.1% sodium azide in PBS) for at least 1 h, followed by 1 h blocking in M.O.M. mouse-on-mouse blocking reagent (Vector lab, MKB-2213). Samples were then incubated with primary antibodies diluted in blocking buffer overnight at 4 °C. After washing with PBS, samples were incubated with secondary antibodies and DAPI for 1 h at room temperature. Antibodies used for immunofluorescence staining were listed in Reagents and tools table. For EdU staining, samples with EdU incorporation were first fixed in 4% PFA in PBS for 10 min. EdU was visualized by Click-iT method (Salic et al, 2008) with red fluorescent dye tetramethylrhodamine azide (Invitrogen). Samples were then subjected to Pax7 staining.

All hematoxylin and eosin staining images were captured using a Nikon D90 digital camera mounted on a microscope with a ×20 objective. All immunofluorescent images were captured using a Leica DM 6000B microscope with a × 20 objective, or Zeiss LSM 700 Confocal with a ×20 objective. Images for WT and KO samples were captured using identical parameters. The number of regenerating and regenerated myofibers per area, CSA of regenerated fibers, and percentage of regenerated area were calculated by ImageJ software. All images shown are representative results of at least three biological replicates.

## In situ TUNEL assay

For TUNEL and Pax7 staining, cultured single myofibers were fixed in 4% PFA for 10 min and then subjected to the TUNEL reaction using the CF488A TUNEL Assay Apoptosis Detection Kit (Biotium) according to the manufacturer's instructions. For negative control, samples were added TUNEL reaction buffer without TdT Enzyme. Samples treated with DNaseI for 30 min before TUNEL staining was set up as positive control. Counterstaining of Pax7 was then performed as regular immunofluorescence staining procedure.

## Single cell RNA-seq data analysis

Single cell RNA-seq data analysis was performed as previously reported (Yue et al, 2022). In brief, Seurat's SCTransform was used for data normalization and the top 10 principal components were used for Louvain clustering and Uniform manifold approximation and projection (UMAP) embedding. Clustering results were analyzed to determine if there was any contamination from non-SCs. Cells in $Lyz2^+$ clusters were removed, as these represented contaminating immune cells from both 5.5 and 10 dpi. Data were then re-analyzed using Seurat's SCTransform, selecting the top 10 PCs for Louvain clustering (resolution of 0.05) and subsequent UMAP embedding. FindAllMarkers function was used to evaluate gene expression enriched in each cluster with a p value of less than 0.05 (Wilcoxon rank-sum test) and ranked by avgLog2Fold-change. The top 20–30 markers were then visualized via Heatmap to determine specificity and aide with nomenclature for cluster identities. Further mapping of known SC markers at various stages of quiescence, proliferation, activation, and differentiation were also evaluated to aide with cluster annotation. These markers included *Pax7*, *Myog*, *Mki67*, *Tnnt2*, and *Acta2*.

For GO term analysis, gene lists for ETC (GO:0022900), TCA cycle (GO:0006099), OxPhos (GO:0006119), mitochondrial FAO (GO:00019395), pyruvate metabolism (GO:0006090), and amino acid metabolism (GO:0006520), Glycolytic process (GO:0006096), lactate catabolism (GO: 0019244), and Acetyl-CoA formation from pyruvate (GO:0006086) were downloaded from MGI. Genes annotated with either positive or negative regulatory roles were excluded from the GO term pathway to avoid contradiction. In addition, genes characterized into peroxisomal FAO in GO:0006635 were analyzed separately from mitochondrial FAO to better compare the two pathways. Density plots were generated to visualize gene sets for GO terms using AddModuleScore function in Seurat, with the control features set as all genes expressed and then plotted using the RidgePlot function.

## Total RNA extraction and real-time PCR

Total RNA was extracted from tissues using TRIzol reagent according to the manufacturer's instructions. The purity and concentration of the total RNA were determined by a spectrophotometer Nanodrop 2000c (ThermoFisher Scientific Inc). In total, 3 μg of total RNA was reverse transcribed using random primers with M-MLV reverse transcriptase (Invitrogen). Real-time PCR was carried out in a Roche Lightcycler 96 Realtime PCR system (Roche) with FastStart Essential DNA Green Master (Roche) and gene-specific primers for *Cpt2* and *18 s* rRNA The $2^{-\Delta\Delta Ct}$ method was used to analyze the relative changes in each gene's expression normalized against *18 s* rRNA expression.

## Bulk RNA-seq and bioinformatics analysis

Total RNA was extracted from FACS-isolated SCs after 7 days culture using TRIzol reagent according to the manufacturer's instructions, and subjected to RNA-seq analysis. Briefly, RNA quality analysis was performed by Agarose Gel Electrophoresis and Agilent 2100. A complementary DNA library was then constructed using polyA selected RNA, and sequencing was performed according to the Illumina HiSeq standard protocol. Raw reads from RNA-seq libraries are filtered to remove reads containing adapters or with low quality. Statistics analysis of data production and quality was performed to confirm the sequencing quality. Reference genome and gene annotation files were downloaded from a genome website browser (NCBI/UCSC/Ensembl). TopHat2 was used for mapping the filtered reads to the reference genome. For the quantification of gene expression level, HTSeq V0.6.1 was used to analyze the read numbers mapped for each gene. The FPKM of each gene was calculated based on the gene read counts mapped to genes or exons. A differential expression analysis was performed using the DESeq2 R/EdgeR R package with the threshold of significance set as adjusted $P < 0.05$.

## Protein extraction and western blot analysis

Total protein was isolated from cells using RIPA buffer containing 25 mM Tris-HCl (pH 8.0), 150 mM NaCl, 1 mM EDTA, 0.5% NP-40, 0.5% sodium deoxycholate and 0.1% Sodium dodecyl sulfate (SDS) with Protease Inhibitor Cocktail (Sigma-Aldrich) and 1 mM PMSF (Sigma-Aldrich). Protein concentrations were determined using Pierce BCA Protein Assay Reagent (Pierce Biotechnology). Proteins were separated by SDS-PAGE, transferred to a polyvinylidene fluoride membrane (Millipore Corporation), blocked in 5% Nonfat dry milk for 1 h at room temperature and then incubated with primary antibodies in 5% milk overnight at 4 °C. Membrane was then incubated with secondary antibody for 1 h at room temperature. Antibodies used for western blot analysis were listed in Reagents and tools. Immunodetection was performed using enhanced chemiluminescence western blotting substrate (Santa Cruz Biotechnology) and detected with a FluorChem R System (Proteinsimple).

## Immunoprecipitation assay

Satellite cell-derived primary myoblasts from WT or $Cpt2^{PKO}$ mice were rinsed with PBS and harvested in 500 μl of lysis buffer per 10-

cm dish. Cellular debris was pelleted and the lysates were subjected to immunoprecipitation with either 4 μg of Pax7 or Acetylated lysine antibody or 4 μg of mouse/rabbit IgG control IgG for overnight at 4 °C with gentle rotation. Protein A/G-agarose (Santa Cruz Biotechnology) was then added into the lysates and incubated for another 2 h at 4 °C with gentle rotation. The beads were washed extensively for six times in lysis buffer, and subjected to western blot analysis.

## In situ proximity ligation assay (PLA)

In situ protein interaction was detected by using Duolink In Situ Fluorescence Kit (Sigma, Cat# DUO92101) according to the manufacturer's instructions with proper optimizations. Satellite cell-derived primary myoblasts grew on collagen-coated glass slides were fixed in 4% paraformaldehyde for 10 min and quenched with 100 mM glycine for 10 min. Slides were incubated with Blocking Solution in a pre-heated humidity chamber at 37 °C for 1 h, and then incubated with primary antibodies at 4 °C overnight. After wash, slides were incubated with PLA probes (Anti-mouse MINUS and Anti-rabbit PLUS) for 1 h at 37 °C, followed with the ligation reaction for another 1 h. After washing off the Ligation-Ligase solution, Amplification-Polymerase solution containing oligonucleotide probes that labeled with a red fluorophore was added to perform rolling circle amplification. The slides were then washed, counterstained with DAPI and mounted with a cover slip for imaging.

## Seahorse mitochondrial respiration analysis

Mitochondrial respiration was measured with Seahorse XFe24 Analyzer (Agilent Technologies, CA) according to the Seahorse XF Cell Mito Stress Test Kit User Guide. Briefly, Satellite cells isolated by FACS were plated on Matrigel-coated XF24 cell culture microplate at a density of 10,000 cells per well. Seahorse sensor cartridge was hydrated with calibrant in a non-$CO_2$ incubator at 37 °C for overnight one day before measurements. On the day of measurements, cells were washed twice and switched to Seahorse bicarbonate-free DMEM (pH 7.4, Agilent Technologies) supplemented with 1 mM sodium pyruvate (Sigma-Aldrich), 1 mM glutamine (Sigma-Aldrich) and 5 mM glucose. Cells were equilibrated at 37 °C non-$CO_2$ incubator for 1 h. For FAO inhibition, Etomoxir were added to cells at a final concentration of 50 μM 15 min prior to the measurement. Oxygen consumption rate (OCR) was monitored at basal state and after sequential injection of the mitochondrial compounds oligomycin (1.5 μM), FCCP (3 μM) and Rotenone/antimycin A (1 μM) that induce mitochondrial stress. All mitochondrial respiration rates were generated and automatically calculated by the Seahorse XF Cell Mito Stress Test Report Generator.

## 13C-palmitate labelling and metabolite profiling by LC/MS

Detection of 13C-palmitate incorporation in TCA cycle metabolites was performed as previously described with proper modification (Schoors et al, 2015). 13C-palmitate was conjugated with fatty acid free BSA (Sigma-Aldrich, A-6003) to generate a 4 mM 13C-palmitate-BSA complex. Satellite cell-derived primary myoblasts were incubated for 24 h with 13C-palmitate-BSA at a concentration of 200 μM. Cells were washed for 3 times with PBS and harvested for metabolite extraction and metabolite profiling as previously reported (Tan et al, 2014).

An Agilent 1260 Rapid Resolution liquid chromatography (LC) system coupled to an Agilent 6470 series QQQ mass spectrometer (MS/MS) was used to analyze TCA intermediates (Agilent Technologies). A Water's Xbridge C8 2.1 mm ×100 mm, 3.5 μm column was used for LC separation (Water's Corporation). The buffers were (A) water + 0.1% formic acid and (B) acetonitrile + 0.1% formic acid. The linear LC gradient was as follows: time 0 min, 10% B; time 0.5 min, 10% B; time 8 min, 100% B; time 10 min, 100% B; time 11 min, 10% B; time 16 min, 10% B. The flow rate was 0.3 ml/min. Multiple reaction monitoring (MRM) was used for MS analysis. Data were acquired in positive electrospray ionization (ESI) mode according to Appendix Table S1. The jet stream ESI interface had a gas temperature of 325 °C, gas flow rate of 7 l/min, nebulizer pressure of 45 psi, sheath gas temperature of 250 °C, sheath gas flow rate of 7 l/min, capillary voltage of 3500 V in positive mode, and nozzle voltage of 500 V. The ΔEMV voltage was 400 V. Agilent Masshunter Quantitative analysis software was used for data analysis (version 8.0) based on the retention time for each. For each metabolite, the relative ratio of isotope 13 C incorporation (M + 1 to M + 6) was calculated by normalizing to the metabolite content of M + 0 (no 13 C incorporation).

## Measurement of metabolic fluxes using radioactive isotope tracers

Both [1-$^{14}$C] and unlabeled palmitic acid were bound to 10% fatty acid free BSA in DMEM prior to the experiment. The incubation medias were calculated to provide a final concentration of 0.5 mM palmitic acid containing 500,000 decays per minute (dpm) in 2 ml of DMEM. The [1-$^{14}$C] palmitic acid incubation media contained: 0.5 mM palmitic acid (including the radiolabel), 0.5% BSA, and low glucose DMEM (Gibco). The [14 C(U)] glucose incubation media contained: 0.5 mM palmitic acid, [14 C(U)] glucose, 0.5% BSA, and low glucose DMEM (ThermoFisher Scientific). The radiolabeled glucose was combined with unlabeled palmitic acid bound to BSA prior to the initiation of the experiment.

For the measurement, primary myoblasts were passaged into 35-mm culture dishes, and grown to 80% confluence and used to trace substrate oxidation to $CO_2$. After transferring to a chamber, cells were incubated with low glucose DMEM supplied either 0.5 mM [1-$^{14}$C] palmitic acid previously bound to BSA or 5.5 mM [14 C(U) glucose previously combined with 0.5 mM unlabeled palmitic acid bound to BSA. The chamber was gassed with 95% $O_2$/5% $CO_2$ for 3 s, capped, and immediately placed in the cell culture incubator set at 37 °C. After 6 h, the chemical reaction with the radiolabel was terminated by the addition of 0.5 mL of 1 N HCl directly to the media, and 0.2 mL of phenethylamine was added to the center well containing the filter paper to trap the respired $CO_2$. After 1 h, the center wells containing the filter paper and phenethylamine were carefully transferred to a liquid scintillation vial, cocktail was added and $^{14}$C determined by liquid scintillation counting. The dpm determined by liquid scintillation counting was used to calculate the rate of substrate converted $CO_2$ adjusted for background counts over the six-hour incubation.

## Incorporation of PA-derived acetyl group in Pax7 protein

The [1-$^{14}$C] palmitic acid and primary myoblasts were prepared as described above. Primary myoblasts in 10-cm culture dishes containing F10 medium were grown to 80% confluence and supplied with 0.5 mM [1-$^{14}$C] palmitic acid previously bound to BSA combined with 0.5 mM unlabeled palmitic acid bound to BSA. Cells were placed in the sealed chamber in incubator. After 12 h, cells were washed with cold PBS and harvested in 500 μl of lysis buffer per 10-cm dish. Cellular debris was pelleted, and the lysates were subjected to immunoprecipitation with either 4 μg of Pax7 or 4 μg of mouse control IgG (Santa Cruz Biotechnology) for overnight at 4 °C with gentle rotation. Protein A/G-agarose (Santa Cruz Biotechnology) was then added into the lysates and incubated for another 2 h at 4 °C with gentle rotation. The beads were washed extensively for six times in lysis buffer and subjected to the measurement of radioactive activity using liquid scintillation counter.

## Cellular acetyl-CoA measurement

Cellular acetyl-CoA contents in satellite cell-derived primary myoblasts were measured by PicoProbe Acetyl-CoA Fluorometric Assay kit (BioVision) according to the manufacturer's instructions. In brief, primary myoblasts were rinsed with cold PBS and lysed with 1× ice-cold Cell Lysis Buffer (20 mM Tris (pH 7.5), 150 mM NaCl, 1 mM EDTA, 1 mM EGTA, 1% Triton X-100, 2.5 mM sodium pyrophosphate, 1 mM Glycerol phosphate, 1 mM Na$_3$VO$_4$, 1 μg/ml Leupeptin) plus 1 mM PMSF. Cell lysate was deproteinized with 1 N perchloric acid and then neutralized with 3 M KHCO$_3$. Samples were quenched for erasing free CoA and cellular acetyl CoA was converted to CoA by adding conversion enzyme. The CoA is reacted to form NADH which interacts with PicoProbe to generate fluorescence. The fluorescence intensity was measured at Ex/Em=535/587 nm with Spark 10 M multimode microplate reader (TECAN), and acetyl-CoA concentration was calculated based on the standards following the manufacturer's instructions.

## Cellular ATP detection

Cellular ATP levels were measured using an ATP Assay Kit (Cayman chemical). Satellite cell-derived primary myoblasts were rinsed with cold PBS and lysed with ice-cold 1× ATP Detection Sample Buffer. Cell lysates and standards were plated into a 96-well plate. In all, 100 μl of freshly prepared reaction mixture (1× ATP Detection Sample Buffer, D-Luciferin, Luciferase) was added to each sample and the plate was incubated at room temperature for 20 min, protected from light. Luminescence intensity was detected by Spark 10 M multimode microplate reader (TECAN), and ATP concertation was calculated according to the manufacturer's instructions.

## Statistical analysis

No statistical methods were used to predetermine the sample size. Sample sizes were selected based on the experiment type and the standard practice in the field of genetics and stem cell biology. That allows to determine statistical differences within isogenic animal cohorts and ex vivo and in vitro culture experiments. The researchers involved in the in vivo treatments were not completely blinded, but all images were randomly captured from sample and analyzed in a blinded manner. All analyses were conducted with Student's *t* test with a two-tailed distribution in PRISM software or one-way ANOVA with Tukey's post hoc comparison. Measurement values that were beyond the boundary determined by the interquartile range were considered as potential outliers and were excluded from statistical analyses. All experimental data are represented as mean ± s.e.m. Comparisons with *P* values < 0.05 were considered statistically significant.

## Data availability

Bulk RNA-seq data in FACS-isolated SCs have been deposited into the National Center for Biotechnology Information's Gene Expression Omnibus under accession code no. GSE158912 at. Single cell RNA-seq data used in this study are available under accession code no. GSE150366 at. All data reported in this paper will be shared by the lead contact upon request. No original code was reported that needed to reanalyze the data generated by this study. Any additional information required to reanalyze the data reported in this paper is available from the corresponding author upon request.

The source data of this paper are collected in the following database record: biostudies:S-SCDT-10_1038-S44318-025-00397-1.

## Peer review information

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

## Acknowledgements

This work was supported by grants from the National Institutes of Health to SK (NIH R01DK132819, R01AR078695, R01AR079235), and the Muscular Dystrophy Association to FY (MDA516161). This work was also supported by NIH Shared Instrumentation Grant S10DO20029. We thank Dr. Daoguo Zhou for access to confocal microscopy, Dr. Susan J Hilger for assistance on radioactive isotope labeling assay, and Jun Wu and Lijia Zhang for mouse colony maintenance and members of the Kuang laboratory for their kind assistance. We are grateful to Purdue Flow Cytometry and Cell Separation Facility for assistance on flow cytometry analysis, and Purdue Metabolite Profiling Facility for assistance on targeted metabolite detection.

## Author contributions

**Feng Yue**: Conceptualization; Data curation; Software; Formal analysis; Supervision; Funding acquisition; Validation; Investigation; Visualization; Methodology; Writing—original draft; Writing—review and editing. **Lijie Gu**: Data curation; Formal analysis; Validation; Investigation; Visualization; Methodology; Writing—original draft. **Jiamin Qiu**: Data curation; Formal analysis; Validation; Investigation; Visualization; Methodology; Writing—review and editing. **Stephanie N Oprescu**: Data curation; Software; Formal analysis; Validation; Visualization; Methodology; Writing—original draft. **Linda M Beckett**: Methodology. **Jessica M Ellis**: Resources; Writing—review and editing. **Shawn S Donkin**: Resources; Methodology; Writing—review and editing. **Shihuan Kuang**: Conceptualization; Resources; Supervision; Funding acquisition; Project administration; Writing—review and editing.

Source data underlying figure panels in this paper may have individual authorship assigned. Where available, figure panel/source data authorship is listed in the following database record: biostudies:S-SCDT-10_1038-S44318-025-00397-1.

## Disclosure and competing interests statement

The authors declare no competing interests.

# Expanded View Figures

**Figure EV1.   Meta-cluster analysis with top markers of each SC state.**

QSC quiescent SCs, SSC self-renewed SCs, ASC activated SCs, PSC proliferating SCs, CSC committed SCs, DSC differentiated SCs.

▶

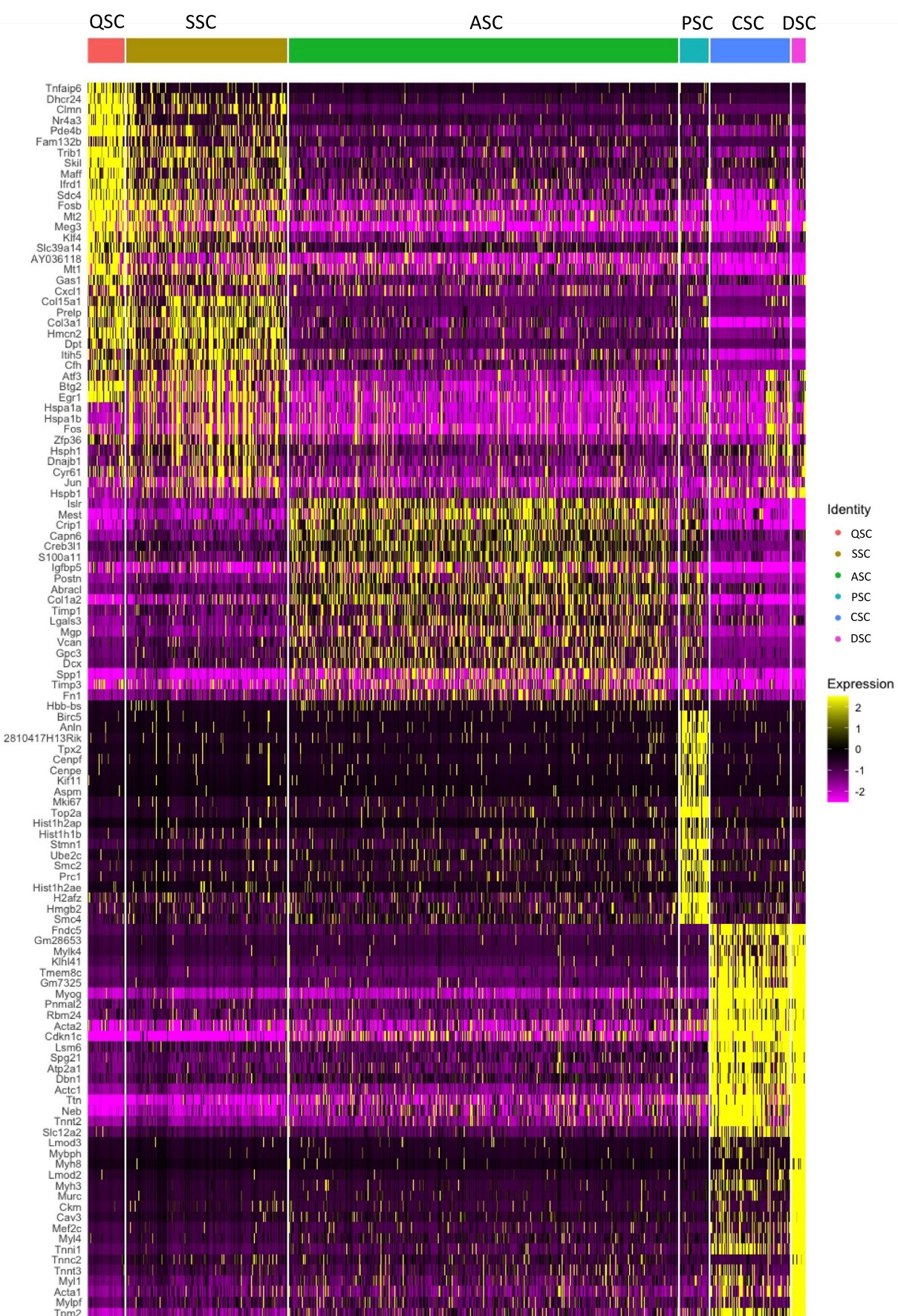

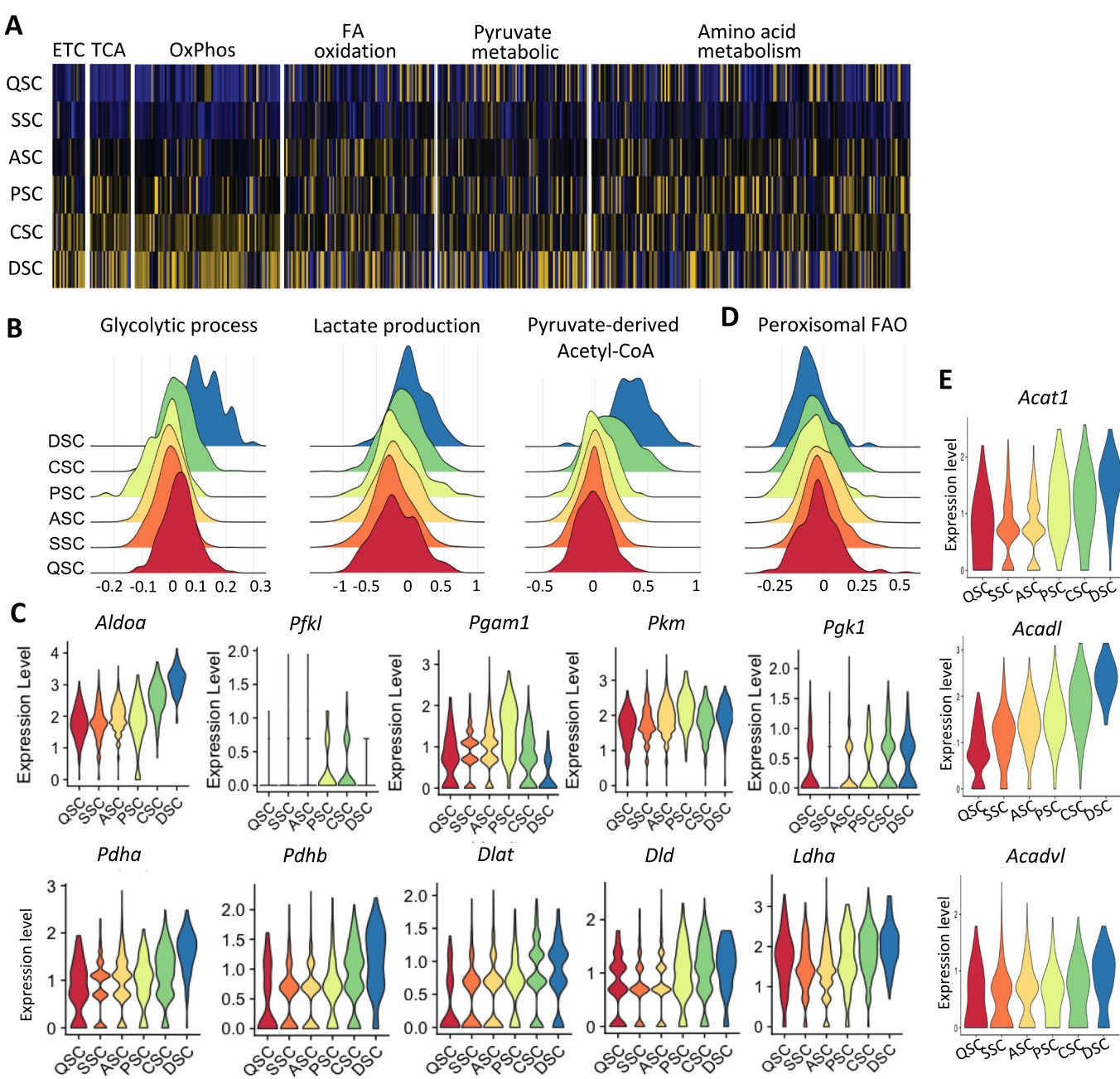

**Figure EV2. Dynamics of metabolic processes during fate transitions of satellite cell.**

(A) Heatmap showing enrichment of genes in various annotated Gene Ontology terms. ETC, GO:0022900; TCA cycle, GO:0006099; OxPhos, GO:0006119; FA oxidation, GO:00019395; pyruvate metabolism, GO: 0006090; amino acid metabolism, GO:0006520. (B) Density plot visualizing the enrichment of genes involving in glycolytic processes over different states. Glycolytic process, GO:0006096; lactate catabolism, GO: 0019244; acetyl-CoA biosynthetic process from pyruvate, GO: 0006086; (C) Violin plot showing the expression of key genes involved in glycolytic process (GO:0006096, top row) and acetyl-CoA biosynthetic process from pyruvate (GO: 0006086, bottom row). (D) Density plot visualizing the enrichment of genes involving in peroxisomal fatty acid oxidation (FAO) over different states. (E) Violin plot showing the expression of key genes involved in long-chain FAO among different states.

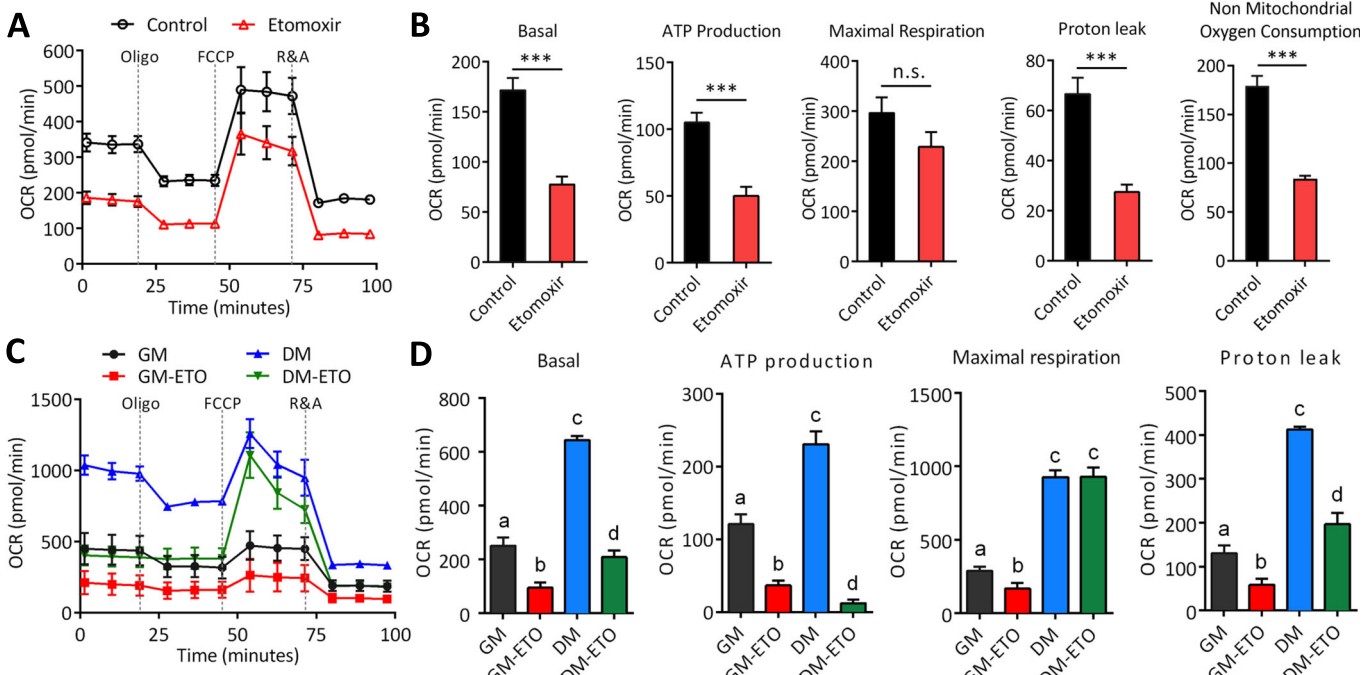

**Figure EV3. Pharmacological inhibition of FAO disturbs mitochondrial respiration and inhibits proliferation and differentiation of myoblasts.**

(A) Seahorse curves showing oxygen consumption rates (OCR) of primary myoblasts treated with vehicle control or 50 μM Etomoxir (a mitochondrial FAO inhibitor). $n = 3$ biological replicates, three technical replicates per run. (B) OCR associated with basal and maximal respiration, proton leak, ATP production and non-mitochondrial respiration, calculated based on data shown in (A). Error bars represent mean ± s.e.m. with $n = 3$ biological replicates (three technical replicates per run). ***$P < 0.001$; two tailed, unpaired Student's $t$ test (Basal, $P = 7.03 \times 10^{-5}$; ATP production, $P = 9.97 \times 10^{-5}$; maximal respiration, $P = 0.1796$; proton leak, $P = 0.00014$; non-mitochondrial oxygen consumption, $P = 9.7 \times 10^{-5}$). n.s. no significance. (C) Seahorse curve showing OCR of undifferentiated myoblasts and differentiated myotubes treated with vehicle control or Etomoxir. (D) Quantification of the OCR regarding to basal respiration, proton leak, ATP production and spare respiratory capacity measured from seahorse assay in (C). Error bars represent mean ± s.e.m. with $n = 4$ replicates. One-way ANOVA with Tukey's post hoc comparison was used.

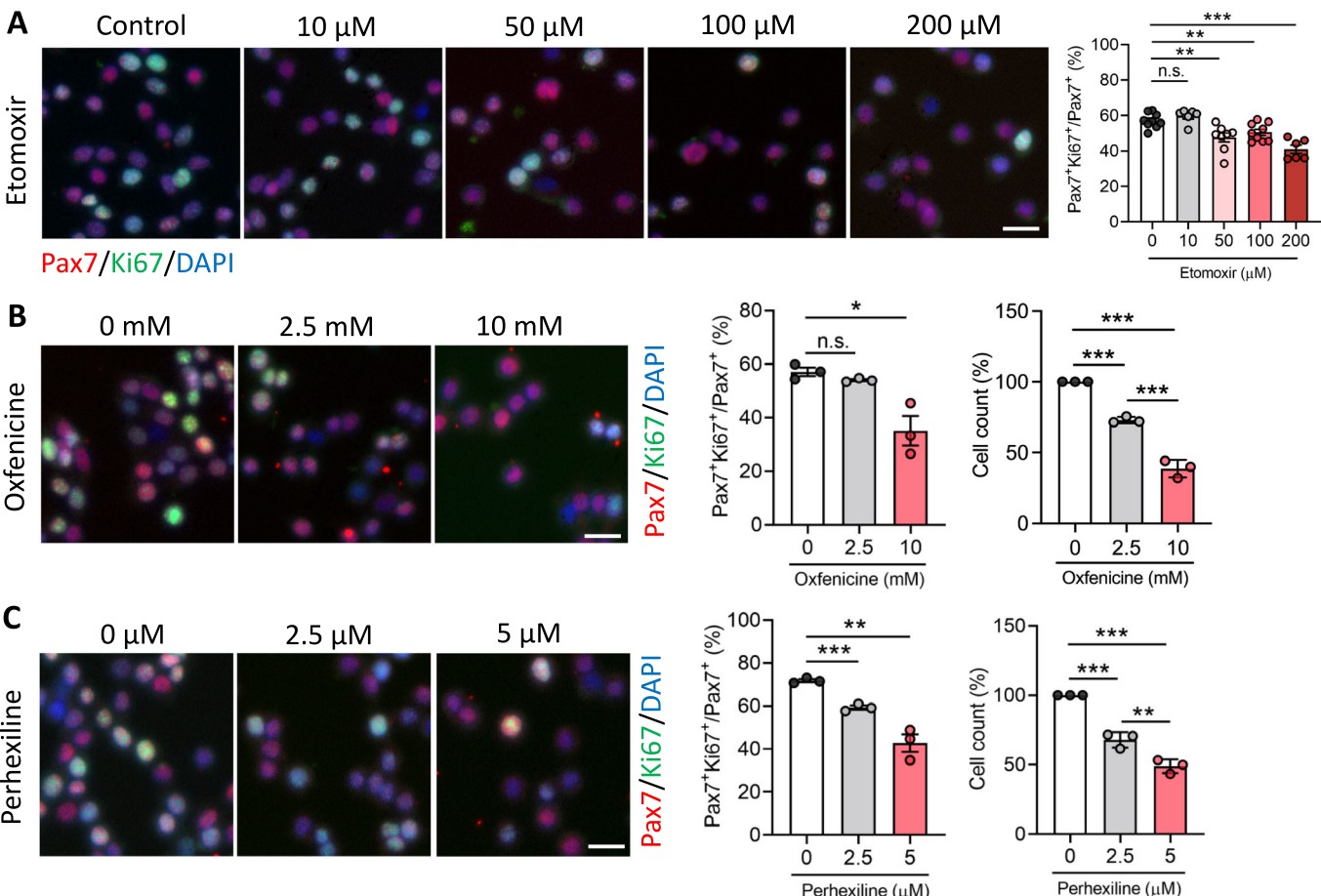

**Figure EV4.   Pharmacological inhibition of FAO inhibits the proliferation of myoblasts.**

(A) Ki67 immunofluorescence (green) and quantification of percentages of Ki67$^+$ primary myoblasts treated with vehicle control or Etomoxir. Error bars represent mean ± s.e.m. with $n = 6$–9 replicates. **$P < 0.01$, ***$P < 0.001$; two tailed, unpaired Student's $t$ test (10 μM vs. control, $P = 0.2519$; 50 μM vs. control, $P = 0.0042$; 100 μM vs. control, $P = 0.0087$; 200 μM vs. control, $P = 2.42 \times 10^{-5}$). n.s. no significance. Scale bar, 50 μm. (B) Ki67 immunofluorescence (green) and quantification of percentages of Ki67$^+$ primary myoblasts treated with vehicle control or Oxfenicine. Error bars represent mean ± s.e.m. with $n = 3$ replicates. *$P < 0.05$, ***$P < 0.001$; two tailed, unpaired Student's $t$ test. For Ki67$^+$/Pax7$^+$ percentage, 2.5 mM vs. control, $P = 0.13$; 10 mM vs. control, $P = 0.0187$. For cell count (%), 2.5 mM vs. control, $P = 0.00016$; 10 mM vs. control, $P = 7.97 \times 10^{-7}$; 2.5 mM vs. 10 mM, $P = 0.00017$; n.s. no significance. Scale bar, 50 μm. (C) Ki67 immunofluorescence (green) and quantification of percentages of Ki67$^+$ primary myoblasts treated with vehicle control or Perhexiline. Error bars represent mean ± s.e.m. with $n = 6$–9 replicates. **$P < 0.01$, ***$P < 0.001$; two tailed, unpaired Student's $t$-test. For Ki67$^+$/Pax7$^+$ percentage, 2.5 μM vs. control, $P = 0.00049$; 5 μM vs. control, $P = 0.00231$; For cell count (%), 2.5 μM vs. control, $P = 0.00011$; 5 μM vs. control, $P = 7.43 \times 10^{-5}$; 2.5 μM vs. 5 μM, $P = 0.0037$. Scale bar, 50 μm.

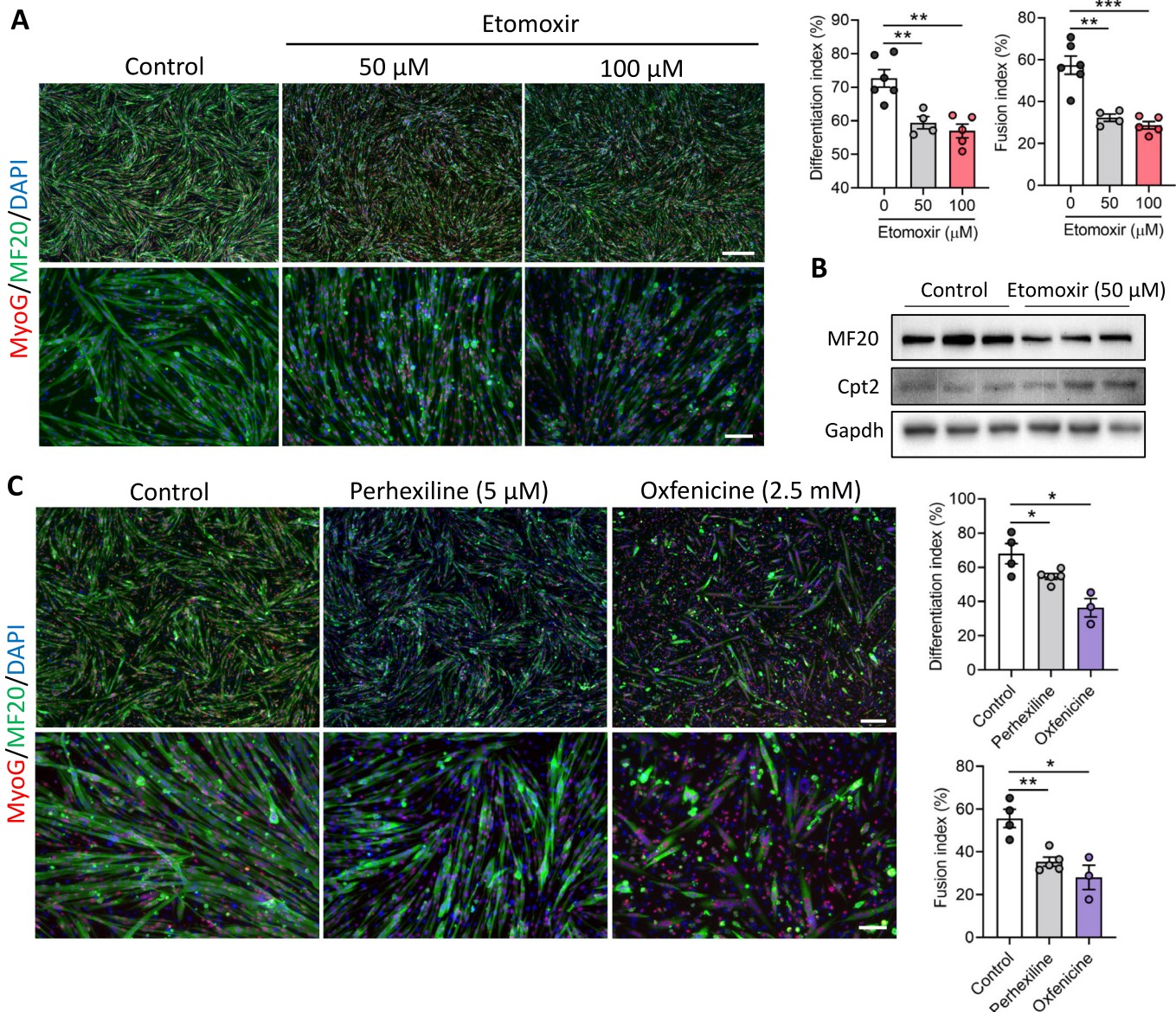

**Figure EV5.  Pharmacological inhibition of FAO inhibits the differentiation of primary myoblasts.**

(**A**) Immunofluorescence of MyoG (red) and MF20 (green) and quantification of differentiation and fusion index in primary myoblasts treated with vehicle control or Etomoxir. Error bars represent mean ± s.e.m. with $n = 4$–6 replicates. **$P < 0.01$, ***$P < 0.001$; two tailed, unpaired Student's $t$-test. For differentiation index, 50 μM vs. control, $P = 0.0057$; 100 μM vs. control, $P = 0.0013$. For fusion index, 50 μM vs. control, $P = 0.0021$; 100 μM vs. control, $P = 0.00032$. Scale bar, 500 μm (top); 100 μm (bottom). (**B**) Immunoblots showing relative levels of MF20 and Cpt2 in primary myoblasts treated with vehicle control or Etomoxir (50 μM). (**C**) Immunofluorescence of MyoG (red) and MF20 (green) and quantification of differentiation and fusion index in primary myoblasts treated with vehicle control or Perhexiline (5 μM) and Oxfenicine (2.5 mM). Error bars represent mean ± s.e.m. with $n = 3$–6 replicates. *$P < 0.05$, **$P < 0.01$; two tailed, unpaired Student's $t$ test. For differentiation index, Perhexiline vs. control, $P = 0.0472$; Oxfenicine vs. control, $P = 0.012$. For fusion index, Perhexiline vs. control, $P = 0.0025$; Oxfenicine vs. control, $P = 0.01052$. Scale bar, 200 μm (top); 100 μm (bottom).

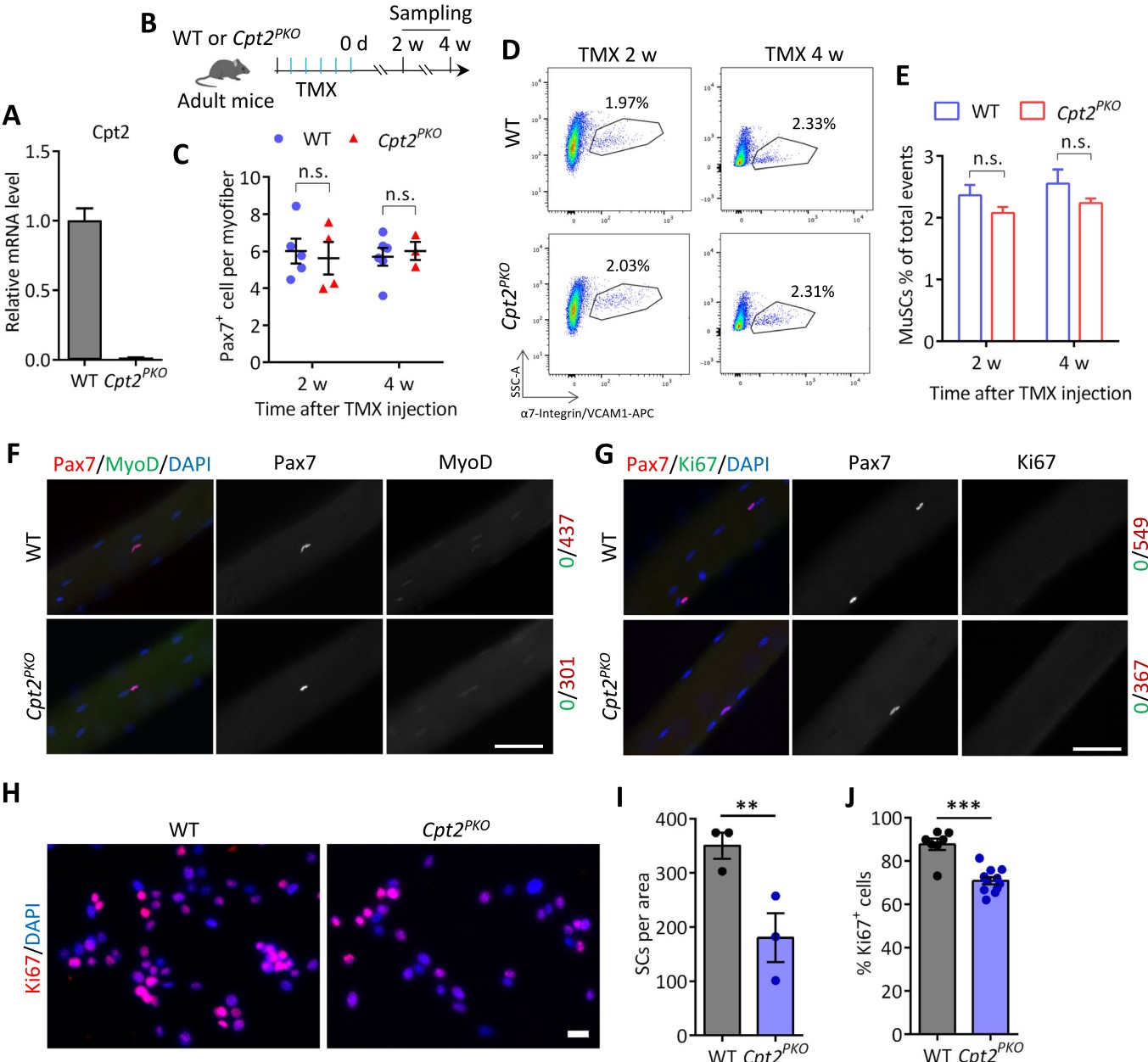

**Figure EV6.   Loss of *Cpt2* does not affect the quiescent maintenance of satellite cells (SCs) but inhibits their proliferative rate upon activation.**

(A) qPCR analysis to validate the KO efficiency of CPT2 gene in FACS-sorted SCs at 3.5 dpi. (B) Schematic illustration of experimental design. TMX injection was used to induce *Cpt2* knockout in the *Cpt2^PKO* mice. (C) Numbers of Pax7+ SCs per EDL myofiber at 2 weeks (WT $n = 5$, *Cpt2^PKO* $n = 4$) and 4 weeks (WT $n = 6$, *Cpt2^PKO* $n = 3$) after the last TMX injection. Error bars represent mean ± s.e.m. with $n = 3$–6 mice. n.s. no significant difference; two tailed, unpaired Student's $t$ test (for 2 weeks, WT vs. *Cpt2^PKO*, $P = 0.737$; for 4 weeks, WT vs. *Cpt2^PKO*, $P = 0.763$). (D) Representative flow cytometry results showing gating of SCs (Sca1−CD31−CD45−ITGA7+VCAM+) in mononuclear cells freshly isolated from hindlimb muscles. (E) Percentage of SCs in WT and *Cpt2^PKO* mice quantified based on flow cytometry analysis as shown in (D). Error bars represent mean ± s.e.m. with $n = 2$–4 mice. n.s. no significant difference; two tailed, unpaired Student's $t$ test (for 2 weeks, WT vs. *Cpt2^PKO*, $P = 0.179$; for 4 weeks, WT vs. *Cpt2^PKO*, $P = 0.31$). (F, G) Immunostaining of MyoD (F) and Ki67 (G), along with Pax7, in EDL myofibers freshly isolated from WT and *Cpt2^PKO* mice 4 weeks after TMX induction. Numbers were the MyoD or Ki67 positive cells among total cells counted. (H) Immunostaining of Ki67 in FACS-isolated WT and *Cpt2^PKO* SCs cultured for 72 h. Scale bar: 10 μm. (I) Quantification of total cell number. Error bars represent mean ± s.e.m. with $n = 3$ mice. **$P < 0.01$; two tailed, unpaired Student's $t$ test ($P = 0.029$). (J) Quantification of percentages of Ki67+ cells. Error bars represent mean ± s.e.m. with $n = 3$ mice (total 8 replicates for WT and 11 replicates for *Cpt2^PKO*). ***$P < 0.01$; two tailed, unpaired Student's $t$ test ($P = 3 \times 10^{-5}$).

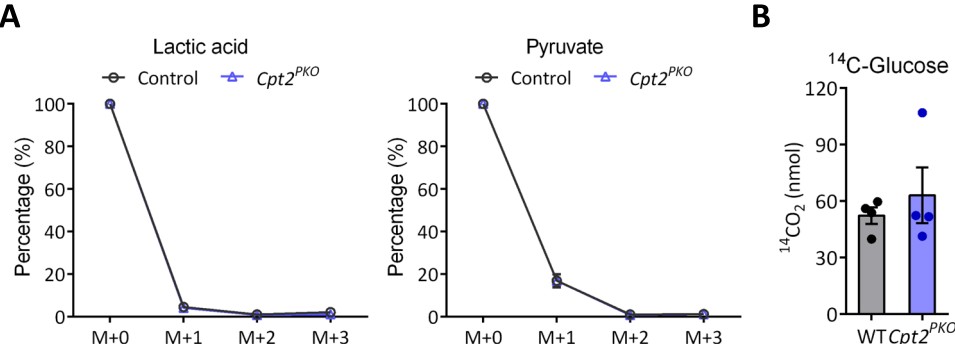

**Figure EV7.  *Cpt2* deficiency does not affect the glycolytic flux.**

(A) Targeted metabolite profiling showing no incorporation of $^{13}$C derived from $^{13}$C-PA in lactate and pyruvate in both WT and *Cpt2*-null myoblasts ($n = 4$, each group). (B) Radioactive glycolytic flux measurements using $^{14}$C-labeled glucose in WT and *Cpt2*-null satellite cells ($n = 4$, each group).

