## [Peer Review File · The EMBO Journal]

Mitochondrial fatty acid oxidation regulates adult muscle stem cell function through modulating metabolic flux and protein acetylation

Feng Yue, Lijie Gu, Jiamin Qiu, Stephanie Oprescu, Linda Beckett, Jessica Ellis, Shawn Donkin, and Shihuan Kuang

Corresponding authors: Shihuan Kuang (shihuan.kuang@duke.edu) , Feng Yue (fengyue@ufl.edu)

Review Timeline:

Submission Date:	19th Sep 24
Editorial Decision:	8th Nov 24
Revision Received:	31st Jan 25
Accepted:	10th Feb 25

Editor: Daniel Klimmeck

Transaction Report:

(Note: This manuscript was transferred to The EMBO Journal after initial review at another journal. The initial reviewers' comments and authors' responses for this article have been made available. With the exception of the correction of typographical or spelling errors that could be a source of ambiguity, letters and reports are not edited. Depending on transfer agreements, referee reports obtained elsewhere may or may not be included in this compilation. Referee reports are anonymous unless the Referee chooses to sign their reports.)

Response to the referees' comments

Reviewers' comments:

Reviewer #1 (Remarks to the Author):

This study reports that expression of *Cpt2*, a mitochondrial fatty acid oxidation gene, is increased in activated satellite cells (SCs), proliferating and committed myoblasts and differentiated myocytes but not in quiescent and self-renewing SCs. Inhibition of mitochondrial, but not peroxisomal, fatty acid oxidation (FAO) with Etomoxir reduced oxygen consumption rate (OCR), ATP production and proton leak and reduced proliferation and differentiation of treated myoblasts. Genetic deletion of *Cpt2* reduced acetyl-CoA levels, acetylation of selective proteins including Pax7, and hampered muscle regeneration. Acetate supplementation restored the energetic flux and proliferation of *Cpt2*-null SCs and the regenerative defects of *Cpt2*PKO mice.

1. Figure 1 and Extended Figure 1. It is unclear how the signatures for the six unique subsets were obtained. Top 20-30 DEGs for the respective subsets should be indicated. Previous studies (PMID: 23810552; PMID: 25600643; PMID: 26738589; PMID: 30054310; PMID: 31242425) have identified an overall increased expression of all metabolic pathways (glycolysis, TCA cycle, mitochondrial FAO and peroxisomal lipid metabolism) in activated/proliferating SCs. Density plots reported in Extended Figure 1b indicate decreased expression of "Glycolytic process" genes and no change of "Lactate production" genes in ASC compared to QSC. 5.5-days post injury (dpi) likely misses the most initial phases of SC activation. In contrast to this study, expression of peroxisomal FAO genes has been reported to increase in 3 and 5 dpi SCs. An important functional contribution of glycolysis during muscle regeneration has been documented by increased ECAR of cultured SCs (PMID: 25600643) and of 3 dpi. SCs compared to quiescent SCs (PMID: 30054310). Glycolysis is essential for SC activation (PMID: 30979776). Since ECAR is primarily a measure of lactate production, it is surprising that *Ldha* expression, found to be increased in four independent studies, was not increased in any of the different cell states (Extended Data Fig.1).

Authors' response: We appreciate the reviewer's comment. SC clusters were obtained using Seurat and manually grouped together into the 6 meta-clusters based on signature gene expression to

simplify visualization. Meta-clusters were defined by the expression of canonical marker genes known to be enriched in quiescent, activated, dividing, committed or differentiating SCs. Meta-cluster identities were also verified by marker gene expression using the Seurat function FindAllMarkers. Top marker genes were filtered for p-value < 0.001 using a Mann-Whitney Wilcoxon test and ranked based on log fold change. The top 20-30 DEGs defined by meta-cluster were consistent with known SC states and are visualized in Extended Data Fig. 1. We have added these descriptions for single-cell RNA-seq analysis in the method part of the revised manuscript.

In our current single cell RNA-seq analysis, we focused on 5.5-days post injury (dpi) in order to capture the cell fate transitions between self-renewing, committed and/or differentiating. We were not investigating the most initial phase of SC activation (activating SCs) in this manuscript. In our original analysis, the density plots were generated based on the expression of all genes in the annotated Gene Ontology terms including the genes involved in the regulation of the pathways. To clarify the changes of the glycolysis, we plotted the specific marker genes of glycolysis and found that glycolytic genes Aldoa, Pfkfb3, Pgam1, Pfkfb1, and Pfkfb2 were increased in proliferating and committed SCs compared to quiescent and self-renewed SCs (Extended Data Fig. 2c).

As previously reported, compared to quiescent SCs, ECAR and lactate production indicating anaerobic glycolysis are increased during the early activation of SCs at 3 dpi, but not in SCs at 5 dpi (PMID 30054310), which is consistent with our observation that no obvious change of Ldha in different fates of SCs collected from 5.5 dpi.

We have added these new data in our revised manuscript.

2. In Extended Data Fig.2, Etoxomir, an irreversible inhibitor of Cpt1 and mitochondrial FAO, is reported to reduce proliferation and differentiation of SCs. In a previous study, blocking mitochondrial FAO by either Etoxomir or Cpt1 knock-down had no effects on in vivo-activated SCs.

Authors' response: We thank the reviewer's comment. Based on most of the literature, the concentration of Etoxomir used for inhibition of FAO varied from 5 μ M to 200 μ M, dependent on the cell types (PMID: 31996743, PMID: 26882249, PMID: 25001241, PMID: 30043753). In the previous study (PMID 30054310), the lower concentration (5 μ M) of Etoxomir was used for cultured SCs, which in our hands was not effective in inhibiting FAO and SC proliferation. In comparison, we used 50 μ M in the present study to study SC function. To further compare the dose effects, we have performed additional Etoxomir treatments in cultured primary myoblasts using a dosage range of 10 μ M to 200 μ M. The results showed that the treatment of Etoxomir at the concentration of 50, 100, and 200 μ M inhibited the proliferation of SCs, but not the 10 μ M (Extended Data Fig. 4). Of note, no apoptotic myoblasts were observed in all the treatments. We have added these new data in our revised manuscript.

A previous study reported that Cpt1b knock-down had no effects on in SC activation, but we don't think the results are a relevant comparison to our study. First, it is common to have a delay in the clearance of target protein in miRNA KD studies, the delay time is dependent on the stability of the pre-existing target protein. In the previous study, SC activation was analyzed at 48h after miRNA induced KD of Cpt1b, and whether Cpt1b protein level is reduced was not examined in that study. In comparison, our pharmacological inhibitor (Etoxomir) is expected to exert the inhibition immediately after administration as the drug functions by directly inhibiting the enzymatic activity of Cpt1. Second, even if Cpt1b protein was successfully reduced in that study, there is no reason why other

Cpt1 isoforms (Cpt1a and Cpt1c) would not functionally compensate the loss of Cpt1b. Therefore, knock-down of Cpt1b alone may not be sufficient to block FAO in SCs, and they did not examine FAO after Cpt1b KD in that study. Third, our single-cell RNA data showed a very low expression of Cpt1b in SCs, indicating the less importance of Cpt1b in SCs. Therefore, our results obtained from either Etoxomir treatment or Cpt2 knockout were more convincing to demonstrate the importance of mitochondrial FAO in SCs.

3. Figure 3. The number of Pax7⁺ cells is reduced in 3.5 d.p.i. Cpt2PKO muscles (Figure 3a). However, though statistically significant, the difference in Edu⁺/Pax7⁺ percentage between WT and Cpt2PKO is marginal (Figure 3g), indicating that SC proliferation is only slightly affected. Do SC Cpt2PKO undergo apoptosis? On the other hand, Myogenin expression is significantly reduced in regenerating TA of Cpt2PKO (Figure 3h,i). This suggests that, rather than transient myoblast amplification, myoblast differentiation is impaired in Cpt2PKO.

Authors' response: We agree that myoblast differentiation is impaired in Cpt2^{PKO}. We have examined the apoptosis of SCs in cultured WT and Cpt2^{PKO} myofibers by performing TUNEL assay. No significant increase of apoptotic SCs was observed in Cpt2^{PKO} muscles, indicating that Cpt2^{PKO} SCs did not undergo apoptosis (Fig. 3e). Even though the percentage of EdU⁺ SCs was only marginally reduced in the Cpt2^{PKO} mice, the EdU⁺ SC number and total SC number were significantly decreased in Cpt2^{PKO} mice compared to the WT (Figure 3h). Combining our observations from myofiber culture *ex vivo* and FACS-sorted SCs *in vitro*, we concluded that SC proliferation is significantly affected by Cpt2 knockout.

4. Figure 5. Pax7 acetylation, investigated by pan-acetyl antibodies, is reduced while that of either MyoD or Myogenin is not in Cpt2PKO. The significance of these findings remains at this time associative as differentiation and fusion (Figure 3k)- which are not regulated by Pax7- rather than SC activation/proliferation (Figure 3g) seem to be affected in Cpt2PKO SCs.

Authors' response: We appreciate the reviewer's comment. Given we observed the defect of both proliferation and differentiation in Cpt2^{PKO} SCs, we examined the acetylation level of Pax7, a master transcription factor for SC proliferation and maintenance, and MyoD and MyoG, two essential transcription factors for myogenic differentiation. Our results indicate that FAO involved in the regulation of SC proliferation by affecting both energy metabolism and Pax7 acetylation, whereas it regulated the differentiation and fusion mainly by affecting energy production instead of the acetylation of MyoD and Myogenin proteins. However, we cannot exclude the possibility that the acetylation of other myogenic differentiation-related proteins was affected by FAO. Indeed, a study published by Dr. Rudnicki's lab during the revision showed that acetylation of Pax7 controls muscle stem cell self-renewal and differentiation potential in mice (PMID: 34059674). Loss of Pax7 acetylation reduced satellite cell numbers following repeated acute injury in young mice or single acute injury in aged mice. In consistent with our observations of SC phenotypes, these findings support the notion that reduced Pax7 acetylation in Cpt2^{PKO} SCs might be one of the mechanisms causing SC defect in Cpt2^{PKO} mice.

Reviewer #2 (Remarks to the Author):

This study examines the role of mitochondrial metabolic regulation of satellite cell (SC) activation and expansion following muscle injury using both *in vivo* and *in vitro* models. Using RNAseq on

single cells, the authors nicely zero in on a set of genes and surmise that acetyl-CoA metabolism is a key regulator of SC expansion. This work is clearly presented and the authors then move on to the use of the Cpt Pax-7 driven KO mouse to ascertain a role for this pathway in vivo. This work is also well executed showing clear defects following injury.

1. Here, the only question this reviewer has is that the impaired regeneration seen in the mutant mice is not as significant as one would have expected based upon the stronger results in vitro. This may suggest that in vitro conditions (i.e the availability of metabolic substrates in the media) limit the metabolic compensatory pathways that are available in vivo. While this does not require further experimentation for the scope of this study, the authors should discuss this result in the discussion (which also is a bit long and otherwise should be shortened).

Authors' response: We thank the reviewer's comment. We believe our results in Figure 2 have shown the obvious regenerative defect in Cpt2^{PKO} muscles compared to the WT muscles, especially at early regenerative stages 3.5 and 5.5 dpi, which is consistent with our in vitro study shown in Figure 3 j and k, and extended data Figures 4 and 5. We agree that SCs have different metabolic compensatory pathways between in vivo and in vitro culture conditions, which might cause the variation of certain results such like cell proliferation. We have discussed this limitation in the discussion section of the revised manuscript.

The remarkable and novel observation made by these authors is that a major target of this pathway is not simply glucose/energy metabolism (which is not really a novel thing) but that the acetylation of key regulatory genes for myogenic progression are affected. This novel observation pushes this study which is already well done, to the level I would expect (in such a venue).

Authors' response: We thank the reviewer for the appreciation of the importance and novelty of our work.

In sum, this reviewer recommends this manuscript.

Reviewer #3 (Remarks to the Author):

In the reviewed manuscript the authors use a murine model of adult muscle stem cell (satellite cell) regeneration after injury and adult murine muscle derived primary myoblasts to show that satellite cell regenerative function is dependent on fatty acid oxidation. The experiments are well presented and executed. However, a number of major and minor issues preclude my recommending the work for publication in (this venue) in its current form.

1. The authors begin by deriving metabolic states or signatures from expression data obtained from a published single cell RNA-seq dataset from a well described model of adult muscle stem cell activation after injury. The data clearly show an increase in signal of gene signatures associated with mitochondrial FAO, TCA cycle, ETC, oxidative phosphorylation as well as glycolysis and pyruvate-derived acetyl-CoA. The expression signals from these gene sets are increased from the committed stages and become even higher in differentiated satellite cells. However, numerous studies comparing expression datasets with bulk metabolomics and flux analysis of metabolic intermediates have shown that the highest determinants of flux through a metabolic pathway are substrate and product levels. In other words, flux through a metabolic pathway is primarily regulated by the levels

of substrate and product, not by the total content of the enzymatic catalyst, a proxy of which would be the mRNA level assayed in expression analysis (<https://doi.org/10.1126/science.aaf2786>). Because the authors do not perform in vivo metabolic tracing experiments, these expression signatures cannot be used to extrapolate in vivo metabolic flux. In fact, signatures derived from mRNA levels may reflect positive and negative feedback loops that greatly confound interpretations.

Authors' response: We agree that mRNA is not direct indicator, but we have used unbiased pathway analysis. Indeed, many of previous studies (PMID: 23810552; PMID: 25600643; PMID: 26738589; PMID: 30054310; PMID: 31242425) were conducted based on similar mRNA analysis, which have identified an overall increased expression of all metabolic pathways (glycolysis, TCA cycle, mitochondrial FAO and peroxisomal lipid metabolism) in activated/proliferating SCs.

To directly measure metabolic flux and satisfy the reviewer, we have performed the metabolic tracing experiments using ¹³C-labeled palmitate to compare the fatty acid metabolic flux in quiescent SCs (QSCs) and proliferating SCs (PSCs) in vivo. We found the percentage of M+2 citric acid was increased significantly in PSCs compared to QSCs (Supporting Data 1c and d), although the overall ¹³C-labeling efficiency in citric acid and malic acid were much lower compared to the in vitro study shown in Fig. 4 a and b. Somehow this data is consistent with our single-cell RNA-seq data, suggesting the increased FAO flux in PSCs compared to QSCs.

However, there is a technical limitation that the efficiency of ¹³C-incorporation in SCs is too low in vivo which results in the extremely less abundance of metabolites measured in the FACS-sorted SCs. Certainly, they are unfortunately big hurdles for performing this in vivo labeling and metabolic profiling. Due to technical limitation, we cannot provide reliable measurements and data currently. As a perspective, future study is necessary to establish the methodology to quantitate metabolic fluxes in stem cells in vivo utilizing advanced metabolite measurement technologies feasible for less biological samples or at single cell level.

Figure for reviewers removed

2. The authors do conduct metabolic tracing experiments with ¹³-C labeled palmitate in primary myoblasts showing an increase in mitochondrial oxidation of fatty acids resulting in labeled acetyl-CoA. As expected, this is impaired in myoblasts where Cpt2 is deleted. This result itself, that differentiation of committed myoblasts results in a transition towards increased mitochondrial metabolism with significant contribution of fatty acids towards acetyl-CoA levels is very clear in their data. However, the finding of increased mitochondrial metabolism in myogenic differentiation is not novel and has been described in numerous prior studies <https://doi.org/10.1152/ajpendo.00343.2001>, <https://doi.org/10.1016/j.molcel.2014.09.024>, <https://doi.org/10.1016/j.molcel.2014.09.024>

[.org/10.1074/jbc.275.4.2733](https://doi.org/10.1074/jbc.275.4.2733), <https://doi.org/10.1016/j.mce.2009.09.029>, <https://dx.doi.org/10.1038/s41598-017-18658-3>). That Cpt2 loss impairs such mitochondrial function is also not novel (<https://doi.org/10.1016/j.celrep.2014.12.023>, <https://doi.org/10.1074/jbc.m117.800839>) and in fact a 2019 study showed that compromising mitochondrial function by loss of p43 (nuclear T3 receptor TR α 1) had similar effects to those presented here for Cpt2 loss (<https://doi.org/10.1038/s41598-019-48703-2>). Mitochondrial biogenesis,

Authors' response: We cannot agree with the reviewer's comment on the novelty, which contrast the second reviewer's comments. We want to emphasize that our study for the first time identified the physiological role of fatty acid beta-oxidation mediated by CPT2 in regulating muscle stem cell function and muscle regeneration. We provide the first evidence that fatty acid metabolism is linked to protein acetylation in stem cells. The reviewer's comments of the previous studies are mostly correlation, or the study of mitochondrial function in other tissues and non-stem cells. The focus of our study is different from the listed studies.

1) We agree with the comment "the finding of increased mitochondrial metabolism in myogenic differentiation is not novel and has been described in numerous prior studies.", but we did not state that as a novelty of our study.

2) The reviewer comment "Cpt2 loss impairs such mitochondrial function is also not novel" based on two publications. However, the two publications were studied the role of Cpt2 in mature adipose tissue and heart function, which are distinct from our study in muscle stem cell metabolism.

3) p43 is a nuclear T3 receptor involving in regulating mitochondrial activity. In the publication the reviewer mentioned, global knockout p43 results in delays in muscle regeneration, increased fibrosis in injured muscle, and mildly reduction of myoblast proliferation. This work is not comparable to our finding of fatty acid oxidation in regulating muscle stem cell function.

3. The authors go on to identify an increase in protein acetylation in primary myoblasts that is dependent on Cpt2. They focus on Pax7 acetylation, which shows higher levels of protein acetylation during commitment and differentiation of myoblasts and is lost upon Cpt2 deletion. The decrease in acetyl-CoA levels resulting from Cpt2 deletion can be rescued by bypassing the need for mitochondrial FAO for generation of acetyl-CoA by supplementing the media with acetate. Exogenous acetate indeed leads to a partial rescue of the differentiation defect in vitro and a striking rescue of the regenerative defects of Cpt2 loss on satellite cells after muscle injury. However, while the authors argue that acetylation of Pax7 may be driving the differentiation program required for muscle regeneration, the authors fail to show that the rescue is accompanied by a rescue of acetylation of Pax7 in the in vitro model. Furthermore, a decrease in acetyl-CoA levels upon Cpt2 loss would be expected to have global effects as a result of decreased protein (histone, transcription factor, and many others) acetylation but also as a result of grossly impaired intermediary metabolism. No data is provided to indicate that the phenotype seen upon Cpt2 loss is a result of loss of Pax7 acetylation. While the rescue with exogenous acetate is convincing, this would rescue any phenotype that results from decreased acetyl-CoA levels. At a minimum, if the authors want to pursue the Pax7 acetylation hypothesis they should substitute the modified lysine residues in Pax7 and demonstrate a phenocopy of Cpt2 loss. Rescue of the impaired primary myoblast differentiation should also be shown upon over-expression of Pax7 or in-vitro assays with acetylated purified Pax7. It would also strengthen their hypothesis if the Cpt2 loss phenotype is rescued with histone deacetylase inhibitors. These may have effects on histone acetylation but also on non-histone protein acetylation. These

experiments could be done using the in vitro model alone but in their absence, the effect of Cpt2 loss cannot be attributed to Pax7 acetylation and remains correlative.

Authors' response: We thank the reviewer's detailed and insightful comments. The importance of the Pax7 acetylation was reported during the revision by Dr. Rudnicki's group (PMID: 34059674). The study found that acetylation of PAX7 controls muscle stem cell self-renewal and differentiation potential in mice. Loss of a single PAX7 acetylation site reduced satellite cell numbers following repeated acute injury in young mice or single acute injury in aged mice. In consistent with our observations of SC phenotypes, these findings support the notion that reduced Pax7 acetylation in Cpt2^{PKO} SCs might be one of the mechanisms causing SC defect. Besides, we observed the reduction of acetylation of selective other proteins in Cpt2^{PKO} primary myoblasts, suggesting that Pax7 acetylation is not only mechanisms contributing to the phenotype. Notably, we demonstrated the drastic energy defect in the Cpt2^{PKO} primary myoblasts while acetate supplement rescues the reduction of energy metabolism and SC defects. We thus proposed that both alterations of cellular energy metabolism and protein acetylation contribute to the phenotype we observed in Cpt2 KO SCs. We have revised the writing of the manuscript to make this statement clear.

As the decrease of Pax7 acetylation was due to the reduction of acetyl-CoA pool but not the change of the activity of histone deacetylases, we will not expect the rescue of Cpt2 KO phenotype by histone deacetylase inhibitors.

In addition to the major points described above, the following points require addressing or commenting:

- Page 5: the authors mention "These results indicate that aerobic instead of anaerobic glycolysis might be essential for SC differentiation.". As mentioned above, little knowledge can be gained about metabolic flux by looking only at expression data and hence one cannot determine whether cells are undergoing aerobic or anaerobic glycolysis in these conditions.

Authors' response: As the reviewer's suggestion, we have restated this conclusion.

- Page 6: "Ki67, a marker of mitosis". Ki67 is a marker of cell proliferation and stains positive in all cell cycle stages of proliferating cells, not just mitotic cells.

Authors' response: We agree with the reviewer and have reworded this statement to "proliferating cells".

- **Page 13:** "These unbiased gene expression analyses demonstrate that the reduced acetylation of Pax7 in Cpt2 KO myoblasts affects the normal transcriptional activity of Pax7". The gene expression analysis shows impaired expression of Pax7 targets, it does not show that the reduced expression of Pax7 is secondary to decreased acetylation. To interrogate the consequences of impaired acetylation, the authors would have to mutate acetylated lysines in Pax7 and confirm similar partial loss of function as well as show that rescue of the loss of acetylation (e.g. with SIRT1 loss, deacetylase inhibitors, in vitro acetylated protein) restores Pax7 activity (as done previously, for MyoD for example: <https://www.sciencedirect.com/science/article/pii/S1097276500803834?via=ihub>).

Authors' response: We proposed that both alterations of cellular energy metabolism and protein acetylation contribute to the phenotype we observed in Cpt2 KO SCs. We made the conclusion based

on several lines of evidence: 1) Cpt2 KO abolishes Pax7 acetylation in SCs, 2) Lack of Pax7 acetylation affects the expression of its target genes (PMID: 34059674), 3) the integrated correlation analysis with our Cpt2^{PKO} RNA-seq and the RNA-seq data from Pax7-overexpression suggested the impaired expression of Pax7 targets. As addressed in the major comment above, Dr. Rudnicki's has now demonstrated that loss of a PAX7 acetylation reduced satellite cell numbers following repeated acute injury in young mice or single acute injury in aged mice, which is consistent with our observations of SC phenotypes in Cpt2^{PKO} mice. We have revised the conclusion accordingly.

- Page 16. In Ref 37, etomoxir results in increased levels of Pax7 mRNA after injury. Can the authors comment on this possible discrepancy with their own data?

Authors' response: In Ref 37, etomoxir was intraperitoneally (IP) injected in mice and the expression of Pax7 mRNA was examined in SCs isolated from the injured muscles. Increased Pax7 expression was detected in etomoxir-treated mice (Figure 6I). However, as shown in Figure 5D in Ref 37, the expression of Pax7 mRNA was not changed in etomoxir-treated satellite cells compared to control. Given the IP injection of etomoxir potentially causes unexpected systemic effects, including at least the dramatic decrease of Sca1⁺ cells in the muscles (Figure 6G). Therefore, the results in Ref 37 may have been confounded by many factors.

- Page 16. "Our analysis (...) provides a greater resolution to capture metabolic dynamics during SC fate transitions". As discussed above in major points, in the absence of validation with metabolic flux, the scRNA-seq data cannot be used to determine whether cells are undergoing more or less glycolysis, FAO, glucose oxidation, etc. In fact, while the submitted manuscript shows clearly impaired mitochondrial FAO in proliferating myofibers and myoblasts when Cpt2 is deleted, this dependence on mitochondrial FAO is not particularly evident on the scRNA-seq data for ASCs and PSCs. Conversely, the scRNA-seq data show marked increases in 'Glycolytic process' and 'Pyruvate-derived Acetyl-CoA' signatures. Have the authors looked to see whether glucose derived carbons contribute significantly to Acetyl-CoA during SCs activation, proliferation and differentiation in their model? Of note, in Ref 43, the authors show that activated/proliferating SCs use glucose-derived carbons for citrate synthesis that serves in part to enhance histone acetylation. Upon differentiation, respiration is enhanced, promoting oxidation of glucose and a decrease in histone acetylation.

Authors' response: We did not examine the contribution of glucose-derived carbons to acetyl-CoA during SC fate transitions.

- Another recently published report (<https://doi.org/10.1016/j.celrep.2017.11.038>) shows that ACLY is required for efficient SC differentiation but does not indicate whether this is at the stage of activation/proliferation or following cell-cycle exit. Since ACLY would also be required for the utilization of fatty-acid-derived carbon towards acetylation reactions, these studies warrant a more detailed look at whether at different steps in the SC differentiation cascade (at least those that can be modelled in vitro), glucose and fatty acids are interchangeable with regards to generation of acetyl-CoA.

Authors' response: This is a great idea to study the exact contribution of glucose and fatty acids to the generation of acetyl-CoA and the protein acetylation at different myogenic states. Previous work used pharmacological ACLY inhibitor and overexpression models. In a separate project, we recently generated Acly conditional KO mice and unfortunately Pax7CreER-driven Acly KO had no effect on satellite cells. We did not include the data as it is not directly relevant to the current study.

Figure 1. As the authors are studying pathways related to mitochondrial metabolism, it is critical that they show how mitochondrial content (e.g. mtDNA) changes between the quiescent, self-renewal, activated, proliferating, committed and differentiating stages. An alternative interpretation of their scRNA seq data would be that mtDNA content increases as cells become activated and ultimately differentiate. This should be assayed both in primary single myofibers as well as primary myoblasts driven to differentiate. Prior studies have shown dramatic increases in mitochondrial content and the scRNA data would be most consistent with this (<https://doi.org/10.1074/jbc.m112.441147>). Fig 1H should also show a western blot of a key mitochondrial protein e.g. components of the ETC.

Authors' response: We thank the reviewer's thoughtful comment. We agree that it will be great of interest to study mitochondrial content changes and how it corresponds to the metabolic properties of SCs at different status. However, it is technically difficult to isolate SCs precisely at each single state for mtDNA analysis. Interestingly, a new study published by Dr. Pura Muñoz-Cánoves's group discovered the dynamics of mitochondrial organization and content in quiescence and activated SCs (PMID: 35998641). The loss of mitochondrial fission in satellite cells due to aging or genetic impairment-deregulates the mitochondrial electron transport chain (ETC), leading to inefficient oxidative phosphorylation (OXPHOS) metabolism. This new study supports our findings that mitochondrial oxidative phosphorylation is crucial for SC function. In addition, we have performed new western blot analysis showed the increase of mitochondrial proteins during myoblast differentiation (Fig. 1h).

Extended Data Fig 2. Panel H. Given the slight decrease in number of proliferating cells shown in panels E-G, the decrease in MF20 may be a result of impaired myoblast proliferation and decreased cells available for myotube fusion AND/OR impaired differentiation. The authors should perform this assay as done for the Cpt2 knockouts in Fig.2 J-K to distinguish between these possibilities.

Authors' response: We need to clarify that in the original Extended Data Fig 2. Panel H, the Etomoxir was administered at the beginning of differentiation when the proliferation of myoblast was already paused. Therefore, the impaired differentiation in Etomoxir-treated cells was not resulted from the difference in proliferation. As the reviewer suggested, we have provided the new results showing the immunostaining of myotubes and the quantification of differentiation and fusion index (Extended Data Fig. 5a). These data were included in the new Extended Data Fig. 5a.

Fig 2. The authors do not show the efficiency of Cpt2 loss in conditional knockouts. Pax7 and Cpt2 IF should be done to determine the fraction of Pax7 cells that have lost Cpt2 after tamoxifen alone (as in extended Fig 2) or after injury. Does the fraction of remaining Pax7 cells have some remaining Cpt2 function in vivo? Similarly, the authors should show that Pax7 purified SCs have lost Cpt2 in conditional KO's treated with tamoxifen as well as in the muscle injury experiment. Cpt2 loss should ideally be demonstrated by loss of protein levels (or band shift if the knockout does not result in loss of protein) or by mRNA levels.

Authors' response: We have sorted the SCs and performed qPCR analysis to confirm the efficiency of Cpt2 KO. The results showed that 98% reduction of Cpt2 mRNA in FACS-sorted SCs at 3.5 dpi of CTX (Extended Data Fig. 6a), suggesting the extremely high efficiency of Cpt2 gene KO.

Fig 3. It is interesting that despite a considerable decrease in Pax7+ cells in Cpt2PKO muscle after injury, the fraction of EdU+ cells is still very high (3F). Can the authors comment on this discrepancy?

This may be related to the earlier point of how many of these Pax7⁺ cells have actually lost Cpt2 and hence showing levels of remaining Cpt2 by IF is key. This also applies to Ext Fig 3G-I.

Authors' response: The original Fig. 3F showed the percentage of EdU⁺ SCs in the muscle section. Indeed, the total number of EdU⁺ SCs is reduced remarkably in muscles at 3.5 dpi, although the percentage of EdU⁺ SCs was comparable (new Fig. 3h). We speculate that Cpt2 KO slowed down rather than completely blocked the proliferation. As responded above, we have performed qPCR to confirm the high efficiency of Cpt2 deletion in SCs (Extended Data Fig. 6a).

Extended Fig 4. The authors use ¹³C-PA incorporation into lactic acid and pyruvate as well as ¹⁴C-labeled glucose oxidation to CO₂ to estimate glycolytic flux. Fatty acids contribute minimally to lactic acid and pyruvate through malic enzyme and/or the malate/oxaloacetate shuttle and complete glucose oxidation cannot distinguish between glycolysis and mitochondrial oxidation. The authors should perform ¹³C-uniformly labeled glucose tracing to determine whether there are changes in glycolytic flux in Cpt2 null primary myoblasts.

Authors' response: We believe our measurement of ¹⁴C-labeled glucose oxidation to CO₂ indicates no changes of glycolytic flux in Cpt2 null primary myoblasts (Extended Data Fig. 7b).

Figure 4. As the authors are claiming that Cpt2 results in energy insufficiency, does the decrease in fatty acid oxidation and ATP levels result in activation of the energy sensing pathways in the cell? The authors could look at ATP/AMP ratios in cells by LCMS and at AMPK activation to determine whether Cpt2 results in significant energetic stress.

Authors' response: We appreciate the reviewer's comment but changes in ATP levels have been shown to activate the energy sensing pathway, which is out of the scope of the current manuscript.

Figure 5. Western blots in panels D (IB: Pax7, input) and E (IP: Pax7, IB: Pax7) appear to be overexposed beyond the linear range. Can the authors include shorter exposures? Panel F: quantification of overlapping puncta carried out blindly should be included.

Authors' response: We have added the quantification of PLA puncta for Panel F. The exposure appeared to be fine and we will upload original images in the final version.

Figure 6. Panels E-F. Have the authors performed the acetate rescue experiment with primary myoblasts, where the proliferation and differentiation defect is more marked than in the cultured myofibers? As shown, the myofiber IF images are difficult to interpret and the quantification shows very subtle increases in Cpt2 null clusters upon acetate addition.

Authors' response: We have addressed this concern by performing acetate rescue assay using FACS-sorted SCs. In consistent with our observations in cultured myofiber, we found similar results that acetate supplementation significantly restored the SC proliferation capacity (Fig. 6g). We have added this new data in the revised manuscript.

Figure 7. Can the authors quantify mononuclear cell infiltrates from different fields to get a better representative picture of the magnitude of this effect?

Authors' response: As suggested, we have quantified the infiltrated mononuclear cells based on the staining of dystrophin and DPAI on muscle sections. The result showed that the number of infiltrated mononuclear cells was significant higher in Cpt2PKO muscle after injury compared to the control mice, while the acetate supplementation normalized the number of infiltrated mononuclear cells to the control level. We have added these results in the revised manuscript.

Dear Dr Kuang,

Thank you again for the submission of your amended manuscript (EMBOJ-2024-119056-T) to The EMBO Journal. We have carefully assessed your manuscript, and the point-by-point response provided to the referee concerns that were raised during review at a different journal. In addition, and as mentioned before, we decided to send the revised version of your work back to the original reviewers for their reassessment with respect to technical robustness, conceptual advance and overall suitability of your work for publication in The EMBO Journal. We have received two re-reports which I enclose below. Please note that while reviewer #2 was at this time not able to assess your amended study, we have considered your response editorially and found the concerns to be addressed satisfactorily. As you will see from the other experts' comments, they are now in favour of the work and supportive of publication at The EMBO Journal, pending minor revision.

We are thus pleased to inform you that we can offer to swiftly move forward towards acceptance of this work at The EMBO Journal, pending minor revision of the following remaining issues, which need to be adjusted in a re-submitted version.

Please consider the remaining points by the referees carefully and introduce complementary experiments or adjust the data presentation and discussion of the results, where appropriate.

We also need you to take care of a number of minor issues related to formatting and data annotation, as detailed below.

Please submit a revised version of the manuscript using the link enclosed below, addressing the advisor's comments.

As you might have seen on our web page, every paper at the EMBO Journal now includes a 'Synopsis', displayed on the html and freely accessible to all readers. The synopsis includes a 'model' figure as well as 2-5 one-short-sentence bullet points that summarize the article. I would appreciate if you could provide this figure and the bullet points.

Please let me know any time should you have additional questions regarding above points.

Thank you again for giving us the chance to consider your manuscript for The EMBO Journal, I look forward to hearing from you and receiving your final revised version of the manuscript.

Kind regards,

Daniel Klimmeck

Daniel Klimmeck PhD
Senior Editor
The EMBO Journal.

>> Please add up to five keywords to your study.

>> Author Contributions: Please remove the author contributions information from the manuscript text. Note that CRediT has replaced the traditional author contributions section as of now because it offers a systematic machine-readable author contributions format that allows for more effective research assessment. and use the free text boxes beneath each contributing author's name to add specific details on the author's contribution.

More information is available in our guide to authors.
<https://www.embopress.org/page/journal/14602075/authorguide>

>> Adjust the title of the 'Competing Interests' section to 'Disclosure and Competing Interests Statement' and move after Acknowledgements.

>> Correct order of manuscript sections: Abstract / Keywords / Introduction / Results / Discussion / Methods / Data Availability / Acknowledgements / Disclosure and competing interests statement // References / Figure legends / Tables and their legends / Expanded View Figure legends

>> Provide a completed Author Checklist.

>> Figure callouts: Please ensure that the figures and panels are called out in sequential order. Currently, Fig 2A-E are called out before Fig. 1C; Fig 2 F-I and Fig 3 E-L are called out after Fig 6, etc. .

>> Figures in separate files: Figures should be removed from the manuscript text and uploaded as individual, high resolution figure files. Legends should be placed after the References.

>> The "additional information" section in the manuscript text can be deleted.

>> Please provide source data for the study as to the separate request e-mail by my colleague Hannah Sonntag.

>>Appendix file with ToC: the file with suppl. information should be renamed "Appendix" and uploaded as a PDF. Please rename the suppl. tables "Appendix Table S1" etc., and the suppl. figures "Appendix Figure S1". Please add a table of contents, including page numbers.

>> Funding: please enter the funding following information in the list of funders in our online system: Muscular Dystrophy Association (MDA516161) and NIH Shared Instrumentation Grant S10DO20029.

>> References: adjust reference format to EMBO Journal format, 10 authors et al, and place References after the Discussion, before figure legends.

>> Data availability section: provide public access to the RNAseq datasets, including URL and dataset ID.

>> Add a Reagents and Tools table to the Methods section, listing key reagents, experimental models, software and relevant equipment.

>> Avoid textual redundancy in the introduction, results and discussion sections with your earlier 2017 study (PMID 28094257).

>> Consider additional changes and comments from our production team as indicated below:

- Figure legends:

1. Please define the annotated p values * as well as provide the exact p-values for the same in the legends of figures EV4 B, EV5 C as appropriate.
2. Please note that the exact p values are not provided in the legends of figures 1E, 2C, F, G, H, I; 3B, D, H, J, L; 4C, D, F, G; 5A; 6A, B, D, F, G; 7D, E, G, I; EV3 B, EV4 A-C; EV5 A, C; EV6 I, J.
3. Please indicate what */ **/ ***/ **** represents; if this represents p value(s), please indicate the statistical test used and where appropriate, specify exact p value in the legends of figures 5F, G.
4. Please define the annotated p values * as well as provide the exact p-values for the same in the legends of figures EV4 B, EV5 C as appropriate.
5. Please note that the exact p values are not provided in the legends of figures 1E, 2C, F, G, H, I; 3B, D, H, J, L; 4C, D, F, G; 5A; 6A, B, D, F, G; 7D, E, G, I; EV3 B, EV4 A-C; EV5 A, C; EV6 I, J.
6. Please indicate what */ **/ ***/ **** represents; if this represents p value(s), please indicate the statistical test used and where appropriate, specify exact p value in the legends of figures 5F, G.
7. Please note that the scale bar is missing for figures 1F, EV4 B; EV5 A, C.

Referee #1:

This study reports that expression of Cpt2, a mitochondrial fatty acid oxidation gene, is increased in activated satellite cells (SCs), proliferating and committed myoblasts and differentiated myocytes but not in quiescent and self-renewing SCs. Inhibition of mitochondrial, but not peroxisomal, fatty acid oxidation (FAO) with Etomoxir reduced oxygen consumption rate (OCR), ATP production and proton leak and reduced proliferation and differentiation of treated myoblasts. Genetic deletion of Cpt2 reduced acetyl-CoA levels, acetylation of selective proteins including Pax7, and hampered muscle regeneration. Acetate supplementation restored the energetic flux and proliferation of Cpt2-null SCs and the regenerative defects of Cpt2PKO mice.

COMMENTS:

Figure 5d. The authors state that: " By immunoprecipitation with acetylated-lysine antibody, a quite low level of acetylated-MyoD and acetylated-MyoG was detected in WT primary myoblast (Fig. 5d). No differences in the acetylation of MyoD and MyoG were observed between WT and Cpt2PKO primary myoblasts (Fig. 5d). In contrast, we detected an abundant level of acetylated-Pax7 protein in WT primary myoblast (Fig. 5d).

If anything, MyoD seems to be highly acetylated compared to Pax7. To correctly interpret the data, acetylated MyoD and Pax7 should be normalized for total input MyoD and Pax7. Visually, it seems that majority of input MyoD is acetylated whereas only a fraction of total input Pax7 is acetylated. The authors should present quantification of acetylated/total MyoD and Pax7.

It should also be noted that, while being also expressed in activated satellite cells/myoblasts, Pax7 is mostly enriched in quiescent satellite cells. On the contrary, MyoD protein is exclusively translated in activated satellite cells/myoblasts and absent in quiescent satellite cells. Both MyoD and Pax7 acetylation have been reported to have a functional role. Since Pax7 acetylation- but not MyoD acetylation- is affected by Cpt2, this may argue for a functional role of Cpt2 in quiescent satellite cells rather than in proliferating myoblasts.

Referee #3:

Feng Yue et al describe a role for mitochondrial fatty acid oxidation (FAO) in the activation, proliferation and differentiation of satellite cells (SCs) of skeletal muscle. They identify differential enrichment of a transcriptional signature associated with FAO and acetyl-CoA metabolism in proliferating and committed SCs as compared to quiescent and self-renewing SCs. Using a Cpt2-KO murine model, the authors propose that the reduced pool of FA-derived acetyl-CoA is responsible for defective muscle regeneration through impaired bioenergetics and protein acetylation, including that of myogenic regulator PAX7.

The experiments are rigorously conducted and mostly well controlled. My main concern with the manuscript in its current form is that the effect of FAO on satellite cell activation and proliferation could be a result of energetics (i.e. FAO for NADH and ATP generation in the mitochondria), acetylation of PAX7 and other proteins after export of acetyl-CoA to the cytoplasm, a third not specified function, or a combination of these three. The authors do rescue the effect of Cpt2-KO with acetate supplementation but this would rescue all these possibilities and no rescue experiments are performed to tease apart which of these downstream consequences is primarily responsible for the muscle differentiation phenotypes. I think this limitation needs to be clearly underscored in the discussion.

Other concerns and comments follow:

Figure 1. Conclusion in the text "(...) these data suggest that the expansion and commitment of SCs is accompanied by increased mitochondrial FAO." This sentence is misleading. Expansion and commitment are accompanied by increased transcription of genes coding for enzymes associated with FAO but FAO per se is not shown in this figure.

Extended Data Figure 3d. Why is Maximal Respiration unaffected by etomoxir in differentiated myotubes?

Extended Data Figure 4a-c. Images are low quality and seemingly out of focus.

Extended Data Figure 5. Images have no scale bar. Differences in MF20 at IF levels are very subtle for etomoxir. Can the authors comment?

Page 12. "To determine if the reduced FAO stimulates a compensatory increase in glycolytic activity, we measured $^{14}\text{CO}_2$ production derived from ^{14}C -labeled glucose". Do the authors mean 'a compensatory increase in glucose oxidation'? $^{14}\text{CO}_2$ production would result from the full oxidation of labeled glucose, not from glycolysis alone.

Cpt2 KO should primarily/directly affect the mitochondrial pool of acetyl-CoA. In Figure 5a-c, the authors argue that this in turn also affects the cytoplasmic pool. Given the clear energetic effects of Cpt2 loss shown in Figure 4 it would be important to assess whether replenishing the cytoplasmic pools of acetyl-CoA can rescue Cpt2KO in the presence of an ETC inhibitor. Alternatively, does loss of SLC25A1, the mitochondrial citrate transporter, phenocopy Cpt2 Kos in satellite cells?

In Figure 5H-I, the authors conduct RNA-seq analysis to determine whether changes in Cpt2 KO versus WT are a result of altered Pax7 acetylation and thus Pax7 function. However, this experiment would not differentiate between Pax7 dependent effects and effects that result from altered energetics as a result of Cpt2 loss.

In the discussion the authors mention (page 18) that "scRNA-seq data obtained from SCs at different regenerating stages provides a greater resolution to capture metabolic dynamics during SC fate transitions." However, RNA expression data cannot be used to infer metabolic flux and metabolic flux data provided in the paper does not match the resolution of the scRNAseq. Therefore, I would be careful to make these sort of drastic extrapolations from transcription to metabolic flux.

In page 19 the authors state "the change of acetyl-CoA level and metabolic flux in SCs caused by genetic Cpt2 knockout is reversible, as supplementation of SCFA acetate restored the acetyl-CoA pool and metabolic flux in Cpt2-deficient SCs, and partially rescues their regenerative defects in vivo." I don't believe the acetate rescue shows reversibility. Rather, it suggests that the effects are dependent on acetyl-CoA levels being maintained. Reversibility would require injury in the knockout followed by impaired regeneration and then reinjury with added acetate and a rescue of the regeneration defect.

On page 5: "Overall, genes of the "total glycolytic process" were enriched only in DSCs, but not in other states (Extended Data Fig. 2a,b). While we plotted the specific marker genes of glycolysis and found that glycolytic genes Aldoa, Pfkfb3, Pgk1, Pkm, and Pfkfb1 were increased in proliferating and committed SCs compared to quiescent and self-renewed SCs"" as shown in Extended Data Figure 2b. What is the difference between the two gene sets? Why is the overall glycolytic signature only higher in DSCs while individual glycolysis-associated genes are not?

Pages 8-9: The authors state that Cpt2-KO does not affect the pool of SCs, but delays their exit from quiescence (Extended Data Figure 6 + Figure 2). Can the authors speculate on how this exit is affected in Cpt2-KO SCs? Is the activation of these cells dependent on the transcriptional activity of ac-Pax7?

In contrast to quiescent cells, proliferating cells require acetyl-CoA for lipogenesis (for cell/organelle membranes). Do the authors think that the phenotype associated with proliferating Cpt2-KO SCs may be associated with reduced fatty acid synthesis/lipogenesis? Does FASN inhibition phenocopy the Cpt2 KO phenotype?

Response to reviewers' comments

Referee #1:

COMMENTS:

Figure 5d. The authors state that: " By immunoprecipitation with acetylated-lysine antibody, a quite low level of acetylated-MyoD and acetylated-MyoG was detected in WT primary myoblast (Fig. 5d). No differences in the acetylation of MyoD and MyoG were observed between WT and Cpt2PKO primary myoblasts (Fig. 5d). In contrast, we detected an abundant level of acetylated-Pax7 protein in WT primary myoblast (Fig. 5d).

If anything, MyoD seems to be highly acetylated compared to Pax7. To correctly interpret the data, acetylated MyoD and Pax7 should be normalized for total input MyoD and Pax7. Visually, it seems that majority of input MyoD is acetylated whereas only a fraction of total input Pax7 is acetylated. The authors should present quantification of acetylated/total MyoD and Pax7.

Authors' response: We thank you for the comment, which reminds us that the comparisons that we made on the relative acetylated levels of different proteins were not appropriate, given that the different antibodies may exhibit different affinity. The appropriate way is to compare the ratios of the acetylated protein to total protein as you suggested, which we have quantified in the revised version (we couldn't quantify MyoG because there were no acetylated bands). We have removed the statement about low levels of acetylated MyoD and MyoG and high levels of Pax7, and also reworded the sentences to better clarify the results.

It should also be noted that, while being also expressed in activated satellite cells/myoblasts, Pax7 is mostly enriched in quiescent satellite cells. On the contrary, MyoD protein is exclusively translated in activated satellite cells/myoblasts and absent in quiescent satellite cells. Both MyoD and Pax7 acetylation have been reported to have a functional role. Since Pax7 acetylation- but not MyoD acetylation- is affected by Cpt2, this may argue for a functional role of Cpt2 in quiescent satellite cells rather than in proliferating myoblasts.

Authors' response: Thank you for this insightful comment. We agree that Pax7 is "mostly enriched" in QSCs, which does not negate the fact that the Pax7 is highly expressed in activated and proliferating SCs. Given the observation that CPT2 protein is only detected by WB and IF in activated and proliferating satellite cells but not in QSCs, it is very unlikely that Cpt2 plays a functional role in QSCs.

Supporting our notion, a recent study from Michael Rudnicki's group reported that abolishing PAX7 acetylation in mice did not affect the number of quiescent satellite cells, but it did impact the self-renewal capacity of proliferating satellite cells (PMID: 34059674). Based on this evidence, it could be speculated that the transcriptional regulatory function of PAX7 differs between quiescent and activated/proliferating satellite cells, with acetylation of PAX7 possibly acting as a regulator that determines these distinct functions. It would be interesting to define in future studies how the acetylation of PAX7 and MyoD is regulated by metabolic pathways. We have included this discussion in the revised manuscript.

Referee #3:

The experiments are rigorously conducted and mostly well controlled. My main concern with the manuscript in its current form is that the effect of FAO on satellite cell activation and proliferation could be a result of energetics (i.e. FAO for NADH and ATP generation in the mitochondria), acetylation of PAX7 and other proteins after export of acetyl-CoA to the cytoplasm, a third not specified function, or a combination of these three. The authors do rescue the effect of Cpt2-KO with acetate supplementation but this would rescue all these possibilities and no rescue experiments are performed to tease apart which of these downstream consequences is primarily responsible for the muscle differentiation phenotypes. I think this limitation needs to be clearly underscored in the discussion.

Authors' response: Thank you for the comment. We agree and have included the limitation raised by the reviewer in the discussion.

“Thus, the effect of mitochondrial FAO on SC activation and proliferation could be a result of combined defects in energy production and protein acetylation, or even through other yet to be defined mechanisms mediated by bioactive lipids and metabolites. Although acetate supplementation rescued the effect of Cpt2 KO on SCs, whether the rescue is mainly mediated through energetic, acetylation or other pathways remained to be dissected.”

Other concerns and comments follow:

Figure 1. Conclusion in the text "(...) these data suggest that the expansion and commitment of SCs is accompanied by increased mitochondrial FAO." This sentence is misleading. Expansion and commitment are accompanied by increased transcription of genes coding for enzymes associated with FAO but FAO per se is not shown in this figure.

Authors' response: We have revised the sentence to make the conclusion clearer and more accurate.

Extended Data Figure 3d. Why is Maximal Respiration unaffected by etomoxir in differentiated myotubes?

Authors' response: We are not entirely sure but one potential explanation is that etomoxir treatment triggered cell stress, which activated metabolic pathways that help the cells to adapt to or compensate for the energy crisis. Our result showed that Etomoxir inhibited OCR for basal and proton leak in myotubes, but increased the spare capacity. Since the maximal respiration is composed of basal respiration, spare capacity, and proton leak, the increased spare capacity may have compensated for the reduced basal respiration, leading to similar levels of maximal respiration.

Extended Data Figure 4a-c. Images are low quality and seemingly out of focus.

Authors' response: This may be due to PDF conversion and compression of file size. As the file size of the supplemental figures is over 200 Mb, the PDF was compressed. We will ensure that the images maintain the same quality as they are converted to PDF.

Extended Data Figure 5. Images have no scale bar. Differences in MF20 at IF levels are very subtle for etomoxir. Can the authors comment?

Authors' response: We have added the scale bar in Extended Data Figure 5. The MF20 IF intensity mainly reflects myosin level within the positively stained area. Despite similar IF levels, the MF20

protein level is significantly lower in the Etomoxir treated group because of the clear reduction in the sizes (area) of MF20⁺ myotubes (protein level should be positively correlated to IF intensity x area).

Page 12. "To determine if the reduced FAO stimulates a compensatory increase in glycolytic activity, we measured ¹⁴CO₂ production derived from ¹⁴C-labeled glucose". Do the authors mean 'a compensatory increase in glucose oxidation'? ¹⁴CO₂ production would result from the full oxidation of labeled glucose, not from glycolysis alone.

Authors' response: Thank you for this comment. We have corrected this description as suggested.

Cpt2 KO should primarily/directly affect the mitochondrial pool of acetyl-CoA. In Figure 5a-c, the authors argue that this in turn also affects the cytoplasmic pool. Given the clear energetic effects of Cpt2 loss shown in Figure 4 it would be important to assess whether replenishing the cytoplasmic pools of acetyl-CoA can rescue Cpt2KO in the presence of an ETC inhibitor. Alternatively, does loss of SLC25A1, the mitochondrial citrate transporter, phenocopy Cpt2 KOs in satellite cells?

Authors' response: We appreciate the reviewer's thoughtful comments. The acetyl-CoA measurements in this study were taken at the whole-cell level. Unfortunately, there is no report on the satellite cell specific SLC25A1 KO mice and we therefore do not know if such a KO will phenocopy *Cpt2* KO. Previous work indicate that SLC25A1 mutation impairs complex V function (ATP production) (PMID: 29226520), which is consistent with our observation in the *Cpt2* KO.

In Figure 5H-I, the authors conduct RNA-seq analysis to determine whether changes in Cpt2 KO versus WT are a result of altered Pax7 acetylation and thus Pax7 function. However, this experiment would not differentiate between Pax7 dependent effects and effects that result from altered energetics as a result of Cpt2 loss.

Authors' response: We agree with the reviewer. We conducted RNA-seq analysis to show the correlation of Pax7-regulated genes and *Cpt2* KO. Our present study does not differentiate the contributions between Pax7-dependent effects and effects that result from altered energetics in determining the dysfunction of *Cpt2* KO SCs.

In the discussion the authors mention (page 18) that "scRNA-seq data obtained from SCs at different regenerating stages provides a greater resolution to capture metabolic dynamics during SC fate transitions." However, RNA expression data cannot be used to infer metabolic flux and metabolic flux data provided in the paper does not match the resolution of the scRNAseq. Therefore, I would be careful to make these sort of drastic extrapolations from transcription to metabolic flux.

Authors' response: Thank you for the comments. We have revised the text to "Our analysis of scRNA-seq data obtained from SCs at different regenerating stages provides a transcriptional snapshot of how expression of metabolic genes changes during SC fate transitions".

In page 19 the authors state "the change of acetyl-CoA level and metabolic flux in SCs caused by genetic Cpt2 knockout is reversible, as supplementation of SCFA acetate restored the acetyl-CoA pool and metabolic flux in Cptg2-deficient SCs, and partially rescues their regenerative defects in vivo." I don't believe the acetate rescue shows reversibility. Rather, it suggests that the effects are dependent on acetyl-CoA levels being maintained. Reversibility would require injury in the knockout followed by impaired regeneration and then reinjury with added acetate and a rescue of the regeneration defect.

Authors' response: Thank you! We rewrote the sentence as “Our results demonstrating that supplementation of SCFA acetate restored the acetyl-CoA pool and metabolic flux in *Cpt2*-deficient SCs, and partially rescues their regenerative defects *in vivo* suggest a causal role of deficient acetyl-CoA level in SC dysfunction in the *Cpt2* KO mice”.

On page 5: "Overall, genes of the "total glycolytic process" were enriched only in DSCs, but not in other states (Extended Data Fig. 2a,b). While we plotted the specific marker genes of glycolysis and found that glycolytic genes *Aldoa*, *Pfkfb3*, *Pgam1*, *Pkm*, and *Pgk1* were increased in proliferating and committed SCs compared to quiescent and self-renewed SCs" as shown in Extended Data Figure 2b. What is the difference between the two gene sets? Why is the overall glycolytic signature only higher in DSCs while individual glycolysis-associated genes are not?

Authors' response: The enrichment analysis (Extended Data Fig. 2b) and individual gene expression data (Extended Data Fig. 2c, top row) were performed from the same gene set for the glycolytic process (GO:0006096). The enrichment analysis considers all genes in this pathway as a whole, highlighting its association with each cell state. However, the expression changes of individual genes may differ from the overall enrichment pattern. Indeed, compared to quiescent and self-renewed SCs, we observed that the glycolytic genes *Pgam1* and *Pkm* were upregulated in PSCs, *Pfkfb3* was elevated in both PSCs and CSCs, and *Aldoa* and *Pgk1* were increased in CSCs and DSCs (Extended Data Fig. 2c). We have revised the description of these findings for greater accuracy.

Pages 8-9: The authors state that *Cpt2*-KO does not affect the pool of SCs, but delays their exit from quiescence (Extended Data Figure 6 + Figure 2). Can the authors speculate on how this exit is affected in *Cpt2*-KO SCs? Is the activation of these cells dependent on the transcriptional activity of ac-Pax7?

Authors' response: We would like to clarify that our data (Extended Data Figure 6 + Figure 3) show that loss of *Cpt2* does not affect the pool of quiescent SCs, but it does inhibit their proliferation. We don't think that 'inhibition of proliferation' means 'exit from quiescence.' Our study suggests that *Cpt2* deficiency disrupts the metabolic flux and acetylation of specific proteins in SCs, which in turn contributes to the observed proliferation defect.

In contrast to quiescent cells, proliferating cells require acetyl-CoA for lipogenesis (for cell/organelle membranes). Do the authors think that the phenotype associated with proliferating *Cpt2*-KO SCs may be associated with reduced fatty acid synthesis/lipogenesis? Does FASN inhibition phenocopy the *Cpt2* KO phenotype?

Authors' response: This is an interesting point. We did not specifically examine the fatty acid synthesis and lipogenesis in *Cpt2*-KO SCs. However, pathway analysis of our RNA-seq data did not reveal significant enrichment for these processes in *Cpt2*-KO SCs. In a separate, ongoing study, we observed that conditional knockout of ATP citrate lyase (*Acly*), a key enzyme involved in the production of acetyl-CoA from citrate in the cytosol, did not affect SC number, function, or muscle regeneration. Based on this evidence, it seems unlikely that reduced *de novo* fatty acid synthesis or lipogenesis, through inactivation of enzymes like *ACLY* or *FASN*, would phenocopy the *Cpt2*-KO phenotype.

Dear Dr Kuang,

Thank you for submitting the revised version of your manuscript. I have now evaluated your amended manuscript and concluded that the remaining minor concerns have been sufficiently addressed.

I am thus pleased to inform you that your manuscript has been accepted for publication in the EMBO Journal.

Please note also that it is The EMBO Journal policy for the transcript of the editorial process (containing referee reports and your response letter) to be published as an online supplement to each paper. More information is available here: https://www.embopress.org/transparent-process#Review_Process

Related, I would like to ask for your consent on keeping the referee figure included in this file.

On a different note, I would like to alert you that EMBO Press offers a format for a video-synopsis of work published with us, which essentially is a short, author-generated film explaining the core findings in hand drawings, and, as we believe, can be very useful to increase visibility of the work. Please see the following link for representative examples and their integration into the article web page:

<https://www.embopress.org/doi/full/10.15252/emj.2019103932>

Best regards,

Daniel Klimmeck

Daniel Klimmeck, PhD
Senior Editor
The EMBO Journal
EMBO
Postfach 1022-40
Meyerohofstrasse 1
D-69117 Heidelberg
contact@embojournal.org